# Completeness of radiosonde humidity observations based on the IGRA

António P. Ferreira, Raquel Nieto, Luis Gimeno

Environmental Physics Laboratory, Faculty of Science, University of Vigo, Ourense, 32004, Spain

*Correspondence to*: António P. Ferreira (ap.ferreira@uvigo.es)

**Abstract.** Radiosonde measurements from the 1930s to present give unique information on the distribution and variability of water vapor in the troposphere. The data of the Integrated Global Radiosonde Archive (IGRA) Version 2 are examined here
until the end of 2016, aiming to describe the completeness of humidity observations (simultaneous measurements of pressure, temperature and humidity) in different times and locations. Upon finding the stations with a non-negligible amount of radiosonde observations in their period of record, thus removing pilot-balloon stations from IGRA, the selected set (designated IGRA-RS) comprises 1723 stations, including 1300 WMO stations, of which 178 belong to the current GCOS Upper-Air Network (GUAN) and 16 to the GCOS Reference Upper-Air Network (GRUAN). Completeness of humidity observations for
a radiosonde station and a full year is herein defined by five basic parameters: number of humidity soundings; fraction of days with humidity data; average vertical resolution; average atmospheric pressure and altitude at the highest measuring level; and maximum number of consecutive days without data. The observations eligible for calculating precipitable water vapor – i.e., having adequate vertical sampling between the surface and 500 hPa – are particularly studied. The present study presents the global coverage of humidity data and an overall picture of the temporal and vertical completeness parameters over time. This
overview indicates that the number of radiosonde stations potentially useful for climate studies involving humidity depends not only on their record length, but also on the continuity, regularity and vertical sampling of the humidity time-series. Furthermore, a dataset is provided with the purpose of helping climate and environmental scientists to select radiosonde data according to various completeness criteria – even if differences in instrumentation and observing practices require extra attention. This dataset consists of two main sub-sets: 1) statistical metadata for each IGRA-RS station and year within period
of record; and 2) metadata for individual observations from each station. These are complemented by 3) a list of the stations represented in the whole dataset, along with the observing periods for humidity (relative humidity or dew-point depression) and the corresponding counts of observations. The dataset is to be updated on a two-year basis, starting in 2019, and is available at https://doi.org/10.5281/zenodo.1332686.

## 1 Introduction

For more than three-quarters of a century, the global radiosonde network designed and developed for weather forecasting has provided in situ observations of humidity from the surface up the middle troposphere, eventually reaching the stratosphere.

Satellite-based remote sensing of atmospheric water vapor is part of modern weather forecasting and climate monitoring (Kley et al., 2000; Andersson et al., 2007). In the present state-of-the-art, some satellite retrievals of moisture-related quantities are used as a reference to compare humidity measurements from different radiosonde types, aiming to monitor radiosonde stations and improve satellite calibration (Kuo et al., 2005; John and Buehler, 2005; Sun et al., 2010; Moradi et al. 2013). However, limb-sounding satellite techniques with high vertical resolution (albeit very course in the horizontal), using GPS radio-occultation, are a recent acquisition, of main interest to access water vapor in the upper troposphere and lower stratosphere so far (Kishore et al., 2010; Shangguan et al., 2016; Rieckh et al., 2018; Vergados et al., 2018). Reanalysis outputs based on past radiosonde data, assimilating also satellite data when available, offer multiple-level, global gridded, synoptic-scale moisture fields up to 4 times daily from a beginning year (e.g.: 1948 in NCEP/NCAR Reanalysis 1; 1979 in NCEP/NCAR Reanalysis 2 and ECWMF's ERA Interim) to present time – even though radiosonde observations are scarce over the ocean, unevenly spaced over land, and taken normally twice a day, with significant differences in vertical coverage. Naturally, since air moisture is highly variable in time and space, humidity data from different reanalysis models show discrepancies and can differ significantly from the collocated radiosonde data (e.g., Noh et al., 2016). Therefore, the radiosonde archives represent the primary source of information on the short and long-term distribution of moisture in the troposphere, despite various data inhomogeneities. Namely, geographical-temporal sampling differences (Wallis, 1998), uncertainties related to observation time and balloon drift (Kitchen, 1989; McGrath et al., 2006; Seidel et al., 2011; Laroche and Sarrazin, 2013), differences in vertical coverage and data gaps related to reporting practices of humidity (Dai et al. 2011 and references therein) and differences in humidity data accuracy – which depend on humidity sensors and vary with measured conditions (WMO, 1995; Nash, 2002; Sappuci et al., 2005; Moradi et al., 2013; Dirksen et al., 2014).

The growth interest in climate change motivated a renewed attention to radiosonde data since the 1990s. Soon it was realized that the usefulness of radiosonde data archives to investigate climate trends relies on homogenization procedures to overcome biases and sudden shifts arising from instrument changes, reporting practices and sampling differences (Elliott and Gaffen, 1991; Schwartz and Doswell, 1991; Parker and Cox, 1995; Luers and Eskridge, 1998; Lanzante et al. 2003). Subsequent climate studies based on radiosondes have mostly focused on the detection of climate change in temperature trends (Free and Seidel, 2005; Thorne et al., 2005; Haimberger et al. 2008). Concerning humidity, radiosonde-based climatic studies are for now confined to the lower and middle troposphere, because of the large uncertainty of measurements and biases in the upper troposphere and lower stratosphere (Elliot and Gaffen, 1991; Soden and Lanzante, 1996; Wang et al., 2003) and the extremely large relative biases and insufficient data in the lower stratosphere (Miloshevich et al., 2006; Nash et al. 2011). Radiosonde data have been used for studying the long-term trends and the annual cycle of several humidity parameters (precipitable water vapor, specific humidity and relative humidity), at least in well-sampled regions of the globe and if data inhomogeneities are removed (Elliot et al., 1991; Gaffen et al., 1992; Ross and Elliott, 1996; Ross and Elliot, 2001; McCarthy el al., 2009; Durre et al., 2009; Dai et al., 2011). On a rather different scale, radiosonde measurements with high vertical resolution near the ground are suitable for studying water evaporation over land and the structure of the planetary boundary

layer, provided that the time-lag of humidity sensors as they move through a rapidly changing environment is accounted for (Sugita and Brutsaert, 1991; Connell and Miller, 1995; Seidel et al., 2010).

Since its creation in 2004, the Integrated Global Radiosonde Archive (IGRA) was meant to be the largest data set of up-to-date weather-balloon observations freely available, by collecting quality-controlled data from upper-air stations across all continents. The first version of IGRA (a successor of the Comprehensive Aerological Reference Data Set (CARDS; Eskridge et al., 1995) contained practically data after 1945 (Durre et al., 2006). The IGRA Version 2 used in this paper, released by the NOAA's National Centers for Environmental Information (Durre et al., 2016) and recently described in Durre et al. (2018), has enhanced data coverage and extends back in time as early as 1905, although (for historical reasons) humidity data begin in 1930 with a sole location in Europe. The extension to observations prior to 1946 resulted mainly from the addition of data from the Comprehensive Historical Upper-Air Network (CHUAN), which is the most important collection of upper-air observations taken before 1958 (Stickler et al., 2009). In view of the huge amount of data collected in IGRA (which is a combination of radiosonde and pilot-balloon observations) and the differences in the observing period, temporal regularity and continuity, vertical resolution and vertical extension of humidity data among different stations, finding the most suitable humidity-reporting stations (or humidity soundings from different stations) for a specific purpose can be difficult to put into practice.

Radiosonde humidity measurements involve the simultaneous measurements of pressure, temperature and relative humidity or dew-point depression. Therefore, except for horizontal wind, which is indirectly measured with the aid of a remote tracking device, humidity represents the most accomplished of the radiosonde observations. The purpose of this paper is to study the completeness of humidity observations collected in IGRA according to various needs − number and latitudinal distribution of observing stations, fraction of observing days in a year, resolution and range of vertical levels, length and continuity of the time-series, minimal sampling between the surface and the 500-hPa level − aiming to facilitate the use of radiosonde humidity data by atmospheric and environmental scientists. The task is two-fold: first, to elucidate the completeness of the humidity observations from IGRA for each year in global terms, including the latitudinal coverage of stations and the length of regular time-series; second, to provide metadata describing the completeness of humidity observations from each station. The observing periods without missing years in humidity data must be clarified. Latitudinal and regional differences should be easily derived from the geographic coordinates of stations.

The remainder of this section is intended to clarify the term "completeness of observations" concerning the use of radiosonde data and to present an historical account of the main factors that limit the completeness of humidity observations from radiosondes: vertical levels available in radiosonde reports; missing observations associated with humidity sensor limitations. The next sections are organized as follows. Section 2 indicates the IGRA data set used in the study; selects the IGRA stations reporting a minimum of radiosonde data (coined as 'IGRA-RS') by discharging stations with practically wind-only data in their period of record; and explains the data analysis. Section 3 presents a global picture of the completeness of humidity observations over the years, as derived from the IGRA-RS stations. Section 4 provides the definition of the metadata

parameters describing the completeness of humidity observations from each IGRA-RS station – either as annual statistics or for individual soundings – and the format description of the corresponding data sets. The availability of the resulting dataset supplied by the study (Ferreira et al., 2018) is reported in Sect. 5. A summary of results and some suggestions for future application are given in Sect. 6.

## 1.1 Completeness of observations for radiosonde humidity studies

Data completeness in a data set refers to the extent to which the data set collects the expected elements: quality-assured data are not left-out; missing or invalid values are properly indicated. This is a basic requirement for data quality, and it is assured in IGRA. In a different way, and uncommon, data completeness may refer to *whether the required data for a specific purpose are available or not.* That meaning is not new in the field of meteorology. For instance, the WMO recommendations on "data completeness" (sic) required for calculating monthly means and climate normals from meteorological surface data refer to the temporal continuity and regularity of observations for different climate elements (WMO, 1989). Broadly, Bellamy (1970) discussed the acceptability of meteorological observations in terms of their degree of completeness, considering that the goal of meteorological observations is to "depict the space-time distributions of everything-atmospheric everywhere always, ever more completely in ever-increasing detail"; appropriately, he used the expression *completeness of observations*. This is the terminology used in the present paper.

Concerning the completeness of radiosonde humidity observations, the vertical coverage and vertical resolution of sounding data is of first concern, chiefly between the surface and the middle troposphere (~500 hPa) regarding the precipitable water vapor content; furthermore, the period of record and the regularity and continuity of radiosonde data is a relevant issue for long-term monitoring of the climate system (Karl et al., 1995), as exemplified by temporal sampling requirements used in trend and seasonal analysis of temperature, humidity and integrated water vapor (Gaffen et al., 1991; Gaffen and Elliot, 1992; Karl et al., 1995; Ross and Elliot, 1996; Zhai and Eskridge 1997, Lanzante et al. 2003; McCarthy et al., 2009). Although the vertical and temporal completeness of station-based humidity time-series can be treated separately from the geographical coverage of stations, studying the completeness of observations in a global, historical data set of radiosonde observations should address both issues simultaneously. This is particularly true concerning the subsampling of radiosonde stations for studies of atmospheric temperature or water vapor trends on a regional or global scale (Wallis, 1997).

Several factors contribute to differences in the completeness of humidity observations among radiosonde stations and individual soundings: i) The geographical coverage of radiosonde stations evolved over time, and so, the period of usage varies among stations; ii) A lack of equipment maintenance may result in interruptions of observations; iii) The number of vertical levels and vertical extent in radiosonde reports depend on the standard pressure levels in use, as well as on the reported significant levels (assuming that the balloon bursts at the proper altitude); iv) Missing humidity observations arise from difficulties associated with the performance of humidity sensors and the observing practices related to their working range.

While (i) and (ii) are of a random nature, points (iii) and (iv) deserve an explanation because of historical changes with implications in the vertical coverage and resolution of radiosonde humidity profiles.

## 1.2 Vertical levels in radiosonde observations

In radiosonde soundings, temperature, relative humidity (and/or dewpoint depression) and wind speed and direction are measured together with atmospheric pressure, while geopotential height is indirectly measured from hypsometric calculations[1] (but may be missing in radiosonde reports). As a common practice, only standard pressure levels and significant levels are stored and reported. Currently, the *standard levels* are 1000, 925, 850, 700, 500, 400, 300, 250, 200, 150, 100, 70, 50, 30, 20, and 10 hPa (WMO, 1996). But historical changes deserve due attention. An inspection of the earliest soundings collected in IGRA – made in 1905 at Lindenberg, Germany, a quarter of a century before radiosondes were available – reveals temperature data reaching sometimes 100 hPa, with the reported levels being 1000, 925, 850, 700, 600, 500, 400, 300, 250, 200, 150 and 100 hPa (although most of those soundings did not reach beyond 700 hPa). Radiosonde humidity measurements at the same station, as collected in IGRA, began in 1950. Nevertheless, the 150- and 100-hPa levels were first recommended by the WMO in 1953, while the levels 70, 50, 30, 20, and 10 hPa were proposed in 1957, the International Geophysical Year. Even so, the levels above 200 hPa were still referred as non-standard by the WMO in 1958, until the 100-hPa level was finally adopted that year (WMO, 1957; WMO 1958). In the years that followed, the pressure levels ≤ 150 hPa (representing roughly the stratosphere) became common worldwide. As to the lower levels ≥ 200 hPa (representing roughly the troposphere), they were in general use since the early 1940s, with two exceptions: first, the 250-hPa level was only adopted in 1970, to satisfy aviation demands (WMO, 1970); second, the 925-hPa (within the planetary-boundary layer above low-altitude stations), although planned since 1977, was first required in WMO Antarctic stations in 1987, given the low surface pressure over the Antarctic Plateau, until it was adopted worldwide by the end of 1991 (WMO, 1977, p. 15; WMO, 1987, pp. 57-58; Oakley, 1993, p. 23). Note, however, that these two levels were in use to in some stations before international agreement, and for a long time as exemplified by the Lindenberg station. Besides the standard levels, some intermediate fixed levels within the troposphere (e.g., 800, 750, 650, 600 hPa) are regularly used in some stations (Shea et al., 1994). The Lindenberg station also indicates the early use of 600 hPa. The additional (high-stratospheric) levels 7, 5, 3, 2 and 1 hPa have been used in agreement with WMO recommendations (WMO, 1970), depending on regional and national practices. E.g., they form part of 'upper-level' observations in the U.S. National Weather Service (OFCM, 1997).

The number of *significant levels* – non-standard levels needed to reproduce the vertical temperature and dew-point temperature profiles, capturing turning points or abrupt changes (such as thermal inversions and the tropopause) – depend on atmospheric conditions, manual rules and, before automation, on the observers' skills. By the late 1950s, the rules for choosing significative levels were still under discussion (WMO, 1957), being established over time by WMO regulations (WMO, 1988).

---

[1] Except in some Soviet/Russian radiosonde-radar systems and the last generation of GPS radiosondes – in which pressure is deduced from the (radar or GPS, respectively) profile of geometric height and the radiosonde profiles of temperature and humidity (Zaitseva, 1993; Nash et al., 2011).

Interestingly, the almost linear increase in the average number of non-standard levels in weather-balloon sounding reports (radiosondes + pilot balloons) from about zero by 1945 to about 30 by 2000 – as revealed from IGRA v1, inferring from Fig. 7 in Durre et al. (2006) –, can hardly be attributed to an increased attention to significant levels alone. It suggests that a significant number of stations have reported additional levels apart from the standard and significant levels (both "mandatory" in WMO's nomenclature).

The *surface level*, which is treated separately in upper-air sounding reports, was reported in most of the radiosonde stations since the mid-1940s. However, it has been reported systematically only since around 2000 (as shown later in Sect. 2.3).

The current migration of radiosonde reports from alphanumeric (TEMP) to the binary universal form for the representation of meteorological data (BUFR), together with the conversion of radiosondes to generate native BUFR messages, allows the transmission of high-resolution data (2 to 10 s sampling rate, i.e., ~ 5 to 50 m resolution in a typical balloon ascent) along with the balloon drift position, the observation time for each level and other metadata (Ingleby et al., 2016). Currently, 20% of the radiosonde stations send high-resolution BUFR reports through the Global Telecommunication System (GTS), many coming from Europe; however, such data are not yet available in an open archive.

## 1.3 Missing humidity observations

Combining adequate spatial and temporal resolution with enough accuracy for synoptic use, modern radiosonde measurements reach the upper troposphere and lower stratosphere, much beyond the layers where most of the atmospheric water vapor resides. That has not always been so. While the vertical sampling of temperature soundings is limited by the burst altitude and the mandatory levels (standard and significant), the maximum height and the vertical resolution of humidity soundings are further restricted by sensor limitations. Upper-air humidity measurements began in the 1930s but became substantial only in the 1940s. Despite radiosonde hygrometers (measuring relative humidity (RH)) have improved over time, humidity has been always difficult to measure in very cold or dry air due to the poor response of many instruments at very small vapor concentrations (by lowering saturation vapor pressure, cold temperatures are associated with low water vapor pressures). As it was once pointed out, "humidity measurements in the free atmosphere are probably the least satisfactory of the regular aerological observations" (Hawson, 1970). Balloon-based chilled mirror hygrometers, designed to measure water-vapor mixing ratios in the stratosphere (an extremely cold and dry environment), has been used for more than half a century but are exclusive to scientific research or comparison with humidity measurements from operational radiosondes (Mastenbrook and Daniels, 1980; Vömel et al. 2007; Hurst et al., 2011; Hall et al., 2016). Since a long time ago, whether services need to rely on meteorological radiosondes consisting of expandable balloons carrying relatively low-cost and light instrument packages (Brettle and Galvin, 2003).

Here is a brief review of the main humidity sensor types and their limitations, since the time when registering balloons were abandoned by national weather services and electric hygrometers began to be incorporated in radiosondes (circa 1940;

DuBois (2002)). The lithium chloride humidity sensors, which were widely used in radiosondes between the mid-1940s and the mid1960s, did not respond to temperatures below around −40º C. From the early 1960s onwards, the new carbon hygristor allowed measurements at lower temperatures − down to −65°C in the early 1990s, however with a time lag in the sensor´s response as large as 10 minutes (Garand et al., 1992). In practice, humidity measurements at temperatures below −40°C were

discontinued in many countries before the 1990s, limiting the vertical extent of routine humidity observations to about 400 hPa (≈ 7 km altitude) (Gutnick, 1962; Gaffen, 1993). Besides, the radiosondes using lithium chloride hygrometers suffered from a low-frequency limitation in the transmission of RH less than 15–20 %, known as motorboating (Wade, 1994). The radiosondes using the carbon hygristor enabled, in principle, measurements in that low RH region − however, the accuracy and reproducibility of low-RH values was little known and suspected to be poor for many years, giving the wrong impression

that relative humidity lower than about 20 % did not occur in the lower troposphere (Wade, 1994; Nash, 2015). Therefore, values of RH below 20 % were usually cut off in humidity reports; in the radiosonde network of the U.S.A. this happened between 1973 and 1992 (Elliott and Gaffen, 1991). Note that changes in instrument and reporting practices in different countries took place at different times: the threshold value of RH varied in the range 10–20%; the lowest temperature of $−40^{0}C$ for reporting humidity, and the shift to lower temperatures, was applied in different periods depending on country; humidity

could be reported up to a specified pressure level. Moreover, mechanical sensors are not exclusive to pre-1940s radiosondes: hair hygrometers were only abandoned in the mid-1950s and rolled hair hygrometers were used in a few places until about 1980; the goldbeater's skin sensors introduced in the 1950s became particularly important in the Soviet Union. [For historical details on these changes, see Gaffen (1993).] The new capacitive thin-film sensors introduced in 1981, with the RS80 radiosonde, have improved the response time at low temperatures and the capability of measuring very low humidity. Two

important enhancements occurred in the late 1990s. First, the protection from chemical contamination arising from outgassing of RS80 radiosonde packages, thus making the dielectric polymer more selective to water vapor molecules and reducing dry bias (Wang et al., 2002). Second, the dual sensors introduced in the RS90 radiosonde, in which two sensors were alternately heated to remove condensation from the measuring sensor, thus preventing wet biases after measurements in saturated conditions. In the RS92 radiosonde, in use since 2004, the lowest temperature of the heating cycle extended down from − 40

$^{0}$C to − 60 $^{0}$C. The smaller size of and the better ventilation of the RS90 and RS92 sensors compared to RS80 improved the response time. However, RS80 sondes were less affected by dry biases in daytime measurements because of the protective rain cap which also prevented direct sunlight (Smit et al., 2013). However, RH reports at temperatures lower than −40 °C did not develop significantly until about 2000; in recent years, for temperatures of −50 °C to −70 ° C only the newest humidity sensors respond quickly enough to make useful measurements; moreover, the best ones had an uncertainty of around 16 % RH

at temperatures as low −70 ºC (occurring over Antarctica and around the tropical tropopause), which is barely acceptable for numerical weather prevision but not suitable for climatic studies (Nash, 2015). Improvements over time were not restricted to sensor type but also to data reduction and calibration. E.g.: measurements from the carbon hygristor in VIZ radiosondes were improved in the 1990s by correcting the low-humidity algorithm; some modern radiosonde systems apply corrections for slow

time constant of response and for daytime heating of the humidity sensor; calibration at low temperatures was perfectioned (Dirksen et al., (2014) and references therein). While radiosonde humidity measurements are now generally reliable in the troposphere, uncertainties remain concerning the upper-stratosphere, with temperatures below −50ºC, in addition to dry conditions found above the lower troposphere and wet conditions that occur in thick clouds (Miloshevich et al., 2006). Although the capacitive thin-film sensors have been widespread (with Vaisala radiosondes RS80 and RS92), two older sensor types continued in use for many years: the carbon hygristor (in VIZ/Sippican radiosondes, currently in disuse, and in the GTS1 radiosonde, in use in China) and the goldbeater's skin sensor used in some radiosonde types made in Russia and China until a few years ago; this peculiar sensor responded too slowly to be useful at temperatures lower than −20 °C and suffered from hysteresis following exposure to low humidity (Nash, 2015; Moradi et al., 2013). For the current radiosonde types, see Ingleby (2017).

The trouble in measuring upper-air humidity affects the completeness of observations in several ways: the vertical extent of humidity soundings varies much among radiosonde stations and over time owing to sensor limitations in very cold air; likewise, vertical gaps in low humidity regions are expected, due to cut-off of RH below sensors' measuring capability; lastly, missing days in radiosonde humidity records may originate from adverse conditions (dry days, wet days, cold days) at individual stations (Garand et al., 1992; Ross and Elliott, 1996; McCarthy et al., 2009; Dai et al., 2011). As explained above, the actual extent of missing data depends on the observing practices combined with sensor limitations. In addition, failures in some part of the radiosonde system can compromise soundings. Faulty ground-equipment used for control checks (sensors' calibration before balloon release), data reduction and data recording or telecommunication of coded reports may cause long inoperative periods; poor signal reception from the radiosonde make sometimes data processing impossible. Radiosonde operations in remote environments, particularly performed from ships, present their own challenges; Hartten et al. (2018) give a vivid illustration. In sum, the vertical extent, vertical resolution, temporal regularity and continuity of humidity reports are quite heterogeneous.

## 2 Input data and methods

We have examined the IGRA 2 main dataset until the end of 2016. Section 2.1 presents briefly that dataset, including the quality assurance of humidity data. Section 2.2 provides a first look on the data, to find out how many and which of the IGRA stations have a non-negligible amount of radiosonde observations (RAOB), and at the same time to give a hint of the amount of humidity data. Section 2.3 describes the data analysis, aiming to explore the completeness of humidity observations in the sense introduced in Sect. 1.

## 2.1 IGRA 2 – sounding data

The IGRA 2 consists primarily of radiosonde[2] and pilot-balloon[3] observations from over 2700 globally and temporally distributed stations, even though the coverage over oceans is limited to ships, buoys and remote islands. This paper concerns with sounding data (Durre et al. 2016), comprising over 45 million soundings from 2761 (2662 fixed and 99 mobile) stations [based on data accessed in September 2017].

The main difference between IGRA 2 and IGRA 1 is the amount of sounding data: 33 data sources instead of the initial 11, implying about 80 % more stations; new data from hundreds of stations before 1946 and the addition of floating stations (fixed weather ships and buoys, mobile ships, and Russian ice islands); furthermore, humidity data prior to 1969 were added. The latter change is related to how humidity values were stored in radiosonde reports. Until 1969, humidity observations were given only as RH; from then on, RH measurements have normally been converted to dewpoint depression (DPD) and reported mostly in that form. Different assumptions in the conversion code can lead to inconsistencies of data (Garand el al., 1992). The former IGRA contained only DPD, while IGRA 2 contains humidity data in either form, as available in original reports, provided they pass the following conditions:

 i) Data completeness: valid temperature accompanies humidity data;

ii) Valid range: 0-100 % for RH; 0 to 70 ℃ for DPD;

iii) Internal consistency: DPD-derived RH differs from reported RH by 10 % at most;

iv) Plausibility: (derived water-vapor pressure) $\leq 0.1 \times$ (atmospheric pressure).

Quality checks (i)–(ii), save for the RH range, are integral to IGRA from its creation (Durre et al., 2006); (ii) for RH (with the later introduction of this variable in the archive) and (iii)–(iv) were added in IGRA 2 (Durre, 2016; Durre et al., 2018). Note that a RAOB message must have at least temperature data at several pressure levels, while humidity or wind data may be missing, and geopotential height is not always given. The recording of pressure levels, and consistence between pressure and geopotential height whenever the latter is reported in source data, has been assured in IGRA since its first version. IGRA uses a consistent data format, irrespective of the provenience of the data (PIBAL or RAOB). Therefore, RAOBs in IGRA can be simply identified by the presence of temperature data. Wind observations from pilot-balloons (PIBAL) have only wind data at several geopotential heights (adjusted from geometrical height measurements and the gravitational field).. Concerning humidity data, the precision and accuracy of RH and DPD data vary substantially as a function of RH and temperature, degrading in dry or cold conditions to a greater or lesser extent depending on the radiosonde type (for a review on the subject,

---

[2] In modern usage, the term radiosonde refers not only to the early radiosondes but also to the rawindsondes (in use since the 1950s), which, besides measuring thermodynamic parameters, provide wind information with the aid of a radio-theodolite, a radar device, a radio navigation system or, more recently, GPS (Dabberdt et al. 2002; Nash et al., 2007). Observations from either radiosonde type are often abbreviated as 'raob' in meteorological jargon.

[3] Free balloon tracked by optical theodolites or radar to measure upper-air winds (Wenstrom, 1937, Hickman, 2015). Often abbreviated as 'pibal'. The common single theodolite technique requires the approximate ascent rate to obtain position, while the double-theodolite method allows a pure trigonometric calculation. In visual tracking, rarely used today but still important where radar tracking or wind measurements from a rawindsonde are not possible, a flashlight is used during night or twilight hours.

see Smit et al. (2013)). The information about instrument changes (stations' history), whenever available, is provided in a separate metadata file in IGRA 2 [update of the metadata given in the first version of IGRA, which were mostly taken from Gaffen (1996)].

The most frequent nominal observation times are 0300 and 1500 UT until 1957 and 0000 and 1200 UT afterwards, which reflects the shift in observing time that occurred in 1957 in major WMO radiosonde networks. In the beginning of 1958 the primary standard hours of WMO upper-air observations were already 0000 and 1200 GMT (WMO, 1958). However, in some countries, different synoptic hours were practiced over the years; sometimes stations have performed up to four soundings per day for certain demands (see Gaffen, 1993).

For a description of data coverage and data sources of IGRA 2, a full description of quality assurance of data, and further detail on the differences between IGRA 1 and IGRA 2, the reader should see Durre et al. (2018) published after the initial draft of this paper was written. NB – Hereafter, IGRA 2 is simply referred to as IGRA, unless stated otherwise.

## 2.2 Identification of radiosonde stations

The examination of the IGRA reveals that 958 stations have wind-only observations in their full period of record, i.e., 34.7 % of the stations represented in the entire archive. These stations form part of the global pilot-balloon (PIBAL) station network, which evolved over time. As to the rest of the stations, some of them changed from a PIBAL launching site to a radiosonde launching site at some point in their period of record, – meaning that they are not, strictly, PIBAL stations nor radiosonde stations. Obviously, the number of PIBAL stations or radiosonde stations at a certain time depends on the stations opened and closed before that time, of either type. In the following, terms like "observations", "soundings" or "reports" refer to the upper-air stations and balloon data of IGRA, which retains most of the source data.

Figure 1a shows the yearly number of stations reporting RAOB any time of the year – meaning they have at least observations of temperature, regardless of simultaneous humidity/wind observations – and of stations reporting PIBAL observations alone – i.e., reporting only wind throughout the year. For comparison, the number of stations reporting any humidity data – indicating that radiosondes are equipped with a hygrometer – and with humidity observations in more than 95 % of the radio soundings, is also shown. Constituting the bulk of the IGRA stations until the early 1940s, the stations with only PIBAL data represent nowadays only 13 % of the total. The reason for the apparent discontinuity in the amount of stations performing only PIBALobservations between 1972 and 1973 is this: beginning in 1973, IGRA data largely come from the GTS and include many more PIBAL data than prior data sources (Imke Durre[4], personal communication, April 12, 2018). The number of stations reporting RAOB increased rapidly since the mid-1940s, staying in the range 800–900 from around 1970 to present (2016). Note that, before the advent of the radiosonde, upper-air measurements of temperature and relative humidity were made using kites, registering balloons and aircrafts; these platforms were gradually abandoned until the

---

[4] Center for Weather and Climate, NOAA's National Centers for Environmental Information, Ashville, NC.

radiosonde era was established in the mid-1940s (DuBois, 2002; Stickler et al., 2010). Since the first radiosonde prototypes were developed between 1929 and 1930, the early observations of temperature collected in IGRA, beginning with one station in 1905 (Lindenberg, Germany), were made by meteographs without radio-telemetry. The first upper-air humidity observations are from 1930, coming from a single station: Kjeller, Norway. According to IGRA, until 1942 there were less than 10 stations reporting humidity, with that number growing rapidly in the following years together with the total number of radiosonde stations, reflecting the widespread use of radiosondes (see 'HUM' and 'TEMP' in Fig. 1a). The major relative change occurred between 1945 and 1946, coincident with the end of World War II, when the global count of radiosonde stations tripled. Note that the replacement of hair hygrometers by the lithium chloride humidity element began in some radiosonde networks shortly before; in the U.S.A., apparently that change took place between 1940 and 1943 (Elliott and Gaffen, 1991; DuBois, 2002), although IGRA does not contain data prior to 1946. Fig 1a shows also that the fraction of radiosonde stations measuring humidity in more than 95 % of the soundings increased over the years, getting very close to 100% in the last decade (see relative difference between the curves 'TEMP' and 'HUM > 95 %').

Figure 1b shows the evolution of the global, annual mean number of soundings performed per day, for the different atmospheric parameters apart from pressure: temperature, humidity and wind. Recall that pressure is always measured in RAOB soundings, while in PIBAL soundings wind is measured as a function of altitude[5]. For clarity, the PIBAL wind soundings are depicted separately from all wind measurements, which also comes from RAOB soundings since the mid-1940s. Although, as a rule, radiosonde launches are carried out twice a day, in fact there is a significant number of missing days in temperature and humidity data, i.e. days without any RAOB data: roughly 1 in 5 days during the year, on average for the years after the mid-1950s [as concluded by comparing the yearly number of observing stations (TEMP, HUM in Fig. 1a) with half the global number of daily observations (TEMP, HUM in Fig. 1b)].

Aiming to study humidity completeness, the IGRA stations having a negligible amount of temperature data in every year of their period of record were excluded, because temperature is required to measure RH or DPD and so all humidity data in IGRA are accompanied by temperature data. Specifically, we have selected the *stations with RAOB soundings in 5 % or more of the annual soundings in at least one calendar year within their full period of record* until the end of 2016. These will be hereafter referred to as *IGRA-RS stations* (RS stands for radiosonde), even if some of them contribute with relatively few RAOB data. The above  selection reduces the number of IGRA stations by 38 %, whereas the number of soundings is only reduced by 13 %, amounting to 39.5 million, out of which there are 30.2 million radio soundings including 29.8 million humidity soundings (see Table 1). Wind-only soundings are still present in 23.6 % of the soundings from the selected stations. Note, however, that 92.2 % of the removed stations are strictly PIBAL stations and the remaining 7.8 % have RAOB data in less than 0.6 % of the corresponding soundings, apart from two cases with a period of record shorter than one year. In sum,

---

[5] Pressure levels are present in the wind-only data coming from 28 IGRA stations (standard levels with missing height), indicating that an on-board pressure sensor with a radio transmitter was used along with the wind-finding system.

the IGRA-RS subset retains practically all the RAOB soundings (99.999%), particularly the humidity soundings, as shown in Table 1.

The IGRA-RS stations and their locations are listed in supplementary Table S1, along with the full periods of record (Full POR), the periods of record for humidity (Hum POR) and the corresponding numbers of humidity observations (Hum Obs), i.e., the number of individual soundings reporting either DPD or RH data. Since humidity time series can be interrupted for long periods of time, the full period of record of one station may be segmented into two or more periods for humidity (both are rounded to years). Table S1 comprises 1723 stations, of which 1300 are WMO stations (denoted by the letter 'M' following the 2-character country code of IGRA identifier codes). Note: in the data from around 120 land stations the early years of record for humidity (normally 2 to 3 years), contains only surface or near-surface data; this happens in about 100 stations of the former Soviet Union, mostly during the years 1946-49.

Focusing on the usefulness for climatic studies, the subset of WMO upper-air stations integrated into the Global Climate Observing System (GCOS), i.e., the GCOS Upper-Air Network (GUAN), deserves attention. Formally established in the 1990s, the GUAN is aimed to provide long-term, consistent, homogeneous and reliable observations needed to monitor the atmospheric component of the global climate system (WMO, 2002; McCarthy, 2008). At present the GUAN comprises 178 stations, all of which are represented in IGRA-RS. The IGRA-derived statistics of humidity observations from the GUAN stations for the period 2001/10/01 to 2016/12/31 is shown in supplementary Table S2, as explained next. 'Hum POR' indicates the years with any humidity data in the year, as found in IGRA, beginning at the time when each station was included in GUAN, or, at least, at the earliest time for which performance indicators for the GUAN stations are available through the NOAA/National Centers for Environmental Information website; this is the first day of the month of 'Begin Date' indicated in Table S2. '# Days' is the number of days in Hum POR, excluding the months before Begin Date. The last three columns give the corresponding count of humidity observations around the principal nominal hours, 0000 UT and 1200 UT (± 1 h), and at any other times (0200 UT through 1000 UT and 1400 UT through 2200 UT). Stations are identified by the WMO region and WMO number, followed by the station name and country. (To find out the corresponding IGRA ID codes in Table S1 it suffices to observe that the last nine characters must be 'M000' followed by the WMO number.) Note that most of the GUAN stations have humidity data at or around 0000 UT and 1200 UT almost every day; however, the exceptions to the rule, and even gap years, are not negligible.

Moreover, the IGRA-RS contains 16 stations that form part of the GCOS Reference Upper-Air Network (GRUAN; Bodeker et al., 2016): half certified and half to be certified according to current GRUAN status, of which eight (half certified too) are also GUAN stations. Those specific GRUAN sites report default data (from radiosonde manufacturers) to the GTS; at present, most of them already send BUFR messages with high resolution (Michael Sommer[6], personal communication, September 18, 2018). The GRUAN aims to serve as reference network for climate applications, satellite validation and in

---

[6] GRUAN Lead Centre, Lindenberg Meteorological Observatory - Richard Aßmann Observatory, Germany.

support of other radiosonde networks, by providing long-term high-quality records of vertical profiles of selected essential climate variables, accompanied by traceable estimates of measurement uncertainties (WMO, 2011a; Dirksen et al., 2014). Naturally, real-time meteorological data transmitted from GRUAN sites to the GTS may differ from GRUAN internal data regarding raw data processing. For reference, the IGRA ID codes of the GRUAN sites appearing in the IGRA-RS station list are underlined on Table S1; of course, other GRUAN sites performing only research measurements are not part of IGRA. Likewise, the WMO numbers of the GUAN stations coincident with GRUAN sites are underlined on Table S2.

## 2.3 Analysis of humidity data

Overall, the analysis of data from IGRA-RS stations, selected as described in the previous section, aims to answer the following questions:

1. What is the spatial coverage of humidity-reporting stations in different years and latitudes?
2. What is the fraction of days in a year with humidity data and the number of consecutive missing days on average?
3. What is the typical vertical resolution and vertical extent of humidity observations?
4. How many stations have enough data in the vertical to allow the estimation of precipitable water?
5. How does the temporal and vertical completeness affect the availability of long-term humidity time-series?

Each question is explored as detailed below in Sects. 2.3.1–2.3.5, with the results presented later in Sect. 3. The description of the related metadata parameters regarding each IGRA-RS station, is deferred to Sect. 4.

## 2.3.1 Global coverage

The geographical distribution of the radiosonde network measuring humidity evolved over time. Its size and distribution were studied in terms of the annual number of stations with any humidity observations during the year in different climatic zones, considering only the fixed IGRA-RS stations. The spatial coverage of observations was further studied by a parameter that is closely related to the average spacing of stations, but it represents better the data coverage if stations are unevenly distributed.

The average separation between adjacent stations ($L$) over a region of the Earth's surface can be estimated by $\sqrt{A/n}$, where $n$ is the number of stations lying on a surface of area $A$, $n/A$ representing the average station density. This measure is, however, insensible to the spatial distribution of stations. The global radiosonde network has highly variable density since the observation stations are concentrated in continental regions, mostly in populated areas of developed countries. Sparse-data areas occur on oceans and seas, near the poles and in certain parts of land continents. $L$ can be alternatively defined as the mean distance between each station and its nearest neighbor; but this definition ignores data-void areas. The average distance from a point on the surface to the nearest station ($\approx L/2$ for a uniform network) is more informative because it depends on the distribution of concentrated- and sparse-data areas. Therefore, to study the global coverage of observations it is convenient to use the *average distance to the nearest station, as measured from every point over the main landmasses or ocean/sea areas within a given latitude band*. Let $s(x)$ be the geodesic distance from a given point $x = (\varphi, \lambda)$ of latitude $\varphi$ and longitude $\lambda$ to

the position of the nearest station: $s = \min\{dist(x, x_i); \; i = 1, 2, \ldots, N\}$, where $x_i$ denotes the positions of individual stations, say $N$ in total. Averaging $s$ over a zonal band bounded by latitudes $\varphi_1$ and $\varphi_2$, under the spherical-Earth approximation,

$$\bar{s}(\varphi_1, \varphi_2) = \int_0^{2\pi} \int_{\varphi_1}^{\varphi_2} \sigma \, s \cos \varphi \, d\varphi \, d\lambda \Big/ \int_0^{2\pi} \int_{\varphi_1}^{\varphi_2} \sigma \cos \varphi \, d\varphi \, d\lambda \qquad (1)$$

where the overbar denotes area-weighted average, and $\sigma(\varphi, \lambda)$ is a mask value that can be used to restrict the calculation to mostly land or water regions by switching the values $\sigma = 0$ and $\sigma = 1$ appropriately. The following method was applied. First, the calculation for main landmasses excludes points on landmasses smaller than Ireland, since they give irrelevant information about the spacing of stations over land; however, continental archipelagos are treated as part of continents. Any regions outside

the above defined main landmasses are treated as belonging to ocean/sea, excluding lakes which are included in continents. Finally, the determination of the nearest station from points on ocean/sea areas involves not only stations surrounded by sea water (stations on oceanic islands plus a few fixed weather-ships, since we focus on fixed stations) but also stations located on the coastline of continents and large islands, as well as on the shores of seas enclosed by continents. This scheme assumes that upper-air observations at such locations are partly representative of atmospheric conditions above the nearby waters, because

the physical frontier between land and sea is blurred in the atmosphere (incidentally, the island and coastal *surface* stations are classified by the WMO as 'sea stations').

We have applied Eq. (1) to the IGRA-RS fixed stations reporting humidity in specific years to examine the global coverage of upper-air humidity observations in different climatic zones over time, regardless of the temporal and vertical completeness of time series. Such information is not part of the dataset introduced in this paper, which focus precisely on the

time series at each station. Nevertheless, Eq. (1), with possible adaptions for the latitude and longitude intervals, may be used to study the spatial coverage of any subset of stations selected according to a given range for the metadata parameters presented in Sect. 4.

### 2.3.2 Annual frequency and temporal continuity

The frequency of humidity observations over time is studied in terms of the *fraction of humidity observing days in the year.*

Although  this gives a sense of the regularity of observations, it says little about the continuity of data over the year. In this respect, it is of interest to know the size of the *maximum interval of consecutive days without humidity data in a year* – denoted hereafter as *'size of missing days'*.

The above defined measures of temporal completeness are critical to study climatic trends (long-term changes in the annual mean or in the seasonal cycle of humidity-related quantities) on specific locations or areas of the globe, which otherwise

requires merging procedures using radiosonde data from nearby locations to circumvent large data gaps. We have averaged both quantities across all fixed stations reporting humidity within each major latitude region, year by year.

### 2.3.3 Vertical resolution and vertical extent

Since the vertical resolution varies with height – according to the height of the reported pressure levels (standard and significant) and depending on the number of levels with non-missing data for humidity –, the vertical resolution of an individual sounding must be defined by a vertical average. Since the vertical distance between consecutive levels, say $dz_k$, generally increases with height, with the lower levels being more populated than the upper layers, a geometric mean is more suitable than an arithmetic mean. So, the *mean vertical resolution of a single humidity sounding* is here defined by the geometric mean of $\{dz_k\}$ for all levels with humidity data in the sounding profile:

$$mean\ vertical\ resolution = \frac{R_d}{g_0} \prod_{K=1}^{M} \left( \bar{T}_k ln \frac{p_{k-1}}{p_k} \right)^{1/M} \tag{2}$$

where $p_k$ is the atmospheric pressure at level $k$ ($k = 0$ denoting the lowest level with humidity data), $M$ is the number of levels with humidity data above the lowest level, $\bar{T}_k$ is the estimated mean temperature between level $k$ and its immediate, relevant lower level $k - 1$, $R_d$ is the specific gas constant for dry air and $g_0$ is the standard gravity. (Note: IGRA's data-quality checks assures that vertical levels with valid humidity data have also valid temperature and pressure data.)

Since geopotential altitude is only given in part of the RAOB reports, the *vertical extent of an individual humidity sounding*, i.e. its *maximum height above mean sea level reached by the humidity measurements,* was estimated by adding the station's elevation to the height from the surface calculated upon pressure and temperature data from the surface level up to the top of the humidity sounding (highest level with a non-missing value for DPD or RH), whenever values of temperature and pressure at the surface are given; otherwise the height from the surface cannot be calculated. For mobile stations (ships and buoys), the elevation of the stations can be approximated to zero, unless the vertical extent of the sounding is too small, requiring data for the balloon release height.. For the purpose, it suffices to neglect moisture in the hypsometric equation; given that the virtual temperature is typically within 4 K above the actual temperature, the error in calculating geopotential height amounts to less than 1 %.

Following the above definitions, we have studied the statistical distributions of the vertical extent and the vertical resolution in humidity soundings from all IGRA-RS stations (including mobile) over time by grouping individual values of both parameters in annual bins. To assess the shortness of humidity observations in RAOB, we have also calculated the vertical extent of temperature observations and their vertical resolution up to the top of the co-located humidity observations.

### 2.3.4 Soundings eligible to estimate precipitable water vapor

Usually, the precipitable water vapor (column integrated water vapor mass per unit surface area) is estimated from the profile of water vapor mixing ratio between the surface to the 500-hPa level – i.e. the layer where ~ 95 % of the columnar mass of water vapor is and where humidity data from radiosondes are more often available and generally more accurate (Elliot et al.,

1991; Gaffen et al., 1992; Ross and Elliott, 1996; Durre et al., 2009). In this paper, a humidity profile is considered eligible to estimate precipitable water vapor under the following conditions:

  i.   *Humidity data are given at the station's surface and at all standard levels laying between the surface and the 500-hPa level, except for the 925-hPa level.*

ii.  *If humidity data is missing at a standard level apart from 925-hPa, a nearby significant level is acceptable if its height from the surface differs from the height of the missing standard level by less than 5 %.*

  iii. *The distance between any consecutive levels with humidity data between the surface and the lowest level located more than 1 km away from the surface should not exceed 1 km, unless the station elevation is larger than 500 m.*

The 925-hPa level is not required here because this level was not standard until 1991. However, condition (iii) assures a
minimal resolution in the planetary boundary layer, by including near-surface significant levels as well as the 925-hPa level when it is given; this is required because water vapor is highly variable and abundant in this region; the exception for very elevated stations contemplates the case when the first upper-air humidity record is at 850-hPa, i.e. $\approx$ 1.5 km above the mean sea level, but the height from the surface is less than 1 km. The IGRA-RS soundings fulfilling the above conditions will be hereafter referred to as *Sfc-to-500hPa humidity soundings*.

15       Typically, the first standard level higher than 1 km from the ground is 850-hPa. Thus, by including enough data at significant levels below the 850-hPa level, instead of requiring data at the current standard levels in the same layer – sorted out of 925-hPa and 1000-hPa, depending on the surface pressure – the definition given above accommodates much more soundings, particularly before 1992 when the 925-hPa level was not mandatory. To be sure, the relative amount of humidity data at near-surface levels is now examined, excluding 1000-hPa (around 0.1 km altitude) since this is frequently placed below
the stations' elevation (339 m on average for the IGRA-RS fixed stations).

       Figure 2 shows the evolution of the global percentage of humidity observations at the 925-hPa level and at any significant level between the surface and 850-hPa, out of the IGRA-RS soundings having humidity data at the surface on condition that the surface level pressure is higher than 925 hPa and 850 hPa, respectively (referred in the following as surface-upper-air soundings). The percentage of surface observations in all humidity soundings is also shown. First, note that the
surface observations began in 1943 (in fact, not only of humidity but also of temperature), rising rapidly in the next 5 years to $\approx$ 95 % of the humidity soundings, decreasing then to a minimum of 60% in 1965 and broadly increasing again until 2000, staying above 95 % since then. In short, the humidity measurements at the surface level are mostly available since 1945 but were in widespread use only since 2000. Secondly, the percentage of the surface-upper-air soundings having humidity data at the 925-Pa increased almost as a step-function around 1992. It increases from only 2 % in 1991 to 60, 86 and 98 % in the
following three years. This change is coincident with the introduction of the 925-hPa level as an additional standard level in radiosonde messages in November 1991, and consistent with the fraction of stations already reporting that level in mid-1993 (Oakley, 1993, p. 24). Lastly, the percentage of the surface-upper-air soundings that have humidity data at any significant level

below the 850-hPa level, beginning in 1948, generally increased with time, mainly in the 1960s, with a value larger than 80 % in recent years.

Using the definition given at the beginning of this section, we have studied the number and the percentage of stations (fixed and mobile) whose Sfc-to-500hPa humidity soundings exceed a given percentage out of the humidity soundings made in each year. The distance between missing standard levels and nearby significant levels was calculated from pressure and temperature data, neglecting moisture. A stricter definition of Sfc-to-500hPa humidity soundings, specifically, having humidity data at the surface and all upper-air, current standard levels up to 500-hPa was also studied for comparison.

### 2.3.5 Current record length of time-series

The 'current record length' of a humidity time-series in a given station and year, is herein defined as the *number of elapsed years in the time-series with no gap years in the interim*. To simplify, a calendar year with any amount of humidity data is counted as one. For past years, the current record length generally differs from the span of the entire time-series, which may continue after the year under consideration. Also, note that one station can have more than one time-series for humidity, as the humidity observations can be interrupted by one or more gap years. So, the full 'period of record' of a radiosonde station may be divided into several sub-periods of record for humidity. Recall that the years with humidity observations at each IGRA-RS station are indicated in Table S1.

We have studied the evolution of the average current record length of the humidity time-series, from all stations, year by year. The same was done for the time-series with humidity soundings 90 % of the days in the year, at least, and particularly consisting of Sfc-to-500hPa soundings, with their own record lengths. The calculation of the distribution of the current record length of the time-series with Sfc-to-500hPa soundings 90 % of the days in the year or more was repeated by restricting the corresponding size of missing days to less than 10 days.

### 3 Overview on the completeness of radiosonde humidity observations

This section gives a general picture of the completeness of humidity observations over the years, using the data from the IGRA-RS stations defined in Sect. 2.2 and following the data analysis described in Sect. 2.3. The results of Sects. 3.1 and 3.2 refer to fixed stations, i.e., over 1600 stations on continents and islands, 14 ocean fixed weather-ships and 2 environmental buoys. In the remaining Sects. 3.3–3.5 mobile stations (99 ships) are equally included (see Table S1; moving stations are denoted by unspecified geographical coordinates).

### 3.1 Geographical coverage of humidity observations

Figure 3 shows the geographical distribution of the IGRA-RS fixed stations at specific years illustrating the growth of the global radiosonde network (cf. Fig. 1a); stations reporting humidity observations are highlighted and counted. Recall that a single station can change from PIBAL observations to RAOB, or even the reverse, during its period of activity. The IGRA-RS retains practically all RAOB data of IGRA; however, some stations have years with only PIBAL observations. Since almost all of the IGRA stations measuring temperature in a given year do also measure humidity at least part of the time (as seen by comparing black and solid blue lines in Fig. 1a), it is clear that most of the IGRA-RS station-years without humidity data (red crosses in Fig. 3) correspond to years of PIBAL observations alone (no RAOB). Concerning humidity observations, Fig. 3 shows that, by 1945, 2/3 of the stations were set in South Asia (British India) and Australia. Most of the data coverage over North-America, Greenland, Europe and North Asia, including the Artic region, as well as over the surrounding oceans took place between 1945 and 1955. By 1975, Central and South-East Asia, Africa, South-America, Antarctica (along its coastline, except for the Amundsen–Scott South Pole Station) and the surrounding oceans were already covered – although not as well as farther north regarding the continental regions, with the noteworthy exception of China territory. While the global number of fixed stations measuring humidity has remained practically unchanged since then (see Fig. 4 ), their geographical distribution changed significantly. At present (2015) there are more observation sites in South America, much less in Central and East Africa, much more in Western Asia, and a more even distribution in the rest of the world.

Although oceans have been mainly covered by island stations, ocean weather-ships were important for more than 30 years, between the late 1940s and the early 1970s. Judging from IGRA data, coverage from these ships was optimal between 1967 and 1972, with 7 to 12 fixed ocean weather-ships transmitting radiosonde observations simultaneously, almost all located in the North Atlantic except for one ship in the Norwegian Sea and another one in the North Pacific. Apparently, upper-air observations from these platforms ended a few years later (save for the station "M" in the Norwegian Sea, which continued until 1990), coincident with the growing use of satellite retrievals in weather forecasting. However, in-situ data coverage on oceans was improved by using balloon sounding systems on board of merchant ships – which obviously are not shown in Fig. 3. About 10 to 20 ships of opportunity have launched radiosondes concurrently along their routes from the 1980s to present. Other floating, mobile stations are also worth mention.- The automated ice-drifting stations surveying the north polar region during 1950–91 were important considering their location and the amount of data gathered over the years. The missions to the Arctic and Antarctica performed by the ice-breaker and research vessel *Polarstern* (Driemel et al., 2016), covering also Atlantic Ocean regions during transit, provided substantial radiosonde humidity data in the periods 1985–1993 and 2000–2014. The weather ship *Polarfront* accounts for the largest amount of moving radiosonde data (after manning station "M" in the 1980's) but it operated only in the Norwegian Sea. For statistics about the humidity observations from all floating stations included in IGRA (fixed and mobile), see end of Table S1.

Figure 4 shows the count of fixed IGRA-RS stations with humidity observations in each year since 1930, by latitude bands of equal area, representing approximately tropical ($0^0$–$30^0$) and extratropical ($30^0$–$90^0$) latitudes in each hemisphere.

One can see that before 1937 there was only one observing station and three years without data (1933-34 and 1936). The few humidity-reporting stations existing by 1940 were placed in the Northern Hemisphere (in fact, most of them were in the Artic). From 1945 on they were located predominantly in the northern extratropics, even though they raised other latitudes, at the fastest rate in the subsequent two decades. From 1970 onwards, the number of stations in the southern extratropics did not

follow the observed growth observed elsewhere. This is not surprising since regions south of parallel $30^0$ S have significantly more ocean and less land. Note that the absence of mobile stations affects very little Fig. 4, since mobile stations are short numbered and have a short period of record (a few years), despite being important in covering the oceans. The northern extratropics have accounted for about half of the world stations for decades, although its relative weight has been decreasing over the years; the decrease in number of stations situated in that region after 1990 is counterbalanced by an increase in the

Tropics since then. Using Eq. (1) and the method outlined in Sec. 2.3.1, Fig. 5a-b shows the average distance to the nearest station ($\bar{s}$) as calculated from points on mostly land or ocean/sea regions within several latitude bands for every 15 years beginning in 1955, when the global radiosonde network was barely established. Figure 5c shows a similar calculation but for the total surface area of each latitude band. Clearly, since at least 1970, $\bar{s}$ has changed little over continents and large islands at all latitudes, with values ranging from $\sim 200$ km in land areas of the northern temperate latitudes ($35.0^0$–$66.5^0$ N) to $\sim 700$

km in land areas of the southern polar region ($66.5^0$–$90.0^0$ S), i.e., in Antarctica. In the Tropics ($23.5^0$ S – $23.5^0$ N), where $\bar{s}\sim$ 500 km over land during the same period, the little change observed despite considerable changes in the distribution of continental stations can be explained as follows: the better spatial coverage over South America at present is offset by the poorer coverage over a large part of sub-Saharan Africa, save for the country of South Africa (see Fig. 3, panels c and d). In contrast, in ocean/sea regions $\bar{s}$ not only is two to three times larger than over land regions – except in the southern polar

region, i.e. in the Southern Ocean when compared to Antarctica – but has also degraded slightly over time in the Northern Hemisphere oceans and seas from the subtropics ($23.5^0$–$35^0$ N) to the Arctic ($66.5^0$–$90^0$ N). From a global perspective, the hemispheric differences in $\bar{s}$ due to the distribution of oceans and continents can be appreciated in Fig. 5c. Note that $2\bar{s}$ gives an estimate of the average separation between adjacent stations in regions where the radiosonde network is relatively regular. For example, Fig. 5a indicates a typical separation of about 400 km in the northern temperate continental regions  (coincident

with the wealthier countries of the North Hemisphere – which is acceptable for synoptic weather forecasting. While the same is impracticable in many other parts of the world and over the oceans, distances up to two or three times larger than ideal need to be filled by satellite-based data and supplemented by surface observations, which are generally much denser than radiosonde stations. Nevertheless, on a scale suitable for climate monitoring, the WMO recommends that upper-air stations should have a maximum average separation of 1000 km (WMO, 2011b). This would require $\bar{s} \leq 500$ km. Figure 5b – which by including

costal stations does not represent the actual station density in deep-ocean areas – indicates that fixed stations are too far apart in most oceans, particularly in the southern midlatitudes where $\bar{s} > 1000$ km.

## 3.2 Average fraction of days in a year with humidity observations and size of missing days

Figure 6 represents the fraction of days in a year having humidity observations, averaged across all humidity-reporting fixed stations in each of the three major latitude regions of both hemispheres, along with the standard deviation. The plot begins in 1945, about when upper-air humidity measurements became routine in radiosonde soundings (cf. Fig. 1). In all regions, the average fraction of the days in a year with humidity observations increased rapidly until around 1960, stabilizing since 1965 to values in the range 70–80 % in low latitudes and 80–95 % in mid- and high latitudes. The corresponding standard deviations indicate that a non-negligible percentage of the stations have observations on every day in the year since the 1950s.

Figure 7 represents the typical 'size of missing days' in humidity observations in each year, as averaged across the fixed stations located within each major latitude zone. Recall that we focus only on sub-year missing days, since gaps of one or more years are exceptions related to interruptions of station's operation or maybe to the lack of a functioning hygrometer. So, Fig. 7 summarizes the typical continuity of humidity time-series having any observations in the year. As a general picture, the average size of missing days dropped from 4–6 months to about 1 month between 1945 and 1960; much of this change occurred before 1950, indicating that radiosonde measurements became rapidly regular in the early years. From 1960 to 2015, the stations at low latitudes present a trend in the size of missing days, from ≈ 30 days to 40 days on average; while stations at mid- to high latitudes present typical values of ≈ 20 days, except during the mid-1990s (≈ 30 days). Nonetheless, the dispersion of values away from the mean (not shown) indicates that some stations have time-series much more continuous than others, e.g. with daily data throughout the whole year.

The fraction of days with humidity observations, the size of missing days and the count of observations, detailed by station and year, is part of the first metadata set presented in this paper (see Sect. 4.1). Note that gap years are also evaluated in the metadata. Similar information is given for the Sfc-to-500hPa humidity soundings alone.

## 3.3 Global vertical extent and resolution of humidity observations

Figure 8a shows the distribution of the vertical-mean resolution in the annual humidity soundings since 1945, along with the homologous distribution in the simultaneous temperature soundings, limited to the highest level with humidity measurements for comparison. The vertical-mean resolution of each soundings was calculated by Eq. (2), rounded to the nearest decameter. For clarity, the curves displaying the mean and the quartiles are smoothed by a 5-year running mean. The differences between the distributions for temperature and humidity indicate missing humidity data in a few percent of the vertical levels with temperature data; although statistically irrelevant, such vertical gaps may be quite significant in individual soundings. The vertical resolution was relatively poor on average and highly variable in the early decades: until around 1965 the mean and median were coincident and varied in the range 1.1–1.4 km, with a midspread (interquartile range) of almost 1 km. Both the

average resolution and the midspread improved consistently from 1965 to between 2000 and 2005. Since 2005, ¾ of the soundings have a vertical-mean resolution better than 0.5 km, with half of the values ranging from 0.3 to 0.5 km.

Figure 8b shows how the maximum height above the mean sea level reached by either temperature or humidity measurements is distributed among the corresponding soundings on each year. Curves are smoothed by a 5-year running mean, as in Fig. 8a. However, the distribution is restricted to temperature and humidity soundings with surface data, these representing, respectively, 86.7 % and 86.0 % of the total RAOB soundings in the period 1945–2016. In addition, the soundings from mobile stations with missing geopotential height at the surface level are excluded, for consistency with the hypsometric calculations used in the dataset presented in this paper. According to Fig. 8b, by 2003, 75% of the temperature soundings with surface data reached an altitude of 22 km, i.e. ~ 50 hPa. Note that Durre et al. (2006) reported that, by the same year, 74% of all IGRA 1 soundings reached at least the 100-hPa level, i.e. ~ 16 km; this lower height is due to the inclusion of PIBAL data in their analysis of IGRA 1, whereas our analysis is restricted to RAOB in IGRA 2.

Contrary to temperature, which can be measured up to the maximum height achieved by the sounding balloon (burst altitude, although the highest reported level is usually limited by the standard levels in use), the vertical reach of humidity measurements depends on the working range of humidity sensors (and, to some degree, on reporting practices). In Fig 8b we can observe that the top of the humidity observations is at present situated almost 5 km below the maximum altitude of the temperature measurements (close to the burst altitude) on global average (either mean or median). But the difference in vertical extent between temperature and humidity observations has changed greatly over the years, with a maximum mean value of 12.5 km by 1980. Until the late 1960s the burst altitude increased at a much faster rate that the top of humidity observations, except for a few years after 1965 when the reverse happened. This last feature is coincident with the introduction of the carbon hygristor. Likewise, the isolated peak of the global mean maximum height of humidity observations around 1970, as seen in Fig. 8b ('HUM'), seems to indicate an exploratory period with the new instrument. Noteworthy, in 1970 the WMO stated that "no routine observations of humidity are made in the stratosphere and no practical use is envisaged for such current observations" (Hawson, 1970). Figure 8b undoubtedly shows that humidity was mostly measured in the troposphere until the mid-1960s. In contrast, from the mid-1980s onwards, the vertical extent of humidity observations has increased consistently and faster than the RAOB-top – roughly by 4 km per decade on global average –, denoting improvements of humidity sensors (see Sect. 1.3). In 2015, three quarters of the humidity-top values extend to over 13 km, and half to over 26 km, indicating that many humidity reports extend into the lower stratosphere. The accuracy of radiosonde humidity measurements in the stratosphere is beyond the scope of this paper. However, only 15 years ago, international experts pointed out that the accuracy of current operational measurements of humidity in the upper-troposphere was inadequate for addressing climate variability and change (despite the usefulness of some sensors) and a challenge for future operational radiosondes (Durre et al., 2005). It is interesting to note that the interquartile range of the top height of humidity soundings has increased much in the period 1985–2000, i.e., individual values became widely dispersed around the median. This is likely related to the proliferation of humidity sensors of different kinds, thus increasing the instrument variations among stations as new instruments coexist with

older ones. For instance, by 1989 the WMO had identified 20 major radiosonde types in use worldwide (Kitchen, 1989a). While this number has fallen to 13 by 2002 (Elms, 2003), Fig. 8b shows that the dispersion in the vertical range of different humidity sensors has increased in the meantime; changes in instrument-dependent observing practices may have a role.

5       The average vertical resolution of the humidity observations, by year and station, as well as the individual values by station, date and time, are given in the metadata sets provided in this paper (see Sect. 4). Similar metadata is provided for the Sfc-to-500hPa humidity soundings, as defined in Sect. 2.3.4, in which case the vertical resolution is calculated for the levels between the surface and the 500-hPa level and is normally finer than depicted in Fig. 8a.

      Since IGRA-RS observations do not have surface humidity data prior to 1945, and surface data is not always given after that year, to account for all soundings the vertical extent of humidity observations must be alternatively represented by 10   the lowest pressure corresponding to humidity data. The same applies to soundings from mobile stations if geopotential height is not given at the surface level (it is missing in 30 % of the corresponding RAOB). Although the local sea level is normally within ± 10 m from mean sea level, taking zero as the baseline height can lead to large relative errors if humidity is only measured very close to the radiosonde station elevation (balloon release height). Therefore, the average pressure of the top of humidity soundings, by station and year, are given in the first metadata set presented in this paper; average values are 15   represented by a geometric mean (see Sect. 4.1). As to the second set for individual observations –metadata by station, date and time – both the top pressure and the corresponding altitude, whenever this can be calculated, are provided (see Sect. 4.2).

## 3.4 Global relative amount of Sfc-to-500hPa humidity soundings

Recall that our definition of Sfc-to-500hPa humidity soundings (Sect. 2.3.4) is intended to represent the soundings with enough vertical level, and almost evenly distributed near the surface, such that the water vapor profiles can be properly described and 20   the precipitable water can be estimated. In this respect, the completeness of such humidity observations based on current standard pressure levels alone is unsatisfactory for two reasons: first, the level 925-hPa was barely used before 1992 (see Fig. 2); second, the sounding data at significant levels, often related to features of temperature rather than RH, or at other additional levels, are equally good provided that the vertical resolution (habitually increasing towards the surface) is not too different. Next, we will compare the distribution of Sfc-to-500hPa soundings among IGRA-RS stations in each year using alternative 25   definitions:

      A)   Humidity data at the surface and all current standard levels above the surface up to 500 hPa, i.e. at pressure levels $\{p: p \in \{1000, 925, 850, 700, 500 \text{ hPa}\} \cup \{p_{SFC}\}, \ p \leq p_{SFC}\}$;

      B)   Definition given in Sect. 2.3.4.

The soundings meeting either of the above two definitions are coined as *Hum-A* or *Hum-B* in the following analysis. Note, 30   however, that only definition B was used to prepare the metadata sets supplied in this paper.

      Let P be the percentage of Sfc-to-500hPa soundings (HUM-A or HUM-B, at our choice), out of all humidity soundings from an arbitrary station in a given year. Figure 9a shows the evolution of the number of stations with P for Hum-A soundings

exceeding given values, since nearly the time radiosonde humidity data at the surface level are first available. Comparing Fig. 9a with Fig. 1a, we can see that between 1945 and 1991 only a small fraction of the radiosonde stations carried out Hum-A observations in most of the soundings, say, with P within the range P > 95 %. Things changed drastically between 1992 and 1994, when the number of stations with a significant percentage of Hum-A soundings, e.g. 20 % < P < 80 %, increased by an order of magnitude. This change reflects the change in the observing practice shortly after 925-hPa was internationally adopted as a standard pressure level (see Fig. 2). In recent years there are over 400 stations with P > 95 % and almost 800 with P > 80 %. Figure 10a shows how the probability of finding a IGRA-RS station with a percentage of Hum-A soundings exceeding a given value was reversed over the last five decades: by 1965, only 12 % of the stations had at least 20 % of Hum-A soundings, whereas by 2015 ≈ 91 % of the stations had at least 80 % of Hum-A soundings.

Figure 9b is the counterpart of Fig. 9a, for the less stringent Hum-B soundings. Recall that 925-hPa is now treated in the same way as any near-surface non-standard level, and, for the rest, significant levels close to standard levels are allowed. (Note: the filling of missing data according to condition (ii) of Sect. 2.3.4 affected 22 % of the total Hum-B soundings.) One can observe that, before 1992, the number of stations having moderate to high percentages P of Hum-B observations is much larger than the number of stations having the same percentages of Hum-A observations; from 1992 on, that difference is moderate and only significant when P is very high. Besides the sudden increase of the stations with many Hum-B observations in 1992, there other two noteworthy change points: a sudden increase around 1970 and another one 2000. These changes are related to the increase of the global percentage of humidity soundings with surface data happening at about the same time (cf. Fig. 2). For example, the number of stations providing Hum-B vertical profiles in more than 80 % of the soundings doubled from ≈ 40 to 80 between 1969 and 1971, it doubled from 250 to 500 between 1990 and 1993, it increased from 520 to 700 between 1999 and 2001; a much more constant value of around 800 is observed in the period 2005–2015. The inverse cumulative distribution function in Fig 10b shows the probability of finding a IGRA-RS station with a percentage of Hum-B soundings exceeding a given value in different trienniums. About fifty years ago, the fraction of stations having at least 20 % of Hum-B soundings was as large as 40 %, increasing gradually over time to nearly 100% at present. Furthermore, 97 % of the presently active stations have at least 80 % of Hum-A soundings.

The number of Hum-B Sfc-to-500hPa humidity soundings in each station and year, as well as the fraction of days in a year having such soundings, are both given in the first metadata set presented in this paper (see Sect. 4.1). Information on other parameters describing humidity completeness but focusing on these soundings is also provided. The metadata set regarding individual observations identifies the Sfc-to-500hPa humidity soundings and provides information on their vertical resolution between the surface and 500 hPa (see Sect. 4.2).

## 3.5 Amount of long-term time-series

Figure 11a (black lines) illustrates the number of humidity time-series, one per station, fixed or mobile, with a 'current record length' (elapsed years until the year in abscissa) exceeding a given number of decades: 1, 3 and 5. The time-series refer

to periods of consecutive years with any observations in the year. For comparison (color lines) the same is shown for the time-series with data 90 % of the days in the year or more. It should be noted that the concurrent series with more than 10 years of back data begin in 1948, even though the first upper-air humidity measurements began in 1930 and by 1949 there were already around 300 stations measuring humidity (cf. Fig. 1a). The initial slope of the curves in Fig. 11a denotes the rapid growth of the global radiosonde network after the second world war, with many stations measuring humidity regularly. Nevertheless, the curves are not monotonous, due to the closing of stations in the past and to the existence of gap years in many stations. E.g., if we want to collect the largest amount of parallel humidity time-series extending back in time to over 10 years, Fig. 11a tells us that we should pick IGRA data until 1987–90, corresponding to about 750 radiosonde stations. However, as of 1976, the number of parallel time-series with humidity observations 90 % of the time or more and extending back in time to over 30 years has been increasing over the years, representing 200 stations in 2016. By restricting the series to those having sufficient vertical sampling between the surface and 500 hPa to estimate precipitable water vapor, the number drops to about 15 between 1978 and 2000, only then increasing steadily to about 50 by 2015. If we consider a much shorter duration, e.g. 10 years as a minimum, the time-series until the same year becomes several times more numerous and start much earlier.

Fig. 11b represents the distribution of the current record length under more restrictive conditions. First, it considers only the time-series consisting of Sfc-to-500hPa soundings (definition of Sect. 2.3.4) covering 90 % of the days in the year or more; secondly, it further limits the time-series to those with less than 10 consecutive missing days in every year, meaning that all months are evenly represented. We can see that the number of parallel time-series with regular (fraction of days in a year $\geq$ 90 %) Sfc-to-500hPa soundings and extending back in time to more than 10 (30) years has increased rapidly since 2000, after four (two) decades with a value of less than 100 (20), only then increasing steadily to about 330 (50) by 2016. However, the Sfc-to-500hPa time-series with the same length but being almost continuous (fraction of days in a year $\geq$ 90 %; size of missing days < 10 days) are much less numerous.

Figure 11 is only illustrative. There is no simple answer to the question of when we have enough data to (in theory) perform climate studies from radiosonde humidity data: it depends on the strictness of completeness criteria. (In practice, it depends also on the accuracy and homogeneity of measurements by different instruments.) Note that the length of the time-series under user-specified conditions for any of the metadata parameters defined in the next section, either backward or forward in time, can be derived from the information contained in the related metadata set (Sect. 4.1). In addition, the years with humidity observations can be found in Table S1 (as well as in a data file accompanying the two main metadata sets): one or more periods per station, depending on whether there are gap years or not in the station's full period of record.

**4 Metadata sets of completeness of radiosonde humidity observations based on the IGRA**

The metadata sets describing the completeness of radiosonde humidity records collected in the IGRA are outlined next. They are constructed upon the metadata parameters introduced in Sect. 2.3 and examined in Sect. 3, except for minor adaptions described in the subsections below. The combination of selection criteria, by simply specifying value ranges for the metadata parameters, offers a plethora of choices to the user. Both sets refer to sounding data from the IGRA-RS sub-set of IGRA 2 stations, as defined in Sect. 2.2 and listed in Table S1, which essentially excludes pilot-balloon stations.

**4.1 Metadata by station and year**

The metadata parameters regarding each IGRA-RS station and year (annual statistics) are defined in Table 2. Since pressure is the vertical coordinate that is always present in humidity radiosonde data, and the calculation of height above mean sea level is impossible when the surface level is missing, the average vertical extent of humidity data at a given station and year is represented by the geometric mean value of the lowest pressure with humidity data. (Given the huge number of values involved, the arithmetic mean of the logarithm of pressure was calculated first and then exponentiated.) Note that the geometric mean pressure of the highest level with humidity data provides a natural measure of the corresponding arithmetic mean altitude: insomuch as pressure decrease almost exponentially with height, the arithmetic mean height above sea level is roughly proportional to the logarithm of the geometric mean pressure. The statistics for the vertical resolution of humidity data consists in the annual mean of the 'mean vertical resolution' of individual soundings at each station, calculated by Eq. (2). The IGRA data meeting the definition of Sfc-to-500hPa humidity soundings as stated in Sect. 2.3.4 have their own metadata: number of observations, average vertical resolution, size of missing days, and fraction of days in a year having humidity data.

The metadata set is as a plain-text file, each data record containing the metadata values, station-by-station and year-by-year. For each station, the yearly variables defined in Table 2 are displayed chronologically. To put humidity metadata into context, the yearly number of soundings and of RAOB soundings are also given in the dataset. The years within a station's period of record with observations of any meteorological variable (wind, temperature, humidity) are kept, using appropriate missing values when humidity data is missing throughout a whole year. While humidity data begin in 1930, pre-radiosonde years are included to preserve the stations history.

NB – The metadata set is complemented by a list of the IGRA-RS stations with information on geographic coordinates (if fixed), name, and country, along with original metadata describing the periods of record for humidity and the corresponding amounts of observations. This is an ASCII version of Table S1, for computing purposes.

**4.2 Metadata by station, date and time**

The metadata parameters regarding individual soundings at each IGRA-RS station are defined in Table 3. The utility of the corresponding metadata set is to allow a fine selection of humidity data, as a complement to a first selection based on the statistical parameters of Table 2. This metadata set is organized into one plain-text file per station, and one data record per

sounding. Below the headline, each record contains: the date, nominal hour, latitude and longitude of the balloon launch, followed by the related humidity metadata.

## 5 Availability of metadata sets

A dataset combining the two metadata sets presented in Sect. 4 is available on Zenodo, DOI: 10.5281/zenodo.1332686. The accompanying 'Readme' file gives the necessary information about the data format and file contents. The update is planned to take place each time two full years have been completed in IGRA, starting on January 2019 for the period 2017–2018.

**6 Summary and recommendations**

This work has studied the completeness of radiosonde humidity observations compiled in the IGRA Version 2 (Durre et al., 2016; Durre et al., 2018) upon setting aside the IGRA stations with more than 95% pilot-balloon data in every year of their period of record until the end of 2016. The selected set (denoted IGRA-RS) retains virtually all RAOBs distributed by 1723 stations, including 1300 WMO stations, of which 178 and 16 are, respectively, current GUAN and GRUAN sites. The earliest
humidity observations are from the 1930s, when the radiosonde era had begun, but the data amount is only significant after around 1945. Several parameters describing the completeness of humidity observations were defined and then examined in statistical terms, providing a global picture of humidity completeness in radiosonde observations over time. The main conclusions, for the years beginning in 1945, are as follows.

- The radiosonde network providing humidity observations in the northern temperate and polar latitudes was essentially
established in the mid-1950s. In the mid-1970 the globe became covered by practically the same number of humidity-reporting stations (nearly the same as radiosonde stations) as it is today, although the distribution of stations has changed considerably over time. The averaged distance to the nearest station measured from points over continents and large islands (from Mindanao up to Greenland in size), ranging from ∼ 200 km in the northern midlatitude countries to ∼ 700 km in Antarctica, has little changed too. However, the spatial distribution of continental stations
continue to show important regional disparities, such as the poor coverage of most of sub-Sharan Africa which has worsened from 1970 to present. Concerning ocean/sea areas, and disregarding mobile stations (mostly 'ships of opportunity'), the oceans of the southern temperate latitudes always exhibit the poorest coverage, with an average distance to the nearest station exceeding 1000 km. Remarkably, since during the last half century the average distance to the nearest station has increased considerably in the North Hemisphere oceans and seas extending from the
subtropics to the Arctic.

- The fraction of days in a year with humidity observations has greatly increased until 1960, on average calculated for each year across the stations on each of the main latitude regions. Since 1965, the mean values are in the range 70–80 % in low latitudes and 80–95 % in mid- and high latitudes annually, although individual values vary widely among stations. Between 1945 and 1955, the 'size of missing days' in humidity observations (largest gap in days) in each year having any humidity data decreased from about five months to one month on global average. After 1960, the size of missing days exhibits averages around 20 days in middle and high latitudes; in low latitudes it has increased over time from around 30 to 40 days.

- Humidity was measured mostly in the lower to middle troposphere until the mid-1960s. Since the mid-1980s, the mean height achieved by humidity measurements has increased by $\approx 4$ km per decade on global average; the gap distance to the burst altitude (denoted by the maximum height of the temperature measurements) more than halved. At present, that gap is $\approx 5$km and three quarters of the humidity soundings reach to at least 13 km altitude. However, the dispersion of the maximum height of humidity data around the mean has increased too, likely due to the coexistence of older humidity sensors with new sensors. The vertical-mean resolution of humidity observations (geometric mean of the distance between consecutive levels with data), has a median in the range of 1.3–1.4 km until 1965, among all humidity soundings in each year, with an interquartile range of $\approx 1$ km. Thereafter, the median and midspread have improved constantly before stabilizing in the early 2000s; since then, half of the annual soundings present a vertical-mean resolution between 0.3 and 0.5 km.

- The number of stations providing Sfc-to-500hPa profiles with adequate resolution for calculating precipitable water vapor in more than 80 % of the reported humidity soundings in a year has changed from a few dozen in the period 1945–1970 to around 800 in recent years (97 % of the active radiosonde stations by 2014/16). In general, the amount of stations having a significant percentage of Sfc-to-500hPa observations shows a sudden increase around the years 1970, 1991/92 and 2000. These change points are associated with the availability of data at the surface level and at non-standard near-surface levels; the latter provide important information in the planetary boundary layer before 1992, when the 925-hPa level was not standard and was rarely used.

- The amount of humidity time-series with a given number of consecutive years of data until a given year depends not only on the year and the length of record, but also on other completeness criteria. For example, the station-based time-series extending back in time for at least 30 years and being 90 % complete (in terms of fraction of days in a year) begin with a few units in 1977 and represent around 200 stations in recent years; by further requiring Sfc-to-500hPa completeness, the other criteria being the same, the number of time-series drops to 50 by 2016. Evidently, the equivalent time-series until the same year but with a much shorter duration, e.g. 10 years as a minimum, are several times more numerous and start much earlier. In short, the amount of humidity time-series that are *potentially* available to perform climate studies, i.e., discounting accuracy requirements and biases due to instrument changes, depends strongly on the strictness of the completeness criteria.

Furthermore, this paper presents a dataset detailing the completeness of humidity observations (RH or DPD together with pressure and temperature) based on the data from the IGRA-RS stations. The dataset (Ferreira et al., 2018) consists of: 1) statistical metadata for each station and year, in a single file for the 1723 stations and their full period of record until 2016; 2) metadata specific to individual soundings, organized in one file per station and covering 39.5 million soundings for the same stations; and (3) list of stations along with the observing periods for humidity and the corresponding number of observations. The metadata parameters were designed to facilitate the selection of upper-air humidity data from IGRA according to a plethora of choices, therefore being able to meet specific research needs in the areas of atmospheric and environmental sciences.

It is widely known that the usefulness of historical radiosonde data depends crucially on metadata information about instrumentation and observing practices (Schwartz and Doswell, 1991; Elliot and Gaffen, 1991; Gaffen et al., 1996; Parker and Cox, 1995). However, sampling differences among stations associated with geographical coverage, observing periods and missing data, are no less important than differences in data precision and accuracy. Reporting practices related to limitations of humidity sensors affect particularly the humidity records (Garand et al., 1992; McCarthy et al. 2009). In this respect, the metadata presented in this paper, if used as a tool to find out the more complete humidity time series – relatively long, regular and continuous; vertically extensive and well resolved –, should be crossed with the coordinates of stations and the station history metadata available in IGRA. On the other hand, the present metadata might accelerate progress in the current research on the homogenization of radiosonde humidity data (McCarthy et al., 2009; Dai et al., 2011), by sampling stations with coincident observing periods and satisfying reasonable completeness criteria.

**Acknowledgments**

We thank Imke Durre for clarifying some points about IGRA Version 2 during the preparation of this work, and David Carlson and two anonymous reviewers for their helpful comments. We also thank to Orlando García for IT support at EPhysLab (Environmental Physics Laboratory at the  Faculty of Science of the University of Vigo, Ourense Campus). This work was funded by the Spanish government within the EVOCAR project (CGL2015-65141-R), co-funded by the European Regional Development Fund, and partially supported by the Galician regional government through the program Consolidation and Structuring of Competitive Research Units – Competitive Reference Groups (ED431C 2017/64-GRC).

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

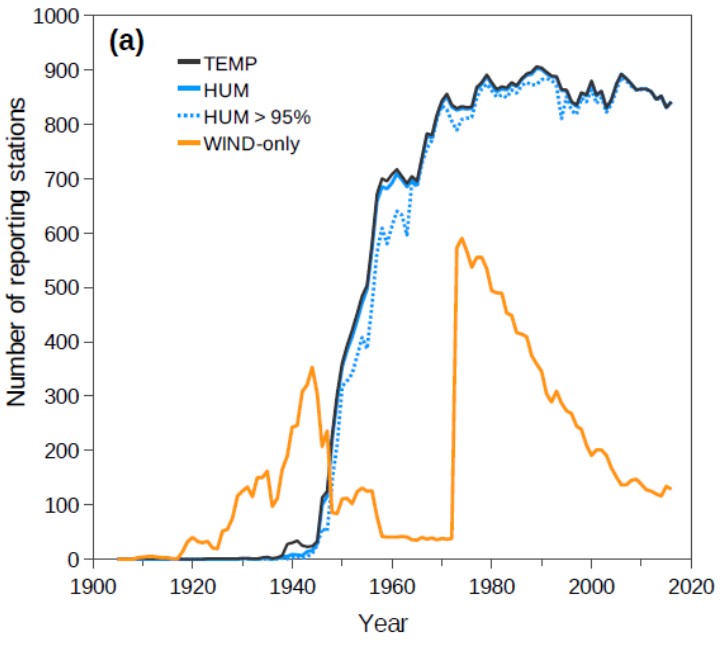 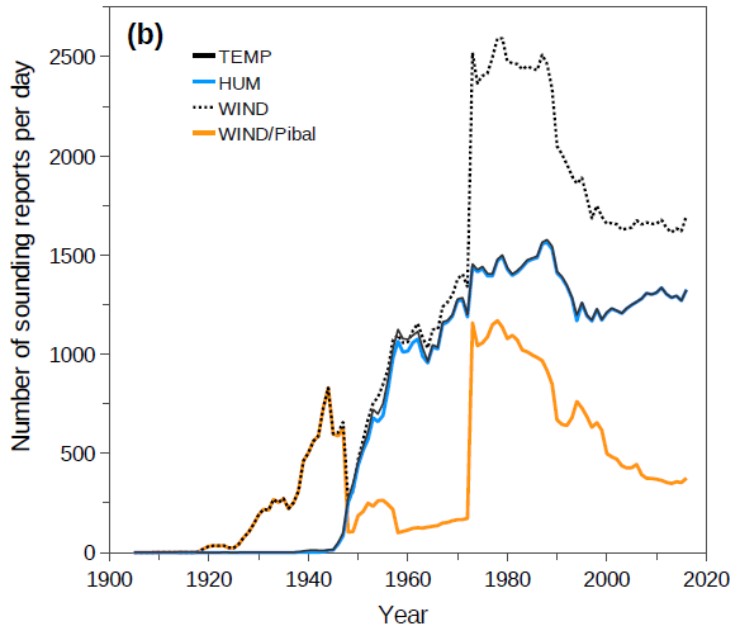

**Figure 1**

(a) Number of IGRA stations, for each year until 2016, reporting: temperature observations, regardless of humidity and wind (TEMP); temperature and humidity observations, regardless of wind (HUM); humidity in at least 95 % of the RAOB (HUM > 95 %); only wind observations (WIND-only). (b) Number of sounding reports per day compiled in IGRA, by atmospheric parameter: temperature (TEMP); humidity (HUM); wind from radiosonde or pilot-balloon measurements (WIND); wind from pilot balloons (WIND/Pibal). Note: in panel (a), the sum TEMP + WIND-only gives the global number of stations with any data in IGRA.

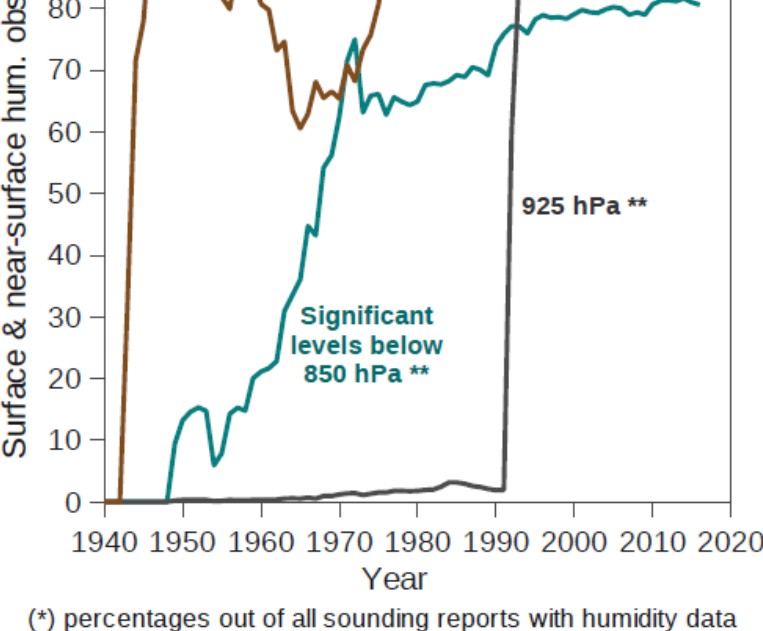

**Figure 2**

**Percentage of humidity observations: at the surface level out of all humidity soundings in each year; at 925-hPa, out of the soundings with surface data below that level; and at any non-standard level – excluding the 925-hPa, irrespective of the year – between the surface and the 850-hPa level, whenever this is above the surface.**

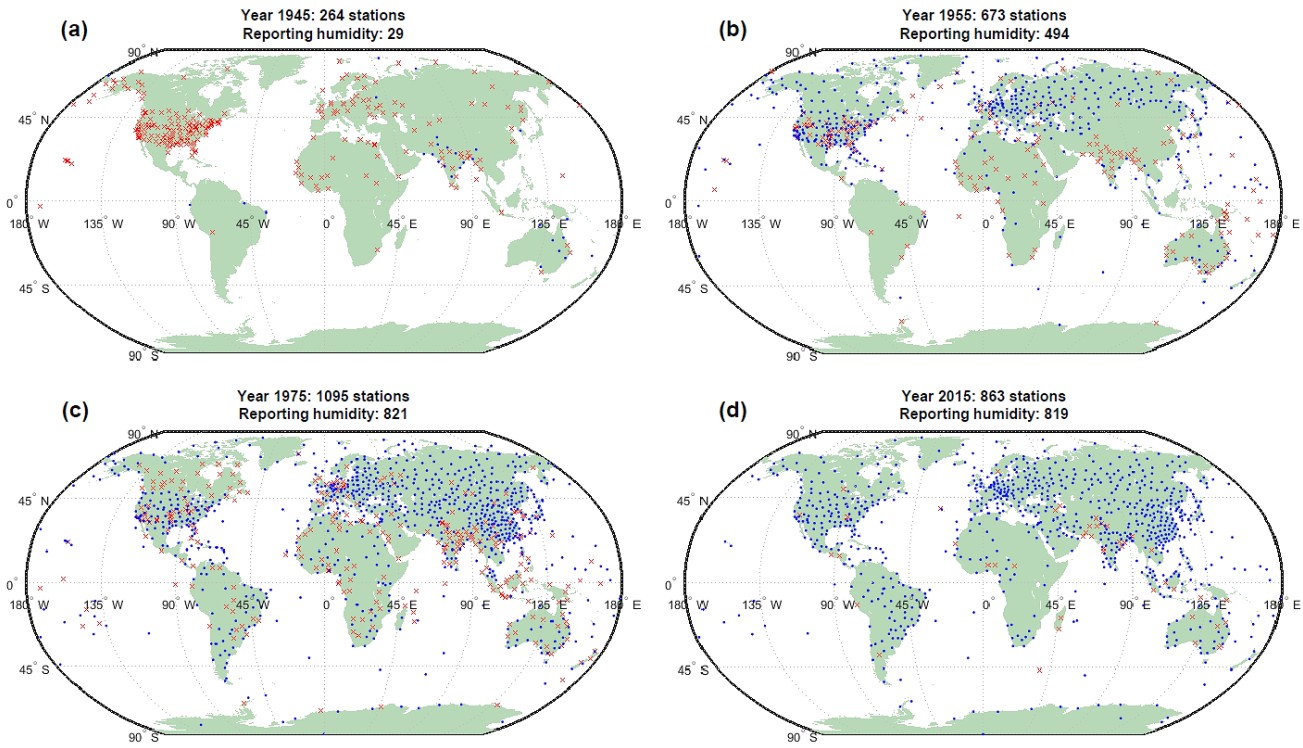

**Figure 3**

Locations of IGRA-RS fixed stations active in the years of (a) 1945, (b) 1955, (c) 1975 and (d) 2015. Note that the interval of time doubles between consecutive panels. Blue dots denote the stations reporting humidity (RH/DPD + temperature) anytime in the year; red crosses denote stations reporting only temperature/wind, or else presenting a data record gap, during the same year. The total number of stations (crosses + dots) refers to the active stations in each year regardless of any data record gaps.

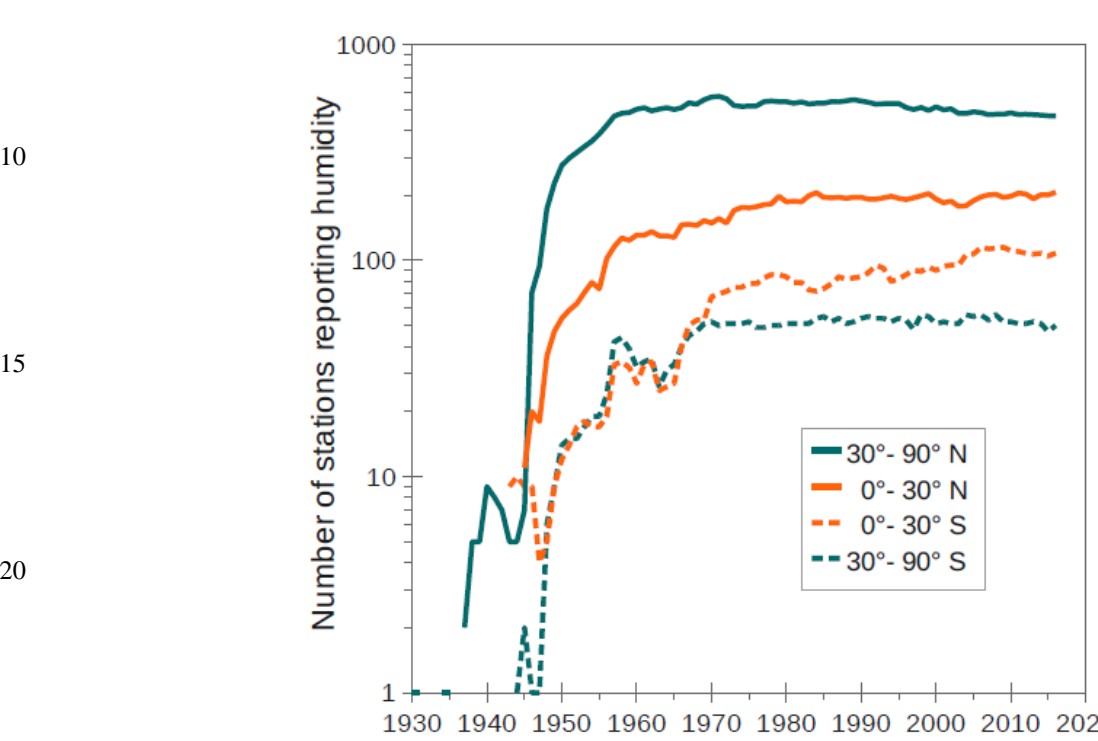

**Figure 4**

**Yearly number of IGRA-RS fixed stations reporting humidity in northern and southern tropical and extratropical latitudes, since the time of the earliest radiosonde humidity observations.**

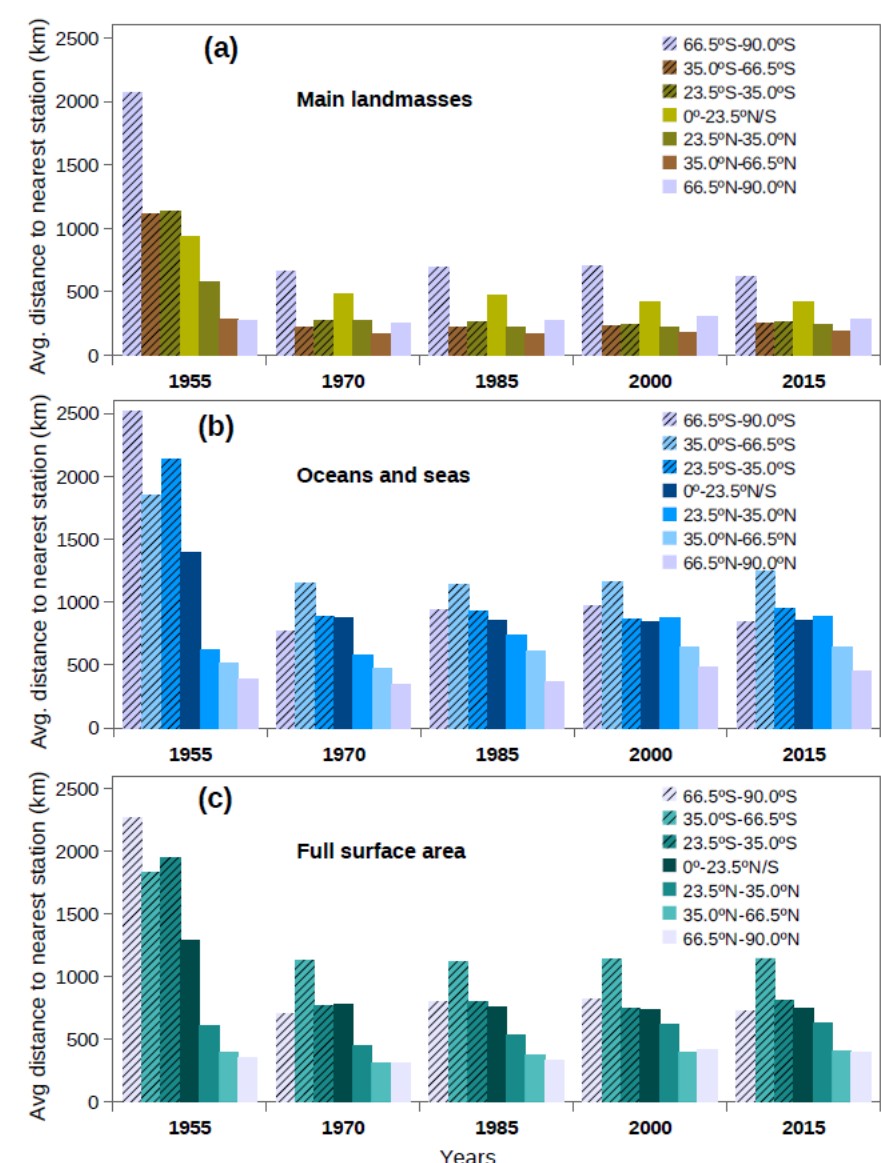

**Figure 5**

Average distance to the nearest station, among the humidity-reporting stations from 1955 to 2015 in 15-year intervals, as calculated on specific areas of each climatic zone: (a) Main landmasses; (b) Oceans and seas; (c) Full surface area. See text for calculation details (Sec. 2.3.1).

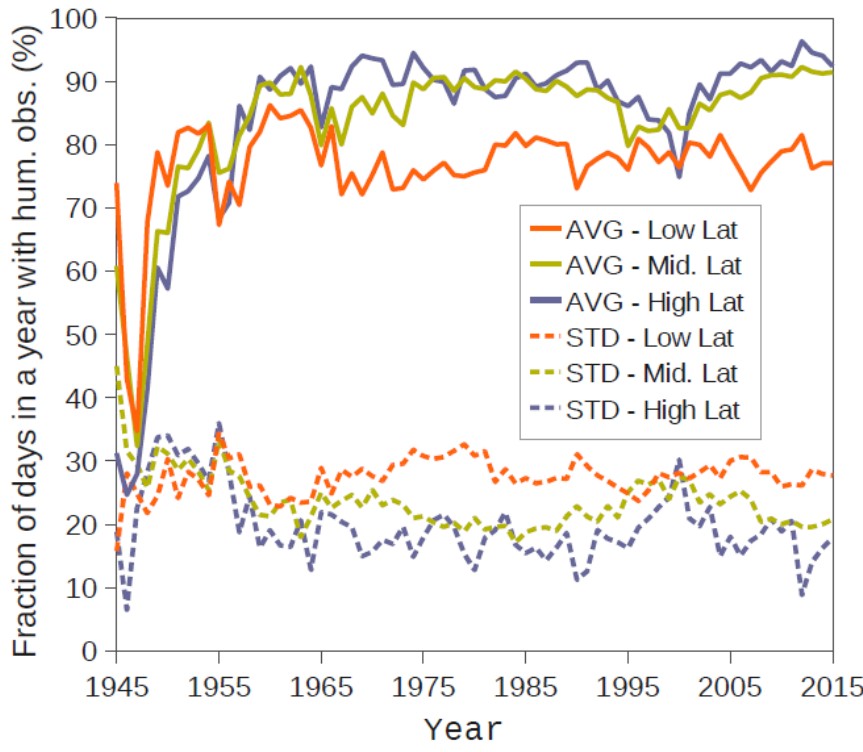

**Figure 6**

Statistical measures of the fraction of days in a year with humidity observations from 1945 to 2015, representing the fixed radiosonde

stations over the major latitude regions for both hemispheres: average among stations with humidity data, and standard deviation.

Low Lat – abs. latitudes < 30º; Mid. Lat – abs. latitudes between 30º and 60º; High Lat – abs. latitudes > 60º.

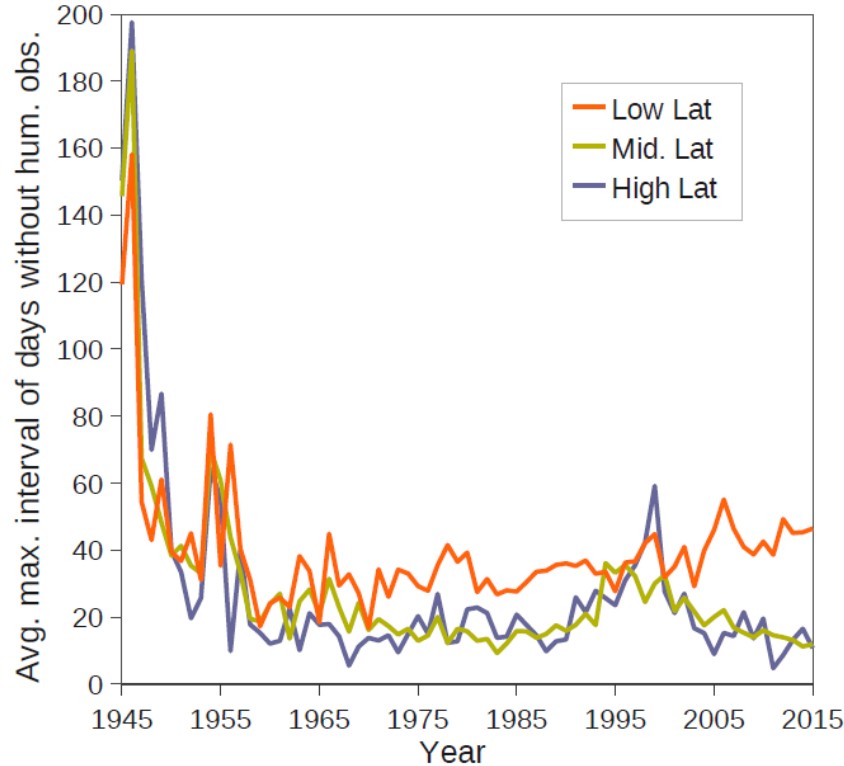

**Figure 7**

**Maximum number of consecutive days without humidity data in each year with some humidity observations ('size of missing days'),**
30  **averaged among the fixed stations within each of the major latitude regions defined as in Fig. 6.**

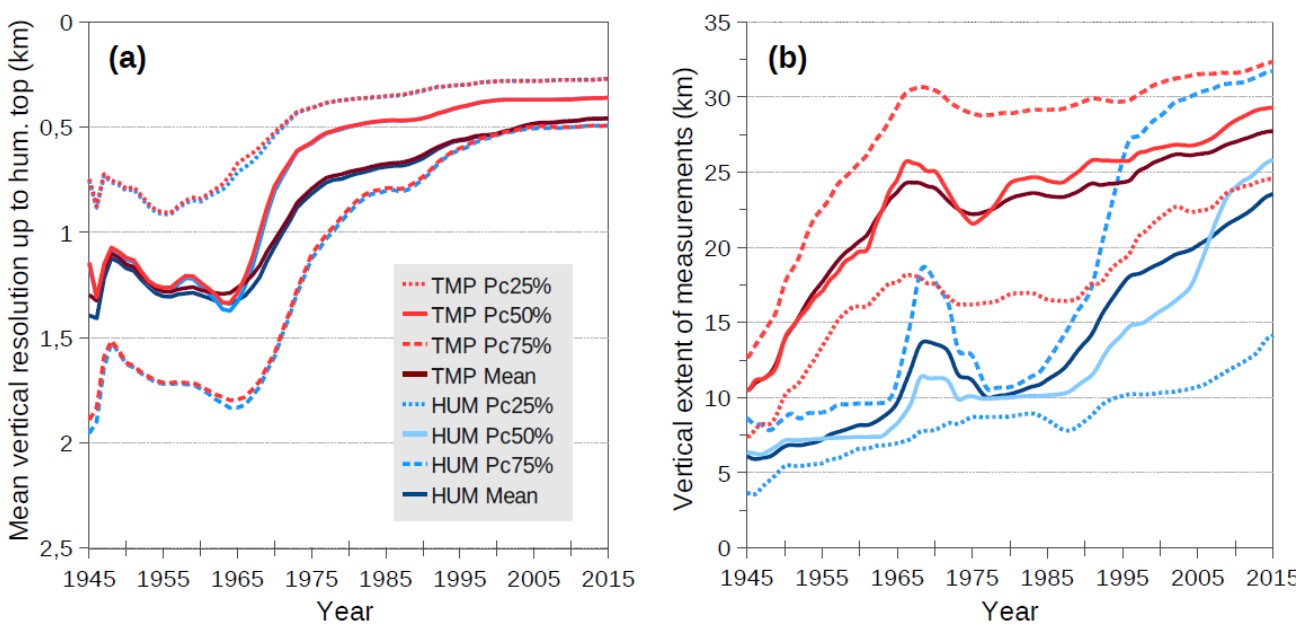

**Figure 8**

(a) Distribution of the 'mean vertical resolution' (definition of Eq. (2)) in the temperature (TMP) and humidity (HUM) observations across the globe, calculated up to the highest measuring level for humidity: Mean and quartiles. (b) As in (a), but for the vertical

10  extent of TMP and HUM observations, restricting to observations with surface temperature data. All the curves are smoothed by a 5-year running mean.

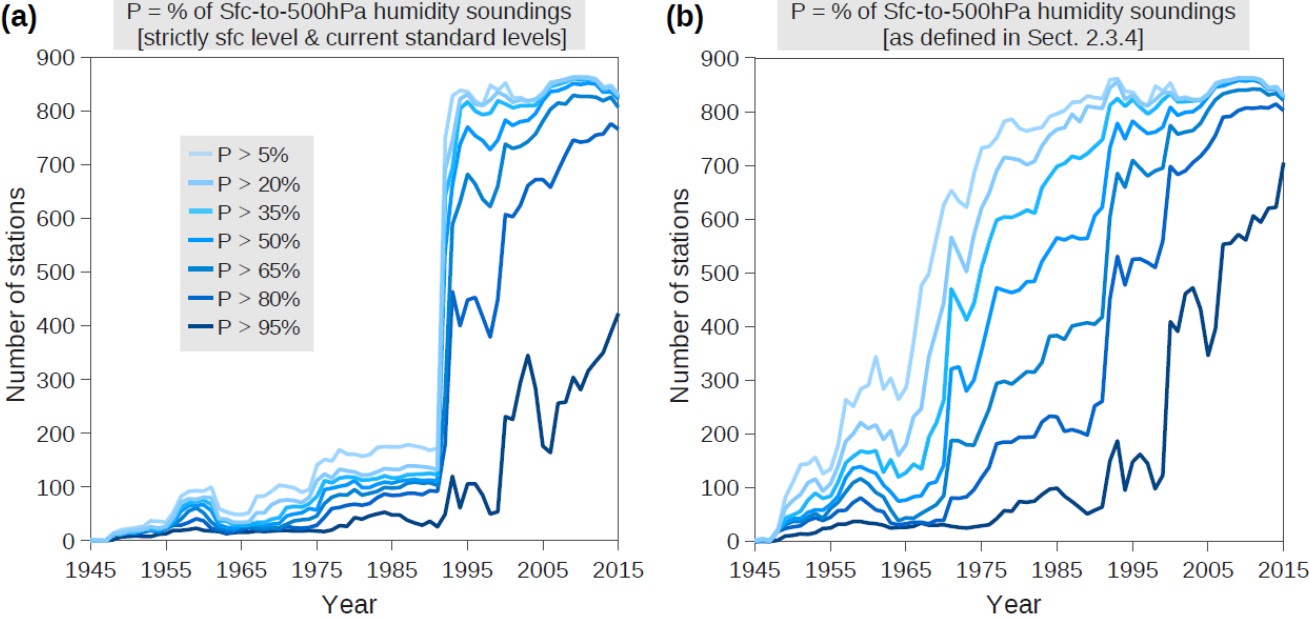

**Figure 9**

10   Number of stations for which the percentage of Sfc-to-500hPa humidity soundings, out of all soundings with humidity in each year, exceeds a given value (see inset of panel a): (a) Requiring humidity data at the surface level and all current standard levels above the surface up to 500 hPa; (b) Using the definition of 'Sfc-to-500hPa humidity soundings' given in Sect. 2.3.4.

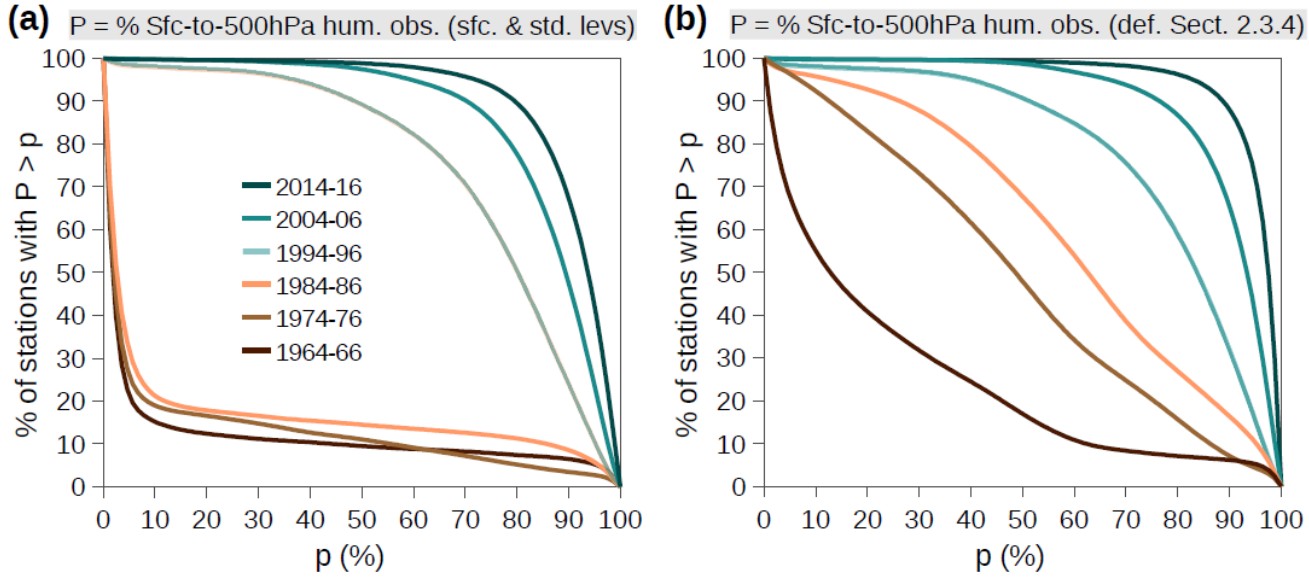

**Figure 10**

Percentage of stations, out of all IGRA-RS stations, with a percentage of Sfc-to-500hPa humidity soundings exceeding a given value (shown in abscissa), out of the humidity soundings in several trienniums with 10 years interval (see inset of panel a): (a) Requiring humidity data at the surface level and all current standard levels above the surface up to 500 hPa; (b) Using the definition of 'Sfc-to-500hPa humidity soundings' given in Sect. 2.3.4.

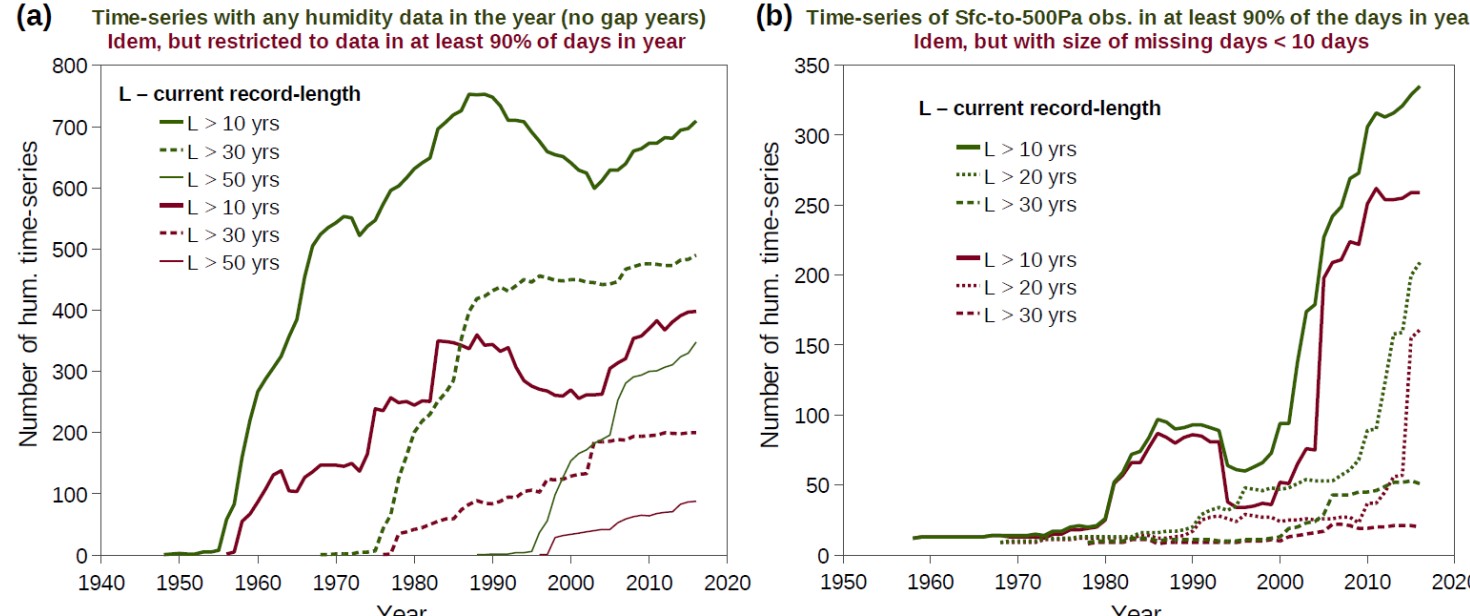

**Figure 11**

Number of humidity time-series with a record length exceeding a given number of years until the year in abscissa (see insets), under different completeness criteria (indicated on top of the graphs).

5    **Table 1. Number of soundings in IGRA and IGRA-RS subset until the end of 2016**

|  | **IGRA** | **IGRA-RS** |
|---|---|---|
|  | (2761 stations) | (1723 stations) |
| Total (PIBAL + RAOB) | 45,677,409 | 39,526,638 |
| PIBAL (*wind-only data*) | 15,463,235 | 9,312,891 |
| RAOB | 30,214,174 | 30,213,747 |
| RAOB with humidity data | 29,801,708 | 29,801,324 |

10    **Table 2 Parameters of humidity completeness for each IGRA-RS station and year**

| **Parameter** | **Description** (*Statistics refer to a year within the period of record of the station*) |
|---|---|
| HUMa | # of soundings with humidity observations |
| HUMb | # of soundings with Sfc-to-500hPa humidity observations[1] |
| RESa | Annual mean vertical resolution of humidity observations[2] |
| RESb | Annual mean vertical resolution of Sfc-to500hPa humidity observations[3] |
| TOPP | Annual geometric mean pressure of the highest level with humidity data |
| GAPa | Largest interval of days in the year for which humidity observations are missing |
| GAPb | Largest interval of days in the year for which Sfc-to-500hPa hum. obs. Are missing |
| FDYa | Fraction of days in a year with humidity observations |
| FDYb | Fraction of days in a year with Sfc-to-500hPa humidity observations |

([1]) As defined in Sect. 2.3.4. ([2]) Arithmetic mean of the 'mean vertical resolution' given by Eq. (2).
([3]) Calculated as RESa, but between the surface and 500 hPa.

**Table 3 Parameters of humidity completeness for each sounding**

| Parameter | Description |
|---|---|
| RAOB | Classifies the sounding according to RAOB data content: 0 = none (only wind data); 1 = TEMP data, but missing HUM; 2 = TEMP and HUM data; 3 = TEMP and Sfc-to-500hPa HUM[1] |
| RESa | Mean vertical resolution of humidity data values[2] |
| RESb | Mean vertical resolution of Sfc-to-500hPa humidity data[3] |
| TOPP | Atmospheric pressure of the highest level with humidity data |
| TOPZ | Altitude above mean sea level of the highest level with humidity data |

[1] As defined in Sect. 2.3.4.  [2] Calculated by Eq. (2).

[3] Calculated as RESa, but between the surface and 500 hPa.

