# Peer review of "Completeness of radiosonde humidity observations based on the IGRA"

_Earth System Science Data, 2018_

## Referee Comment (RC1) · Anonymous Referee #1 · 23 Nov 2018

The manuscript by Ferreira et al. presents an analytical description and a rationalized statistical analysis of the radiosounding data archive available through the NOAA-IGRA initiative. The authors report an extensive and precious description of a huge number of information about the changes occurred in the number, geographical density and type of radiosoudings and payloads used around the globe since the beginning of last century to present for the measurement of atmospheric humidity. The authors focus they study on humidity measurements and they investigate the datasets in a critical way to show if, in term of continuity and coverage, the available data may support climate studies, though they do not assess this aspect on a scientific sound basis. The paper is well written and curated and the provided analysis and it update expected on a biannual basis may support many activities currently ongoing, not ultimately the

[Figure]

Copernicus Climate Change Service (C3S). I am in favor of the manuscript publication. Nevertheless, I have to provide the authors with major revisions, because to my opinion a few concepts must be clarified and modified in the text of the manuscript, though this does not affect my positive view on the manuscript itself. Below I enclose my general comment and a bunch of minor issues to solve.

General comments. 1. In the manuscript, starting form the abstract, it is mentioned that the IGRA V2 datasets is investigated according to various completeness criteria in order to show if the length of the available data records is sufficient for the purposed to estimate climate variability trends. The authors mainly refer to the temporal continuity of the observation and not the spatial gaps which are anyhow investigated in the statistics. Despite this purpose, the authors clarify in the conclusion that their work can only be supportive of climate studies or works related to the climate homogenization of the time series, and nor they do not discuss on a quantitative basis what is the length of data records required for climate studies neither they refer to past literature to asses this aspect. Assuming that the authors are not interested in the going ahead in the assessment of the effect of gaps in the data records on the estimation of climate signals (which is well beyond the scope of the paper), I'd ask them to change the tone of the manuscript, where needed, to better clarify the scope of the manuscript. 2. The authors make use of IGRA data V2, which is the most updated version of IGRA which embed several improvements compared to the previous data version (V1). Nevertheless, in my personal researches I had a chance to find many bugs in the IGRA V2 where many data present in the V1 are missing and this is not dependent on the extended quality check applied within the IGRA V2. I had also a chance to report this bug to Imke Durré (PI of IGRA). I got similar feedbacks from other EU and US colleagues during discussion at various workshops. The station of Lindenberg in Germany (WMO index=10393), so accurately described in its past history in the manuscript, is one those affected by this issue (at lest until a few weeks ago, my last access to IGRA). Therefore, I am wondering whether a comparison between IGRA V1 and V2 has been carried out and if gaps have been found and fixed somehow in the datasets investigated in the manuscript. 3. The

description of statistical analysis of IGRA V2 data is very extensive way, providing several details and a long description of each figure and table. When reading, I have been very interested by the content, though sometimes the reader may get tired by the way the manuscript is written. In a way similar to the conclusions, I'd suggest to the authors to change the style of their writing and privilege a description in "bullets" to describe the results whenever possible. this will help also to clarify the text itself. For example, for the relative humidity observations, At pag.5, it is reported that the average number of non-standard levels in weather-balloon sounding reports increased from about zero by 1945 to about 30 by 2000, but later in the text it is said that RH observation were already available since the 1930, and again later on (pag. 18) it is stated that RH are becomes more abundant since 1949. Though all of this information are exact the reader may get confused and their comparability and usefulness could be limited if the statical analysis is not described in a more schematic way. 4. The use of the metric present in the Eq.1, "zonal coverage index" at pag 11 is not clear to me. Have been it used in the past and its adde value with respect to other metrics was shown? What's the added value with respect to a station density per 1000 km, for example? Has the ocean surface been excluded from the global surface, given that this can be calculate more clearly for fixed stations? There are many concerns to me on the use of this index, which requires clarification from the authors. Personally, I think that the user can make use of a much simpler index or statistics (a few of these are used later on in the manuscript by the authors themselves) to show if a zonal belt are under and over- represented. This is also depending on the different atmospheric circulation occurring in the different zonal belts, so I am wondering what's the usefulness of adopting this metric. I ask the authors to clarify in the manuscript of the added value due to the use of this metric. 5. The average vertical resolution could be useful information, but to my opinion, thinking about different applications it could be useful to have a statistic of the available level for the different regions of the atmosphere: Planetary Boundary Layer, Free Troposhere, Upper Troposphere/Lower Stratosphere (UT/LS), Upper Stratosphere. This classification may have a stronger
impact to orienting users' application in the selection of the available data, e.g for trends calculation. 6. Please check the use of the term "error" throughout the manuscript and replace it with "uncertainties" where more appropriated.

Specific comments Pag.2, lines 10-12: the authors could mention that recent reanalysis products, for example ERA5 from ECWMF, will improve the 4 times daily frequency of the products up to hourly.

Pag.2, line 13: replace "not to mention ….." with "with not negligible ….."

Pag.2, lines 17-19: uncertainties due to balloon drifting and observation time are considered negligible citing the publication by Kitchen (1989) as well as the radiosonde profile accuracy. To my opinion these cannot be considered minor, and anyhow if minor or not this is depending on the considered application. First of all there more recent papers by Seidel et al. (2011) dealing with radiosonde balloon drifting. Estimates of elapsed time from balloon launch to various pressure levels, due to vertical balloon rise, have median values increasing from about 5 min at 850 hPa to about 1.7 h at 10 hPa, with ranges of about 20% of median values. Observed elapsed times always exceed those estimated using assumed 5 or 6 m/s rise rates. Regarding the data data accuracy, if we are referring to the ensemble of effect which may alter the sensors' optimal measurement conditions, like the solar radiation effect to the effect of a sensor time-lag, these have been better quantified for the more recent radiosonde types and are quite relevant for any kind of climate application (see for example the GRUAN quantification by Dirksen et al. 2014). This is also the reason why many scientific groups have developed homogenized dataset of radiosonde data for climate application. For there reason above, I'd reformulate these line in the manuscript and I'd provide more updated references.

Pag.2, line 26: please replace "errors" with uncertainties and then add also that the radiosonde sensors may have a limited sensitive to ppm water vapor concentrations in the UT/LS as one of the main reason because humidity data above 300hPa are

unreliable.

Pag.2, line 32: please change "lag" time "time-lag".

Pag.3, lines 19: "the remainder of this section".

Pag.3, lines 21: put "available" in between of "levels" and "in radiosonde"

Pag3, line 26: "….relevant for the study". Study of what?

Pag3, line 26: " please change "indicated" with "reported".

Pag4, lines 7-10: Please put a reference related to the importance of data continuity for climate studies (trends, annual cycles).

Pag4, Line 10-12: I tend to disagree with the introductory sentence while I like the authors approach in the manuscript; therefore, I think in this sentence you must report which is needed to investigate the simultaneous spatial and time sub-sampling on the data whatever challenging this might be in the practice.

Pag. line 24: pressure in not measured anymore in the most recent radiosonde types (e.g. RS-41 operation since a couple of years); please for completeness you may mention this.

Pag.5, line 17-19: this sentence could be a good opportunity to claim for the importance of having high resolution measurements and, therefore, more levels available in the radiosounding report. This is in line with the high resolution BUFR files already flowing to the Met services from more than 100 station worldwide.

Pag. 7, line 13: "anything but uniform" can be modified "quite heterogeneous".

Pag. 7, lines 22-24: these sentence is the "official" IGRA description, but going through the data the authors may realize that among the 2761 stations, many of them are "near-surface" stations and not radiosounding station. All the reports are empty (-9999) for many station. These must be clarified and the reported number estiamted in a more

precise way.

Pag. 7, line 16: Is the reported typical average accuracy in the troposphere only related to the most recent radiosonde types? Please clarify.

Pag.11, line 1: please put a descriptive reference for GRUAN, I suggest Bodeker et al., 2016 BAMS.

Pag.11 line 3: It is not true that all of the GRUAN sites are transmitting data to the GTS. A few sites are still working to establish this data flow. This is also connected to the sentence at line 9, reporting the fact that not all GRUAN station are present within IGRA.

Pag.11, line 6: The added value of GRUAN is not only to provide data the quality of which should be "above the average", but to provide traceable uncertainties and a fully disclosed data processing described in peer-reviewed literature. Please add more details to show the real added value coming from GRUAN

Pag.11, eq.1: see my general comments above.

Pag.12, line 18: what's the meaning of the "relevant" in this sentence?

Pag.12, line 19: How did you choose the value of the constant temperature T0, please clarify in the text.

Section 2.3.3: this section refers to the quantification of the number of soundings which can be qualified to estimate precipitable water vapour. I am not sure to what extent this section may really confuse the reader. From one side, I think this is redundant with assessment of other indicators in the manuscript and would add value if then the selected radiosoundings according the criteria reported in this action may ready represent soundings for which an accurate estimation of the water vapor is feasible. i think the authors should clarify that thorn the radiosoundings selected according to the presented criteria allow to calculate an estimation of water vapor content which is the closest possible to the true one given the small number of vertical level available.

The accuracy of the calculation of precipitable water vapour for these radiosoundings is anyhow affected by many other aspects: presence of clouds affecting the measurement sensors, homogeneity the water vapor field close to the surface, non-linearity of the water vapor variability along the vertical profile and so on.

Pag.16, line 26: Please provide more details to explain the differences in the maximum range covered by the measurement of these parameters. For example, the way wind and humidity have been measured in the past compared to temperature?

Pag. 21, line 1: Did the metadata adhere to any international standards like ISO19115 or WIGOS? Please clarify this aspect.

Pag.22, line 32: "exhibits" instead of the plural.

Finally, a general comment about the plots in the different Figure. They are good and clear but the quality of the figures must be improved for the printing. Supplementary material is quite useful for the reader.

---

## Referee Comment (RC2) · Anonymous Referee #2 · 23 Nov 2018

The manuscript examines the completeness of newly released IGRA V2 humidity data. It is useful to have such a documentation to help users decide whether IGRA V2 has enough data for their own research before putting more efforts into downloading and analyzing the data. I would also like to appraise the authors for making their results (data) available and plan to update it on a two-year basis. I am little bit surprised on why the authors only look at humidity data, not including temperature and wind data. In "Introduction", the authors did not provide the rationale for only studying humidity observations, such as less humidity data than temperature data and degraded performance for hygrometers. Based on my evaluation, I think that the manuscript in current version needs some revision. Some of specific comments are listed below. Specific comments: 1. P1 L16: spell out GUAN. 2. P2 L30, add some of new references, such

as Dai et al. (2011). Dai, A., J. Wang, P. W. Thorne, D. E. Parker, L. Haimberger, and X. L. Wang, 2011: A new approach to homogenize daily radiosonde humidity data J. Climate, 24, 965-991. This applies to other places in "Introduction". 3. P3, L4: Durre et al. (2018, JTECH) should be used for the IGRA V2 reference. 4. P4, L29: "although most of the soundings did not reach beyond 700 hPa", I think that this is outdated. Most of modern radiosonde soundings can reach above 700 hPa. 5. P7, L2: China doesn't use goldbeater skin anymore. Again, this info is outdated. 6. P13, L3: 500 hPa threshold might be too high for high elevation sites. I think that it would be 300 hPa. 7. Fig. 4: Is decreasing of radiosondes stations in 35-65N due to reducing number of radiosonde launches in Europe given the budget constrain?

---

## Author Comment (AC2) · 8 Jan 2019

We would like to thank the Reviewer for the constructive comments, which we found helpful to improve the manuscript.

Please also note the supplement to this comment:
https://www.earth-syst-sci-data-discuss.net/essd-2018-95/essd-2018-95-AC2-supplement.pdf

---

## Author Comment (AC3) · 9 Jan 2019

The authors would like to draw attention to an error in their point-by-point reply to Anonymous Referee #1: on page 8, where it reads "by limiting RH", it should read "by lowering saturation vapor pressure".

Please also note the supplement to this comment:
https://www.earth-syst-sci-data-discuss.net/essd-2018-95/essd-2018-95-AC3-supplement.pdf

---

## Author Response (AR1)

07-02-2019

Dear David Carlson,

I am pleased to submit the revised version of essd-2018-95 'Completeness of radiosonde humidity observations based on the IGRA' for peer-review completion. In revising the manuscript, I and my coauthors have carefully considered the reviewers' comments. We must acknowledge that the revised version has benefited much from them.

Please note that the former authors' reply to referee comments (AC1, AC3) already contained a point-by-point response along with a detailed description of the authors' changes in manuscript. Therefore, the authors´ responses to the Referees are copied below without editing *save for the following amendments*:

→ regarding changes that were just summarized as "Planned changes in the manuscript", we now refer their accomplishment, denoted by "Actual changes" and highlighted in grey.

→ additional changes to fully address the reviewers' comments are denoted by "Further changes", also highlighted in grey.

→ a few minor corrections to the previous responses are highlighted in yellow.

Indications of page and line numbers, as well as figure numbers, refer, of course, to the discussion version of the manuscript, unless stated otherwise. The marked-up revised manuscript is provided after the responses to Referees.

Sincerely,

António P. Ferreira

**Authors' Response to Referee #1 Comments on**

"Completeness of radiosonde humidity observations based on the IGRA" by António P. Ferreira, Raquel Nieto, Luis Gimeno

*Referee comments (highlighted in blue) are copied before the authors' responses.*

The manuscript by Ferreira et al. presents an analytical description and a rationalized statistical analysis of the radiosounding data archive available through the NOAA-IGRA initiative. The authors report an extensive and precious description of a huge number of information about the changes occurred in the number, geographical density and type of radiosoudings and payloads used around the globe since the beginning of last century to present for the measurement of atmospheric humidity. The authors focus they study on humidity measurements and they investigate the datasets in a critical way to show if, in term of continuity and coverage, the available data may support climate studies, though they do not assess this aspect on a scientific sound basis. The paper is well written and curated and the provided analysis and it update expected on a biannual basis may support many activities currently ongoing, not ultimately the Copernicus Climate Change Service (C3S). I am in favor of the manuscript publication. Nevertheless, I have to provide the authors with major revisions, because to my opinion a few concepts must be clarified and modified in the text of the manuscript, though this does not affect my positive view on the manuscript itself. Below I enclose my general comment and a bunch of minor issues to solve.

**General comments**

1. In the manuscript, starting form the abstract, it is mentioned that the IGRA V2 datasets is investigated according to various completeness criteria in order to show if the length of the available data records is sufficient for the purposed to estimate climate variability trends. The authors mainly refer to the temporal continuity of the observation and not the spatial gaps which are anyhow investigated in the statistics. Despite this purpose, the authors clarify in the conclusion that their work can only be supportive of climate studies or works related to the climate homogenization of the time series, and nor they do not discuss on a quantitative basis what is the length of data records required for climate studies neither they refer to past literature to asses this aspect. Assuming that the authors are not interested in the going ahead in the assessment of the effect of gaps in the data records on the estimation of climate signals (which is well beyond the scope of the paper), I'd ask them to change the tone of the manuscript, where needed, to better clarify the scope of the manuscript.

The abstract states the paper´s purpose: "The sounding data compiled in the Integrated Global Radiosonde (…) are examined here until the end of 2016, aiming to describe the completeness of humidity observations from radiosondes in different times and locations" [P10, L10-12], further expressing the expectation that "the derived metadata will help climate and environmental scientists to find the most appropriate radiosonde data for humidity studies by selecting upper-air stations, observing years or individual soundings according to various completeness criteria – even if differences in instrumentation and observing practices require extra attention" [P1, L25-27]. Similarly, the introductory section states: "The purpose of this paper is to study the completeness of humidity observations collected in IGRA according to various needs – number and latitudinal distribution of observing stations, fraction of observing days in a year, resolution and range of vertical levels, length and continuity of the time-series, minimal sampling between the surface and the 500-hPa level – aiming to facilitate the use of radiosonde humidity data by atmospheric and environmental scientists." [P3, L14]. In short, our main purpose is not exactly to show if the available data may support climate studies. Still, the interest

of our work to climate studies is stressed in several parts of the manuscript. The next comment summarizes how climate studies are addressed in the body of the text.

The use of radiosonde data for climate studies is mentioned in the Introduction (P2, L25-29) because radiosonde archives provide the longest atmospheric humidity records available. While the notion of climatic variability may refer to different time scales, it usually involves a few decades (to define a reference mean state). Concerning climatic trends, the longer and stable the time series, the better. Therefore, Section 2.3.4 draws attention to the importance of accessing large interruptions in long term humidity time series, by identifying the time series without gap years and measuring the continuity of data within each year. The availability of multidecadal humidity time series of upper-air humidity is illustrated in Figures 9 and 10 of Section 3.4, using different completeness requirements for time series without gap years (concerning the fraction of observing days in a year, the maximum size of gap days and the vertical sampling in the lower troposphere). As explained in that section, "There is no simple answer to the question of since when we have enough data to (in theory) perform climate studies from radiosonde humidity data: it depends on the strictness of completeness criteria." [P20, L4-6]. Figs. 10 11 simply indicates how much the availability of long term humidity time series can depend on specific data completeness requirements, besides the length of the period of record of course. We think that the summary results in the concluding Sect. 6 (last bullet paragraph on P23, L1-8) are sufficiently objective in this respect. The paper ends with a suggestion on the selection of time series for studying the homogenization of humidity data.

Regarding the availability of data for climate studies, the abstract reads: "For illustration, the study presents a global picture of the completeness of radiosonde humidity observations over the years, including their latitudinal coverage. This overview shows that the number of radiosonde stations having a long enough record length for studies on the climatic variability and trends of humidity-related quantities depends critically on the temporal continuity, regularity and vertical sampling of the humidity time-series." [P1, L22-25]. We understand that the latter sentence can give the wrong impression that our paper quantifies the completeness *requirements* for performing climate studies on humidity-related quantities. This misperception can be briefly solved. We further understand that the Short Summary appearing on the ESSDD publication can be slightly misleading about the scope of the paper. This can be changed too.

**Changes in the Abstract:**

**Paragraph on P1, L23-25:**
This overview shows that the number of radiosonde stations having a long enough record length for studies on the climatic variability and trends of humidity-related quantities depends critically on the temporal continuity, regularity and vertical sampling of the humidity time-series.
**It will be changed as follows:**
This overview indicates that the number of radiosonde stations having a record length potentially long enough (multidecadal) for climate studies involving humidity-related quantities depends not only on the temporal range, but also on the continuity, regularity and vertical sampling of the humidity time-series.

**Amendment to the Short Summary:**

**Where it reads:**
The work shows that the potential use of radiosonde humidity data for climate studies depends on the continuity, regularity and vertical sampling of long time series.

**It should read:**
The work illustrates how humidity data potentially available for climate studies depend on the length, continuity, regularity and vertical sampling of time series.

Further changes:

Following the rearrange of Sect. 3 [see bellow response to point 3 of general comments], Sect. 3.4 discussing the availability of multidecadal humidity time series was retitled and numbered as Sect. 3.5. The related Fig. 10 was renumbered as Fig. 11.

**2. The authors make use of IGRA data V2, which is the most updated version of IGRA which embed several improvements compared to the previous data version (V1). Nevertheless, in my personal researches I had a chance to find many bugs in the IGRA V2 where many data present in the V1 are missing and this is not dependent on the extended quality check applied within the IGRA V2. I had also a chance to report this bug to Imke Durré (PI of IGRA). I got similar feedbacks from other EU and US colleagues during discussion at various workshops. The station of Lindenberg in Germany (WMO index=10393), so accurately described in its past history in the manuscript, is one those affected by this issue (at least until a few weeks ago, my last access to IGRA). Therefore, I am wondering whether a comparison between IGRA V1 and V2 has been carried out and if gaps have been found and fixed somehow in the datasets investigated in the manuscript.**

Such apparent data loss during the processing of data sources in IGRA 2 is concerning. We have used IGRA 2 as is, because of the extended data coverage and improvements of quality assurance on humidity data. On the positive side, any future corrections to sounding data in IGRA will translate into the updated versions of the meta-dataset introduced in the present paper.

**3. The description of statistical analysis of IGRA V2 data is very extensive way, providing several details and a long description of each figure and table. When reading, I have been very interested by the content, though sometimes the reader may get tired by the way the manuscript is written. In a way similar to the conclusions, I'd suggest to the authors to change the style of their writing and privilege a description in "bullets" to describe the results whenever possible. this will help also to clarify the text itself. For example, for the relative humidity observations, At pag.5, it is reported that the average number of non-standard levels in weather-balloon sounding reports increased from about zero by 1945 to about 30 by 2000, but later in the text it is said that RH observation were already available since the 1930, and again later on (pag. 18) it is stated that RH are becomes more abundant since 1949. Though all of this information are exact the reader may get confused and their comparability and usefulness could be limited if the statical analysis is not described in a more schematic way.**

We believe that the largest sections (Sect. 2 and 3) will become much easy to read once the tables and figures are placed besides the text. Please note that paragraphs were used extensively to organize ideas around themes. Also, number bullets were used four times before the concluding section. Data analysis is schematized in the following way: Sect. 2.1, 2.2 and 2.3 describe, respectively, the IGRA sounding data set used in our work, the identification of mostly-radiosonde data series, and the main data analysis needed to compute the parameters used to describe 'completeness of humidity observations'. Figures 1 and 2 are auxiliary to Sect. 2.1 and 2.3. Each subsection of Sect 2.3 ('Analysis of humidity data') introduces the corresponding completeness parameters and ends

with a description of the secondary statistical analysis used later in Sect. 3 ('Overview on the completeness of radiosonde humidity observations' – in which the several completeness measures are graphically illustrated by Figs. 3-10. The introductory part of section 3 and several parts of subsections 3.1 to 3.4 recall the data and methods used to draw the related figures, each time they are first mentioned in the text.

On P8 (not P18), L2-4, what we have said is that *until 1969 humidity was reported only in the form of RH* (not that RH data became more abundant since 1949), as documented in IGRA's Readme file. This is unrelated with the fact that the earliest upper-air humidity observations are from 1930 [see P5, L28; P9, L16]; or that in 1945 the average number of significant levels in weather-balloon reports was nearly zero [see P5, L15] – meaning that standard pressure levels were virtually the only ones reported by then. None of this information has to do with the data analysis explained later in Sect. 2.3.

Further changes:

Concerning Sects. 2 and 3, we hope that the following changes bring more clarity to the statistical analysis. First, the material of Sect. 2.3 and Sect. 3 – which now both have five subsections instead of four – was slightly rearranged to separate issues of temporal completeness from other issues. Second, the introductory part of Sect. 2.3 and the numbering of figures in Sect. 3 were modified accordingly.

On P5 it is clarified that the number of vertical levels in Durre (2006) refers to IGRA v1

**4. The use of the metric present in the Eq.1, "zonal coverage index" at pag 11 is not clear to me. Have been it used in the past and its adde value with respect to other metrics was shown? What's the added value with respect to a station density per 1000 km, for example? Has the ocean surface been excluded from the global surface, given that this can be calculate more clearly for fixed stations? There are many concerns to me on the use of this index, which requires clarification from the authors. Personally, I think that the user can make use of a much simpler index or statistics (a few of these are used later on in the manuscript by the authors themselves) to show if a zonal belt are under and over- represented. This is also depending on the different atmospheric circulation occurring in the different zonal belts, so I am wondering what's the usefulness of adopting this metric. I ask the authors to clarify in the manuscript of the added value due to the use of this metric.**

The "zonal coverage index" index defined by Eq. 1 was intended to compare the fraction of stations among in different climatic zones, normalized by the surface area fraction. To our knowledge it was never used before. [Note: by mistake, the modulus term in Eq. (1) is not raised to -1, as it should be]. Although that index was used to construct Fig. 4b, note that it does not form part of the parameters represented in the dataset introduced in the paper.

We agree that the average station density in different latitude belts – or alternatively the average spacing of stations – would give a more direct information about the data coverage. Moreover, the plot of Fig. 4a representing the fraction of humidity-reporting stations at different latitude bands is not essential, since Fig. 3 (which uses yearly absolute numbers) gives a detailed picture of the same distribution. Therefore, the index

in question can be replaced by a more familiar metrics, as suggested in the Referee comment, without affecting the dataset introduced in the paper.

Planned changes in the manuscript:

**Section 2.3.1 will be fully revised,** with the purpose of replacing Eq (1) by a parameter describing the average separation between adjacent stations in a given zonal band. Separate calculations will be presented for land and ocean areas. The **plots in Figure 4a and 4b** will be replaced to represent both regions. The **comments to Fig. 4 in Sect. 3.1** and **the first bullet paragraph of Sect. 6** will be modified accordingly (but retaining the essential about the relative amount of stations in each climatic zone as shown in Fig. 3).

Actual changes:

The full revision of Sect. 2.3.1, as sketched above, involved the addition of a new figure in Sect. 3.1 showing the locations of IGRA-RS stations: this is the newly Fig. 3 [see p. 43 of the marked-up manuscript]. Consequently, former Fig. 3 was renumbered as Fig. 4. In addition, the former Fig. 4 was redesigned according to the major change in Sect. 2.3.1 and the replacement of Eq. (1), being now Fig. 5. Figure 5 was renumbered as 6; Fig. 9 was moved and is now Fig. 7; Figures 6, 7, 8, 10 were thus renumbered as 7, 8, 9, 11.

**NB –** The rearrange of the subsections of Sect. 2.3 [see response to point 3 above] implied a similar rearrange in the subsections of Sect. 3 [idem]as well as in the bullet paragraph of Sect. 6.

**5. The average vertical resolution could be useful information, but to my opinion, thinking about different applications it could be useful to have a statistic of the available level for the different regions of the atmosphere: Planetary Boundary Layer, Free Troposhere, Upper Troposphere/Lower Stratosphere (UT/LS), Upper Stratosphere. This classification may have a stronger impact to orienting users' application in the selection of the available data, e.g for trends calculation.**

Figure 6 only shows the distribution of vertical resolution among humidity observations over time. The dataset introduced in the discussion paper, however, contains four related parameters for each IGRA-RS station:

  i.   Annual mean vertical resolution of yearly humidity obs. from each station
  ii.  Annual mean vertical resolution of yearly Sfc-to-500hPa humidity obs. from each station
  iii. Vertical resolution of individual humidity obs. from each station
  iv.  Vertical resolution of individual Sfc-to-500hPa humidity obs. from each station

where 'resolution' stands for geometric mean resolution. Parameters (i) and (iii) refer to reported humidity data up to the highest measuring level. Parameters (ii) and (iv) refer to humidity data between the surface and the 500-hPa level, requiring humidity data at the surface and at a minimum of upper-air levels to estimate precipitable water vapor. Therefore, the dataset accompanying the paper provides specific information on the vertical resolution at the lower troposphere.

Moreover, the dataset also provides information on the highest measuring level:

    v. Annual geometric mean pressure of the highest level with humidity data at each station
   vi. Pressure of the highest level with humidity data in individual obs. at each station
  vii. Altitude of the highest level with humidity data in individual obs. at each station

where (vii) is obviously only calculated if humidity is not missing at the surface level. Such information allows to select the soundings with humidity observations reaching the UT and the LS, as well as the stations and years that normally have observations in the UT/LS, even if the vertical resolution at those regions is not detailed.

Considering the parameters describing temporal completes of humidity observations [periods of record for humidity; fraction of observing days in a year; largest number of consecutive missing days], analysing vertical resolution for each separate atmospheric layer (PBL, lower/middle troposphere, UT, LS) – each one possessing its own temporal completeness – would imply extending the work beyond what might be expected from a first approach. Undoubtedly, a recommendation for future research.

**6. Please check the use of the term "error" throughout the manuscript and replace it with "uncertainties" where more appropriated.**

The term 'error' is used only twice in the manuscript: "Concerning humidity, radiosonde-based climatic studies are for now confined to the lower and middle troposphere, because of the large errors and insufficient data in the upper troposphere and lower stratosphere." [P2, L25-26] "For the purpose, it suffices to neglect moisture in the hypsometric equation; given that the virtual temperature is typically within 4 K above the actual temperature, the error amounts to less than 1 %." [P12, L28-29]

*In the first case*, we refer to measuring errors related to the poor accuracy of humidity sensors in the upper troposphere (UT) and lower stratosphere (LS). Given that different sensors have different accuracies under the same conditions, equal instruments behave differently in different conditions, and climate studies use data averaging from many observations, we must accept that the expression 'uncertainty' is more appropriate. Moreover, missing data arising from varying observation practices (low RH reporting, low-temperature cutoff) introduce uncertainties in the time series and their averages. This affect particularly the measurements above 500 hPa. (On passing, we should refer the issue of observation practices right away on P2, L22.) However, systematic biases in the UT are well known for certain radiosonde models. Thus, to include both situations, we would say 'uncertainty of measurements and biases'. In the LS, most measurements show a wet RH bias which is very large in relative terms. We think it's also adequate to add a few citations regarding accuracy and uncertainty of humidity measurements in the UT and LS.

*In the second case*, "error" is the error in calculating geopotential height; this will be clarified

Changes in the manuscript:

**P2, L22**
**Where it reads**

instrument changes and sampling differences
**It will read**
instrument changes, reporting practices and sampling differences

**P2, L25-26**
**Where it reads:**
the large errors and insufficient data in the upper troposphere and lower stratosphere
**It will read:**
the large uncertainty of measurements and biases in the upper troposphere and lower stratosphere (Elliot and Gaffen, 1991; Soden and Lazante, 1996; Wang et al., 2003) and the extremely large relative biases and insufficient data in the lower stratosphere (Miloshevich et al., 2006; Nash et al. 2011)

**P12, L29**
**Where it reads:**
the error amounts to
**It will read:**
the error in calculating geopotential height amounts to

Additions to the reference list:

Wang, J., D. J. Carlson, D. B. Parsons, T. F. Hock, D. Lauritsen, H. L. Cole, K. Beierle, and E. Chamberlain (2003), Performance of operational radiosonde humidity sensors in direct comparison with a chilled mirror dew-point hygrometer and its climate implication, Geophys. Res. Lett., 30, 1860, doi:10.1029/2003GL016985, 16.

Soden, Brian & Lanzante, John. (1996). An Assessment of Satellite and Radiosonde Climatologies of Upper-Tropospheric Water Vapor. J. Climate. 9. 1235-1250. 10.1175/1520-0442(1996)009<1235:AAOSAR>2.0.CO;2.

Nash, J., T. Oakley, H. Vömel, and L. I. Wei, 2011: WMO intercomparison of high quality radiosonde systems (Yangjiang, China 12 June–3 August 2010). WMO Instruments and Observing Methods Rep. 107, 238 pp. [Available at www.wmo.int]

**Specific comments**

**Pag.2, lines 10-12: the authors could mention that recent reanalysis products, for example ERA5 from ECWMF, will improve the 4 times daily frequency of the products up to hourly.**

Perhaps, we should rather refer some ECMWF reanalysis as it is now, with a different period of record. This can be amended.

Actual change:

**P2, L12**
**Where it reads:**
up to 4 times daily from 1948 (as in NCEP/NCAR products) to present time
**It should read:**
up to 4 times daily from a beginning year (e.g.: 1948 in NCEP/NCAR Reanalysis 1; 1979 in NCEP/NCAR Reanalysis 2 and ECWMF's ERA Interim) to present time

**Pag.2, line 13: replace "not to mention : : :.." with "with not negligible : : :.."**
Agree. Changed to "with significant".

**Pag.2, lines 17-19: uncertainties due to balloon drifting and observation time are considered negligible citing the publication by Kitchen (1989) as well as the radiosonde profile accuracy. To my opinion these cannot be considered minor, and anyhow if minor or not this is depending on the considered application. First of all there more recent papers by Seidel et al. (2011) dealing with radiosonde balloon drifting. Estimates of elapsed time from balloon launch to various pressure levels, due to vertical balloon rise, have median values increasing from about 5 min at 850 hPa to about 1.7 h at 10 hPa, with ranges of about 20% of median values. Observed elapsed times always exceed those estimated using assumed 5 or 6 m/s rise rates. Regarding the data data accuracy, if we are referring to the ensemble of effect which may alter the sensors' optimal measurement conditions, like the solar radiation effect to the effect of a sensor time-lag, these have been better quantified for the more recent radiosonde types and are quite relevant for any kind of climate application (see for example the GRUAN quantification by Dirksen et al. 2014). This is also the reason why many scientific groups have developed homogenized dataset of radiosonde data for climate application. For there reason above, I'd reformulate these line in the manuscript and I'd provide more updated references.**

We understand that uncertainties related to the observation time and balloon drift are currently recognized to be relevant for data assimilation in models (both in numerical weather prediction and reanalysis), besides impacting forecast verification, satellite validation and climatic statistics, specially at high levels. Kitchen (1989) pointed out its importance for weather forecasting. We appreciate that the referee comment calls for more recent literature to substantiate the subject.

We remove "minor" and improve the whole sentence, adding several citations and new references to make a balance between the different problems mentioned. Dirksen et al. (2014) will be cited on P 6, L30.

Changes in the manuscript:

**P2, L17-19**
**Where it reads:**
despite sampling differences among stations and over time, minor uncertainties related to observation time and balloon drift (Kitchen, 1989), and differences in data accuracy – which depends on humidity sensors and varies with measured conditions (e.g., Nash, 2002).
**It will read:**
despite geographical-temporal sampling differences (Wallis, 1998), uncertainties related to observation time and balloon drift (Kitchen, 1989; McGrath et al., 2006; Seidel et al., 2011; Laroche and Sarrazin, 2013), differences in vertical coverage and data gaps related to reporting practices of humidity (Dai et al. 2011 and references thein) and differences in humidity data accuracy – which depends on humidity sensors and varies with measured conditions (e.g., Nash, 2002; Sappuci et al., 2005; Moradi et al., 2013; Dirksen et al., 2014).

**P6, L30**
**Insert reference:** (Dirksen et al., (2014) and references therein)

Additions to the list of references:

Seidel, D. J., B. Sun, M. Pettey, and A. Reale, 2011: Global radiosonde balloon drift statistics. J. Geophys. Res., 116, D07102, Doi: 10.1029/2010JD014891.

Laroche, S. and R. Sarrazin, 2013: Impact of Radiosonde Balloon Drift on Numerical Weather Prediction and Verification. Wea. Forecasting, 28, 772–782, Doi: 10.1175/WAF-D-12-00114.1

McGrath, R., T. Semmler, C. Sweeney, and S. Wang, 2006: Impact of Balloon Drift Errors in Radiosonde Data on Climate Statistics. J. Climate, 19, 3430–3442, https://doi.org/10.1175/JCLI3804.1

Sapucci, L.F., L.A. Machado, R.B. da Silveira, G. Fisch, and J.F. Monico, 2005: Analysis of Relative Humidity Sensors at the WMO Radiosonde Intercomparison Experiment in Brazil, J. Atmos. Oceanic Technol., 22, 664–678, doi:10.1175/JTECH1754.1

Wallis, T.W., 1998: A Subset of Core Stations from the Comprehensive Aerological Reference Dataset (CARDS). J. Climate, 11, 272–282, doi: 10.1175/1520-0442(1998)011<0272:ASOCSF>2.0.CO;2

 ← IT WAS ALREADY CITED

**Pag.2, line 26: please replace "errors" with uncertainties and then add also that the radiosonde sensors may have a limited sensitive to ppm water vapor concentrations in the UT/LS as one of the main reason because humidity data above 300hPa are unreliable.**

Regarding the word replacement, please refer to our response to point 6 of the general comments. The details about limitations of humidity sensors were given in Sect. 1.3. However, we understand that the limited response to low water vapor pressures should be directly mentioned in Sect 1.3.

Change in the manuscript:

**P5, L29**
**Where it reads:**
humidity has been always difficult to measure in very cold or dry air.
**It will read:**
humidity has been always difficult to measure in very cold or dry air due to the poor response of many instruments at very small vapor concentrations (by lowering saturation vapor pressure, cold temperatures are associated with low water vapor pressures).

**Pag.2, line 32: please change "lag" time "time-lag".**
Correct.

**Pag.3, lines 19: "the remainder of this section".**
Correct.

**Pag.3, lines 21: put "available" in between of "levels" and "in radiosonde"**
Fine.

**Pag3, line 26: ": : :.relevant for the study". Study of what?**

It means it is relevant to our study, referring to the subset of IGRA without pilot-balloon stations (as explained in lines 23-25 right above). See rephrasing.

Change in the manuscript:

**P23, L26**
**Where it reads:**
from each IGRA station relevant to the study
**It will read:**
from each relevant IGRA station

**Pag3, line 26: " please change "indicated" with "reported"**
Okay (referring to P3, L28).

**Pag4, lines 7-10: Please put a reference related to the importance of data continuity for climate studies (trends, annual cycles).**

Previous literature regarding temporal inhomogeneities in radiosonde time series have mainly focused in instrument changes (responsible for the largest time-varying biases), paying less attention to temporal sampling. Nevertheless, this issue has been assumed relevant to climate studies, as detailed next.

*Concerning standards for long-term monitoring of the climate system*: Karl et al. (1995) puts the *continuity* and *frequency* of in-situ and other observations – as well as the maintenance of local observations with a long and uninterrupted record – among the "critical issues for long-term climate monitoring".

*Concerning temporal homogenization of radiosonde time series for determining long-term temperature and humidity trends*: Lanzante et al. (2003), mentioned the interest of "counts of numbers of observations per month as a function of time and by level; these aid in finding sampling biases or less reliable time periods". Moreover, they have computed separate monthly means at 0000 and 1200 UT from 1947-98 CARDS data with the requirement of at least 16 valid values per month. [Since this paper focus on the detection of changing points in temperature time series related to instrument changes and reporting practices, it will also be cited in Sect.1; see P2, L20-23]. McCarthy et al. (2009) [already cited on P7, L8] addresses the problem of "important sampling biases in the raw humidity data, from missing dry observations and missing cold observations"; they also require at least 15 days of observations within a month to calculate monthly mean temperatures.

*Concerning the temporal completeness requirements in climate studies on humidity or integrated water vapor*: Gaffen et al. (1991), on studying spatial-temporal variability of global tropospheric moisture, discuss temporal sampling, arguing that a minimum of three observations per month is required to obtain an estimate of the monthly mean specific humidity that falls within the 0.1 confidence bands. Using a 1978-85 dataset from 119 stations, they noted that almost 5% of the station months did not meet that

criterion. Gaffen and Elliot (1992) [already cited in the manuscript] selected radiosonde stations with at least one observation per day to accurately estimate the seasonal cycle of RH at different locations of the globe. Zhai and Eskridge (1997) used Gaffen´s criterion to study changes in PW over China, further rejecting stations with more than 3 years of data missing at the same level. Ross and Elliot (1996) [already cited in the manuscript], on studying water vapor trends over North America have required at least two months to estimate seasonal anomalies and 10 months to annual anomalies, a month being considering as missing if less than 10 observations during the month were missing.

The manuscript changes to the lines in question consists of rephrasing and inserting the proper references.

**Changes in the manuscript:**

**P4, L9-10**
**Where it reads:**
in addition, the regularity and continuity of humidity profiles are important to trend analysis and to detail annual cycles  ← THIS WAS NOT ACTUALLY IN THE MS
**It will read:**
furthermore, the period of record and the regularity and continuity of radiosonde data is a relevant issue for long-term monitoring of the climate system (Karl et al., 1995), as exemplified by temporal sampling requirements used in trend and seasonal analysis of temperature, humidity and integrated water vapor (Gaffen et al., 1991; Gaffen and Elliot, 1992; Karl et al., 1995; Ross and Elliot, 1996; Zhai and Eskridge 1997, Lanzante et al. 2003; McCarthy et al., 2009)

**P2, L23 | Add citation:**
Lanzante et al. (2003)

**Additions to the list of references:**

Gaffen, D.J., T.P. Barnett, and W.P. Elliott, 1991: Space and Time Scales of Global Tropospheric Moisture. J. Climate, 4, 989–1008, https://doi.org/10.1175/1520-0442(1991)004<0989:SATSOG>2.0.CO;2

Karl, T.F., Derr, V.E., Easterling, D.R., Folland, C.K., Hofmann, D.J., Levitus, S., Nicholls, N., Parker, D.E., & Withee, G.W. (1995). Critical issues for long-term climate monitoring. Climatic Change, 31(2/4), 185-221.. Doi:10.1007/BF01095146

Zhai, P. and R.E. Eskridge, 1997: Atmospheric Water Vapor over China. J. Climate, 10, 2643–2652, https://doi.org/10.1175/1520-0442(1997)010<2643:AWVOC>2.0.CO;2

Lanzante, J.R., S.A. Klein, and D.J. Seidel, 2003: Temporal Homogenization of Monthly Radiosonde Temperature Data. Part I: Methodology. J. Climate, 16, 224–240, doi:10.1175/1520-0442(2003)016<0224:THOMRT>2.0.CO;2

**Pag4, Line 10-12: I tend to disagree with the introductory sentence while I like the authors approach in the manuscript; therefore, I think in this sentence you must report which is needed to investigate the simultaneous spatial and time sub-sampling on the data whatever challenging this might be in the practice.**
We think that advancing what is needed to investigate the simultaneous spatial and time sub-sampling on the data is beyond the scope of our work. Furthermore, it depends on the application of radiosonde data. Anyway, knowing the content of a comprehensive

radiosonde data archive such as IGRA 2 represents a first step. The present paper might be useful in two ways: first, it gives a general picture of the spatial-temporal distribution of in-situ humidity observations worldwide. Second, it provides a tool (dataset) that helps to select either humidity time-series or individual observations based on their temporal and vertical completeness of humidity (*i.e. of simultaneous observations of pressure, temperature and humidity*). It is left to the user to find the data coverage and spatial continuity from the geographical coordinates of stations [please note the final recommendation on P23, L20-23].

However, we agree that the sentence in question can be improved in the following way. First, a station's period of record cannot be separated from its temporal completeness. Secondly, it is appropriate to cite again the paper of Wallis (1998)* since it represents a major effort towards a selection of radiosonde stations for trend studies on a regional or global scale.

(*) Wallis, T.W., 1998: A Subset of Core Stations from the Comprehensive Aerological Reference Dataset (CARDS). J. Climate, 11, 272–282, doi: 10.1175/1520-0442(1998)011<0272:ASOCSF>2.0.CO;2

Changes in the manuscript:

**P4, L10-12**
**Where it reads:**
Although the vertical and temporal completeness of station-based humidity time-series can be treated separately from the issues of spatial coverage and record length of stations, it seems to us that studying the completeness of observations in a global, historical data set of radiosonde observations should address these issues too.
**It will read:**
Although the vertical and temporal completeness of station-based humidity time-series can be treated separately from the issue of  geographical coverage, studying the completeness of observations in a global, historical data set of radiosonde observations should address that issue too. This is particularly true  concerning the subsampling of radiosonde stations for studies of atmospheric temperature or water vapor trends on a regional or global scale (Wallis, 1997).

**Pag. line 24: pressure in not measured anymore in the most recent radiosonde types (e.g. RS-41 operation since a couple of years); please for completeness you may mention this.**

Right, not only in the most recent GPS radiosondes but also in some Russian radiosonde systems which used ground radar and a radiosonde without pressure sensor. Being an exception to common radiosondes, this detail deserves a footnote. As to a reference to modern GPS radiosondes (with or without a pressure sensor), Nash et al. (2011) will be also cited in the former footnote 1 on page 30 [as well on page 2 – please see response to general comment 6]

Changes in the manuscript:

**P4, L22-24**
**Where it reads:**
In radiosonde soundings, temperature, relative humidity (eventually dewpoint depression too) and wind speed and direction are measured together with atmospheric pressure; geopotential height is indirectly measured from hypsometric calculations but may be missing in radiosonde reports.
**It will read:**

In radiosonde soundings, temperature, relative humidity (and/or dewpoint depression) and wind speed and direction are measured together with atmospheric pressure, while geopotential height is indirectly measured from hypsometric calculations[1] (but may be missing in radiosonde reports).
* * *
([1]) Except in some Soviet/Russian radiosonde-radar systems and the last generation of GPS radiosondes – in which pressure is deduced from the (radar or GPS, respectively) profile of geometric height and the radiosonde profiles of temperature and humidity (Zaitseva, 1993; Nash et al., 2011)

**P7, L30 [former footnote 1, now footnote 2 according to the above change]**
**Where it reads:**
(Dabberdt et al. 2002)
**It will read:**
(Dabberdt et al. 2002; Nash et al., 2011)

Additions to the list of references:

Zaitseva, N. A., 1993: Historical developments in radiosonde systems in the former Soviet Union. Bull. Amer. Meteor. Soc., 74, 1893–1900, doi:10.1175/1520-0477(1993)074<1893:HDIRSI>2.0.CO;2.

Nash, J., T. Oakley, H. Vömel, and L. I. Wei, 2011: WMO intercomparison of high quality radiosonde systems (Yangjiang, China 12 June–3 August 2010). WMO Instruments and Observing Methods Rep. 107, 238 pp. [Available at www.wmo.int]

**Pag.5, line 17-19: this sentence could be a good opportunity to claim for the importance of having high resolution measurements and, therefore, more levels available in the radiosounding report. This is in line with the high resolution BUFR files already flowing to the Met services from more than 100 station worldwide.**

The current migration from TEMP to high-resolution BUFR reports will be referred in Sect. 2.1; the fact that BUFR data are not currently available in open archives (e.g. IGRA) should be noted.

Planned change to the manuscript:

**P5, L19:** Insert sentence about the migration to BUFR messages sent to the GTS, with a reference to Ingleby et al. (2016).

Actual change:

**P5, L23 – Inserted text:**
The current migration of RAOB reports from alphanumeric (TEMP) to the binary universal form for the representation of meteorological data (BUFR), together with the conversion of radiosondes to generate BUFR messages, allows the transmission of high-resolution data (2 to 10 s sampling rate, i.e., ~ 5 to 50 m resolution in a typical ascent) along with the balloon drift position, the observation time for each level and other metadata (Ingleby et al., 2016). Currently, 20% of the radiosonde stations send high-resolution BUFR reports through the Global Telecommunication System (GTS), many coming from Europe; however, such data are not yet available in an open archive.

Addition to list of references:

Ingleby, B., P. Pauley, A. Kats, J. Ator, D. Keyser, A. Doerenbecher, E. Fucile, J. Hasegawa, E. Toyoda, T. Kleinert, W. Qu, J. St. James, W. Tennant, and R. Weedon, 2016: Progress toward High-Resolution, Real-Time Radiosonde Reports. Bull. Amer. Meteor. Soc., 97, 2149–2161, doi:10.1175/BAMS-D-15-00169.1

**Pag. 7, line 13: "anything but uniform" can be modified "quite heterogeneous".**

Correct. After all, there is substantial uniformity as to the launching time and STD levels.

**Pag. 7, lines 22-24: these sentence is the "official" IGRA description, but going through the data the authors may realize that among the 2761 stations, many of them are "nearsurface" stations and not radiosounding station. All the reports are empty (-9999) for many station. These must be clarified and the reported number estiamted in a more precise way.**

The NOAA's IGRA web page says, "over 2700", without giving a precise number, since it may increase as new sources are added. Maybe what is lacking here is the time when the data were downloaded (or the date of the IGRA stations list, since the data files were downloaded using a script based on that list). Regarding "near-surface stations", we have checked on this and, based on the dataset complementary to the paper, it seems there are about 120 stations (among the IGRA-RS subset given in Table S1) in which *the first one or two years of record* contains only surface data. This happens mostly during  1946-49. This can be mentioned in Sect. 2.2.

Planned changes in the manuscript:

**P7, L22:** Indication of the time when the data (or the stations list) was downloaded.
**P10, L18:** Mention to only-surface data at the beginning of record at some stations.

Actual changes:

**P7, L16 – Inserted word:**
IGRA 2 dataset → IGRA2 main dataset
**P7, L16 – Deleted/moved text:**
comprising over 45 million soundings

**P7, L22-23**
**Where it reads:**
taken at 2761 globally and temporally distributed stations (2662 fixed and 99 mobile)
**It should read:**
taken at over 2700 globally and temporally distributed stations

**P7, L24 – Inserted text:**
comprising over 45 million soundings from 2761 (2662 fixed and 99 mobile) stations [based on data accessed in September 2017].

**P10, L18 – Inserted text:**
Note: in the data from around 120 land stations the early years of record for humidity (normally 2 to 3 years), contains only surface or near-surface data; this happens in about 100 stations of the former Soviet Union, mostly during the years 1946-49.

**Pag. 7, line 16: Is the reported typical average accuracy in the troposphere only related to the most recent radiosonde types? Please clarify**

(Referring to P8, L16) Recent radiosondes achieve an accuracy of 2% RH, while very old radiosondes have worse accuracy (10%). 5% is roughly the accuracy accepted for the

troposphere, the layer that presents the most favourable conditions for measurements. The remainder of the sentence specifies that accuracy can in fact deviate significantly from this round number. At extremely low temperatures is not uncommon to see errors of 15%. Accuracy is compromised in high resolution profiles if the response time turns too long. And then there is precision for RH or DPD. In 1991, Elliot and Gaffen reported that by that time radiosondes presented a precision of about 3.5% RH. But this depends on environment conditions too. Moreover, TEMP reports may have different restrictions for precision over time. For all these reasons, maybe it is wise to withdraw the "typical accuracy", and shorten the whole sentence keeping the main idea (Note: the related references were already cited in other parts, mainly in Sect. 1.3)

**Changes in the manuscript:**

**P8, L15-18**
**Where it reads:**
Note that RH measurements from radiosondes have a typical absolute accuracy of ∼ 5 % on average tropospheric conditions. However, accuracy varies substantially as a function of RH and temperature, degrading in dry or cold conditions to a greater or lesser extent depending on the radiosonde type (Nash, 2002; Miloshevich et al., 2006; Nash, 2015).
**It will read:**
Note that the precision and accuracy of RH and DPD data varies substantially as a function of RH and temperature, degrading in dry or cold conditions to a greater or lesser extent depending on the radiosonde type.

**Pag.11, line 1: please put a descriptive reference for GRUAN, I suggest Bodeker et al., 2016 BAMS.**
**Pag.11 line 3: It is not true that all of the GRUAN sites are transmitting data to the GTS. A few sites are still working to establish this data flow. This is also connected to the sentence at line 9, reporting the fact that not all GRUAN station are present within IGRA.**
**Pag.11, line 6: The added value of GRUAN is not only to provide data the quality of which should be "above the average", but to provide traceable uncertainties and a fully disclosed data processing described in peer-reviewed literature. Please add more details to show the real added value coming from GRUAN**

We will add the suggested reference, clarify that by "All GRUAN sites" we mean the ones present in IGRA, and add something about the real value of GRUAN.

**Changes in the manuscript:**

**P11, L1-3**
**Where it reads:**
Moreover, the IGRA-RS contains 16 stations that form part of the GCOS Reference Upper-Air Network (GRUAN), (half certified and half to be certified according to current GRUAN status), of which eight (half certified too) are also GUAN stations. All GRUAN sites report
**It will read:**
Moreover, the IGRA-RS contains 16 stations that form part of the GCOS Reference Upper-Air Network (GRUAN; Bodeker et al., 2016): half certified and half to be certified according to current GRUAN status, of which eight (half certified too) are also GUAN stations. Those specific GRUAN sites report

**P11, L4 – Insert text:**

GRUAN aims to serve as reference to other radiosonde networks, by providing long-term high-quality records of vertical profiles of selected essential climate variables, accompanied by traceable estimates of measurement uncertainties.

Addition to list of references:

G. E. Bodeker, S. Bojinski, D. Cimini, R. J. Dirksen, M. Haeffelin, J. W. Hannigan, D. F. Hurst, T. Leblanc, F. Madonna, M. Maturilli, A. C. Mikalsen, R. Philipona, T. Reale, D. J. Seidel, D. G. H. Tan, P. W. Thorne, H. Vömel, and J. Wang: Reference Upper-Air Observations for Climate: From Concept to Reality. Bull. Amer. Meteor. Soc., 97, 123–135, doi:10.1175/BAMS-D-14-00072.1, 2016.

**Pag.11, eq.1: see my general comments above.**

See our response to general comment 4.

**Pag.12, line 18: what's the meaning of the "relevant" in this sentence?**

It means of interest, i.e. a vertical level among the levels with non-missing humidity data. It is understood in the context, considering the preceding phrase in the whole sentence "*M* is the number of levels with humidity data above the lowest level", as well as the previous sentence which presents Eq. (1): "the mean vertical resolution of a single humidity sounding was defined by the geometric mean of $\{dzk\}$ for all levels with humidity data in the sounding profile"

**Pag.12, line 19: How did you choose the value of the constant temperature T0, please clarify in the text.**

The way Eq. (1) was written, the terms inside the product operator are non-dimensional. To prevent *underflow* in a calculation for high-resolution profiles and limited magnitude range for real numbers (because the logarithm terms are inferior to 1), $T_0$ should rather be chosen so that $RT_0/g \approx$ *average distance between vertical levels* (but this is precisely what we don't know).

Since we used double precision real numbers, and the number of levels with humidity data is not excessively large in IGRA, the result was insensible to the value used (250 K).

However, the term $T_0$ is not needed providing that Eq. (1) is reformulated in a way that (i) is mathematically equivalent and (ii) it can be safely implemented as is in a computer programme, without leading to underflow when the number of levels is too large. (Although we could alternatively define geometric mean by the anti-log of the arithmetic mean of log-transformed values, this is not needed.) In the revised and straightforward form, each individual term between curved brackets is raised to 1/*M* inside the product operator (instead of doing the product first and then the *M*th root, as usually seen in the formal definition of geometric mean).

Changes in the manuscript:

**P12, L15 | Reformulation of Eq(1):**

$$mean\ vertical\ resolution = \frac{R_d}{g} \prod_{K=1}^{M} \left( \bar{T}_k \ln \frac{p_{k-1}}{p_k} \right)^{1/M}$$

**P12, L19-20 | Delete text:**

**Section 2.3.3: this section refers to the quantification of the number of soundings which can be qualified to estimate precipitable water vapour. I am not sure to what extent this section may really confuse the reader. From one side, I think this is redundant with assessment of other indicators in the manuscript and would add value if then the selected radiosoundings according the criteria reported in this action may ready represent soundings for which an accurate estimation of the water vapor is feasible. i think the authors should clarify that thorn the radiosoundings selected according to the presented criteria allow to calculate an estimation of water vapor content which is the closest possible to the true one given the small number of vertical level available. The accuracy of the calculation of precipitable water vapour for these radiosoundings is anyhow affected by many other aspects: presence of clouds affecting the measurement sensors, homogeneity the water vapor field close to the surface, non-linearity of the water vapor variability along the vertical profile and so on.**

We know that the calculation of precipitable water vapour (PWV) is affected by radiosonde data imperfections like defective RH measurements inside (or on the way in/out to) clouds, poor resolution near the surface (where specific humidity is highest and it rapidly varying with height), and possibly insufficient resolution to describe the vertical variations of humidity in the troposphere. Leaving aside wet data biases, our approach deals precisely with vertical resolution.

To our knowledge, the effect of the vertical resolution of in-situ observations on the accuracy of PWV estimation is poorly, if ever, quantified. However, all practical studies try to guarantee that at least the surface and standard levels, normally up to 500 hPa (infrequently, up to 400 hPa) are represented in humidity soundings. This is also the requirement used by NOAA/NCEI in their IGRA-derived data set.

In the Abstract and in other parts of the manuscript, including the concluding section, we clarify that our Sfc-to-500hPa soundings are those with adequate/suitable vertical sampling (coverage and resolution) to *estimate* PWV. Our approach is this: 1) To allow an additional near-surface level to fill the lack of the 925 hPa level before 1992, on the condition that such additional level provide a similar resolution near the surface (firs 2 km); 2) To include soundings in which some standard level is missing, providing that an additional level is given in its close vicinity.

In short, the proposed definition of Sfc-to-500hPa eligible to estimate PWV [P6, L5-13] is more inclusive than the usual requirement; furthermore, it assures that the near-surface resolution is similar to that of current standard levels, regardless of the absence of 925 hPa in a large portion of past radiosonde data. The definition is explained in detail in Sect. 2.3.3, with the aid of Fig.2, and is recalled in a straightforward manner in the beginning paragraph of Sect. 3.3.

We must agree that the wording "eligible to estimate PWV", as used in the beginning line of Sect 2-3-3, is better than "qualified to estimate PWV". This applies to the Abstract and to the title of Sect. 2.3.3.

Changes in the manuscript:

**P1, L19 and P13, L1:** Replacement of "qualified" by "eligible"

**Pag.16, line 26: Please provide more details to explain the differences in the maximum range covered by the measurement of these parameters. For example, the way wind and humidity have been measured in the past compared to temperature?**

The limited vertical range of humidity measurements was mentioned throughout Sect. 1, and its relationship with the working range of humidity sensors was explained in Sect. 1.3. Regarding wind, we had in mind the limitations of PIBAL (optical theodolite and radar) wind observations, but we cannot ascertain the same about wind profiles using radio-theodolites, as in most of the radiosonde systems of the past.

In the section where page 16 is we focus on humidity data, which must be accompanied by collocated temperature data. So, we think it is wise to withdraw wind and to clarify the difference between the top of humidity and temperature data in RAOB reports. (Of course, the burst altitude evolved with rubber balloons technology; we avoid such detail, since the increase of the vertical range of mandatory levels in the early years must have adapted to the vertical reach of balloons.) For the sake of accuracy, several minor corrections will be made in other parts of the manuscript.

Changes in the manuscript:

**P7, footnote 2 regarding PILOT**
**Insert text:**
The common single theodolite technique requires the approximate ascent rate to obtain position, while the double-theodolite method allows a pure trigonometric calculation.

**P16, L20-21**
**Where it reads:**
of temperature measurements in RAOB soundings
**It will read:**
of temperature measurements (i.e. of RAOB)

**P16, L26-27**
**Where it reads:**
Contrary to wind and humidity, temperature is usually measured up to the height achieved by the sounding balloon (or close to it), i.e., the burst altitude.
**It will read:**
Contrary to temperature, which can be measured up to the maximum height achieved by the sounding balloon (burst altitude, although the highest reported level is usually limited by the standard levels in use), the vertical range reach of humidity measurements depends on the working range of humidity sensors (and, to some degree, on reporting practices).

**P16, L28**
**Where it reads:**
at present situated almost 5 km below the burst altitude
**It will read:**
at present situated almost 5 km below the maximum altitude of RAOB (close to the burst altitude)

**P17, L3-4**
**Where it reads:**
faster than the burst altitude
**It will read:**
faster than the RAOB-top

**P22, L18**
**Where it reads:**
to the burst altitude (denoted by the maximum height of the temperature measurements)
**It will read:**
to the maximum height of the temperature measurements (close to the burst altitude)

**P34, L4-5 | Caption of Figure 6**
**Where it reads:**
in the annual soundings of temperature (TMP) and humidity (HUM) across the globe: mean and quartiles. (b) As in (a), but for the vertical extent of soundings.
**It will read:**
in the temperature (TMP) and humidity (HUM) observations across the globe, calculated up to the highest measuring level for humidity: Mean and quartiles. (b) As in (a), but for the vertical extent of TMP and HUM observations.

**Pag. 21, line 1: Did the metadata adhere to any international standards like ISO19115 or WIGOS? Please clarify this aspect.**

The answer is no. In fact, such standards are of little use to our (meta) dataset for the following reasons. The present metadata do not substitute IGRA's own metadata (data sources, quality assurance, data format, stations list, periods of record, stations history). They are very specific in content, consisting on a set of values describing completeness of humidity data in a subset of the IGRA v2 main dataset – the sounding data from the stations reporting a minimum of RAOB data. Incidentally, as far as we understand, the metadata pertaining to IGRA do not adhere strictly to 'WIGOS metadata categories' or any other standard for geographic information. To avoid confusion with conventional metadata, we have named the dataset supplementary to the paper without using the word metadata: "A dataset of completeness of radiosonde humidity observations based on the IGRA".

As to metadata of the supplementary dataset, Sect. 4 and 5 of the manuscript provide primary information on the dataset, including a brief description of the represented parameters (see Tables 2 and 3), the spatial-temporal organization of data, data usability and access. The reader is referred to the Readme file for further details. Besides the short summary posted in the corresponding Zenodo's landing page, the Readme file provides the essential, mandatory elements: identification, purpose, data resources, sampling, spatial and temporal schema, data file description and format, authorship, contact. Ownership and data police are of course defined by Zenodo.

**Pag.22, line 32: "exhibits" instead of the plural.**
Correct.

**Finally, a general comment about the plots in the different Figure. They are good and clear but the quality of the figures must be improved for the printing. Supplementary material is quite useful for the reader.**

The original size of some figures is smaller than others because they were designed as in-text-figures, although observing the minimum recommended size (8 cm). If necessary, this issue can be corrected in due time.

**Authors' Response to Referee #2 Comments on**

"Completeness of radiosonde humidity observations based on the IGRA" by
António P. Ferreira, Raquel Nieto, Luis Gimeno

*Referee comments (highlighted in blue) are copied before the authors' responses.*

**The manuscript examines the completeness of newly released IGRA V2 humidity data. It is useful to have such a documentation to help users decide whether IGRA V2 has enough data for their own research before putting more efforts into downloading and analyzing the data. I would also like to appraise the authors for making their results (data) available and plan to update it on a two-year basis. I am little bit surprised on why the authors only look at humidity data, not including temperature and wind data. In "Introduction", the authors did not provide the rationale for only studying humidity observations, such as less humidity data than temperature data and degraded performance for hygrometers. Based on my evaluation, I think that the manuscript in current version needs some revision. Some of specific comments are listed below.**

Although the present work focuses on humidity, for comparison purposes, Sect. 2 and 3 provide some information about global wind and temperature data collected in IGRA [amount of stations and daily observation of the three parameters since 1905 (Fig. 1); vertical resolution/extent of temperature and humidity observations since 1945 (Fig. 6)].

The opening paragraphs of Sect. 1 recall the significance of radiosondes in accessing atmospheric humidity, as well as the role and limitations of historical radiosonde data to humidity studies. The problem of missing humidity observations due to limitations of operational radiosonde hygrometers and reporting practices, among other factors was pointed out in Sect. 1.2 and explained Sect. 1.3. Differences in vertical coverage related to reporting practices of humidity measurements will be also briefly mentioned on P2 [please refer to our reply to Referee #2].

Nevertheless, there is a particular reason why we have examined the completeness of radiosonde humidity observations, which in fact was not expressed on the manuscript. Humidity measurements (either as relative humidity or dew-point depression) require simultaneous temperature measurements, both quantities being measured at specified pressure levels. So, radiosonde humidity observations, in fact, represent simultaneous observations of pressure, temperature and humidity. Leaving aside horizontal wind – which is indirectly measured with the aid of a tracking device – humidity observations represent the most accomplished of the radiosonde observations. We think this is worth to be remarked in the revised version, not only in Sect. 1, but also in the Abstract and the concluding Sect. 6.

Changes in the manuscript:

**P1, L11-12**
**Where it reads:**
aiming to describe the completeness of humidity observations from radiosondes
**It will read:**
aiming to describe the completeness of humidity observations (i.e., of simultaneous observations of temperature and humidity) from radiosondes

**P3, L11 (beginning of paragraph) – Insert text:**

Radiosonde humidity measurements involve the simultaneous measurements of pressure, temperature and relative humidity or dew-point depression. Therefore, except for horizontal wind – which is indirectly measured with the aid of a remote tracking device – the 'humidity observations' represent the most accomplished of the radiosonde observations.

**P23, L9**
**Where it reads:**
a dataset detailing the completeness of humidity observations
**It will read:**
a dataset detailing the completeness of humidity observations (RH and/or DPD together with pressure and temperature)

**Specific comments:**

**1. P1 L16: spell out GUAN.**

GUAN is now spelled right in the Abstract, except for the letter corresponding to GCOES which (like WMO) we believe is better known to most readers and is only translated the first time it appears in the main text.

**P1, L16**
**Where it reads:**
GUAN network
**It will read**
GCOES Upper-Air Network (GUAN)

**2. P2 L30, add some of new references, such as Dai et al. (2011). Dai, A., J. Wang, P. W. Thorne, D. E. Parker, L. Haimberger, and X. L. Wang, 2011: A new approach to homogenize daily radiosonde humidity data J. Climate, 24, 965-991. This applies to other places in "Introduction".**

Dai et al. (2011) was only cited in the concluding section; we agree it should be first mentioned in Sect. 1. [Please note that we add a few more updated refs regarding Referee #1 comments.]

**P2, L29; P7, L8 – Insert citation:**
Dai et al. (2011)

**3. P3, L4: Durre et al. (2018, JTECH) should be used for the IGRA V2 reference.**

In fact, that paper was not referred before because it was published after the submission date of our manuscript. A proper citation will be inserted in several places of the revised manuscript. We think we should also put a reference to CARDS, since it is the predecessor of IGRA v1. On P7, L24, Durre et al. (2016) refers specifically to the IGRA dataset used in our work.

Note: the information on the humidity quality checks on P8, L5-12 was kindly provided to the authors by Imke Durre; however, we have lately found that it became part of the documentation publicly available on the NOAA website for IGRA since April 18, 2018.

Changes in the manuscript:

**P3, L3-4**
**Where it reads:**
The first version contained practically data after 1945 (Durre et al., 2006). The IGRA Version 2 used in this paper (Durre et al., 2016), extends back in time as early as 1905
**It will read:**
The first version of IGRA (a successor of the the Comprehensive Aerological Reference Data Set (CARDS; Eskridge et al., 1995) contained practically data after 1945 (Durre et al., 2006). The IGRA Version 2 used in this paper, very recently documented in Durre et al. (2018), has enhanced data coverage and extends back in time as early as 1905

**P8, L11-12**
**Where it reads:**
(i) for RH (with the later introduction of this variable in the archive) and (iii)–(iv) were added in IGRA 2 (Imke Durre, personal communication, April 12, 2018)
**It will read:**
(ii) for RH (with the later introduction of this variable in the archive) and (iii)–(iv) were added in IGRA 2.

**P8, L26 (before "NB") – Insert text:**
For a description of data coverage and data sources of IGRA 2, a full description of quality assurance of data, and further detail on the differences between IGRA 1 and IGRA 2, the reader should see Durre et al. (2018) published after the submission date of this work.

Additions to the reference list:

Durre, I., X. Yin, R.S. Vose, S. Applequist, and J. Arnfield, 2018: Enhancing the Data Coverage in the Integrated Global Radiosonde Archive. J. Atmos. Oceanic Technol., 35, 1753–1770, doi:10.1175/JTECH-D-17-0223.1

Eskridge, R. E., O. A. Alduchov, I. V. Chernykh, Z. Panmao, A. C. Polansky, and S. Doty, 1995: A Comprehensive Aerological Reference Data Set (CARDS): Rough and systematic errors. Bull. Amer. Meteor. Soc., 76, 1759–1775, doi:10.1175/1520-0477(1995)076<1759:ACARDS>2.0.CO;2.

**4. P4, L29: "although most of the soundings did not reach beyond 700 hPa", I think that this is outdated. Most of modern radiosonde soundings can reach above 700 hPa.**

The phrase, inside parenthesis at the end of a sentence, reports to the earliest soundings in IGRA, coincident with the earliest upper-air temperature data at Lindenberg station. We will rephrase it for clarity.

**P4, L29**
**Where it reads:**
most of the soundings
**It will read**
most of those soundings

**5. P7, L2: China doesn't use goldbeater skin anymore. Again, this info is outdated.**

Nor Russia (!), according to the very recent Bruce Ingleby's report "An assessment of different radiosonde types 2015/2016, ECMWF Technical Memorandum 807 (2017). We believe that fast-response hygristors are not yet completely outdated (e.g., GTS1-1 radiosonde).
Changes in the manuscript:

**Where it reads:**

Although the capacitive thin-film sensors have been widespread (with Vaisala radiosondes), two older sensor types are still in use: the carbon hygristor (in VIZ radiosondes) and the goldbeater's skin sensor, introduced in 1950s and still used in radiosondes made in Russia and China; this peculiar sensor responds too slowly to be useful at temperatures lower than –20 °C and suffers from hysteresis following exposure to low humidity (Nash, 2015; Moradi et al., 2013).

**It will read**

Although the capacitive thin-film sensors have been widespread (with Vaisala radiosondes RS80 and RS92) two old humidity sensor types continued in use for many years: the carbon hygristor (in VIZ/Sippican radiosondes, currently in disuse, and in the GTS1 radiosonde, in use in China) and the goldbeater's skin sensor introduced in 1950s and used in some radiosonde types made in Russia and China until a few years ago; this peculiar sensor responded too slowly to be useful at temperatures lower than –20 °C and suffered from hysteresis following exposure to low humidity (Nash, 2015; Moradi et al., 2013). For the current radiosonde types, see Ingleby (2017).

Addition to the reference list:

Ingleby, B. An Assessment of Different Radiosonde Types 2015/2016; ECMWF Technical Memorandum No. 807; European Centre for Medium Range Weather Forecasts: Reading, UK, 2017, http://www.gaia-clim.eu/system/files/publications/17551-assessment-different-radiosonde-types-20152016.pdf

**6. P13, L3: 500 hPa threshold might be too high for high elevation sites. I think that it would be 300 hPa.**

Yes, most likely. However, according to usual practice, we take 500 hPa as minimal requirement to estimate PWV for all stations. A very few studies have extended this upper limit to 400 hPa and even 300 hPa (without making any distinction between low and elevated stations). Evidently that turns out to be too restrictive for past radiosonde observation. For a first approach, we think it's necessary to pay more attention to the vertical resolution up to the 850-hPa level, since the estimation of PWV is known to be particularly sensible to the humidity profile in that region.

**7. Fig. 4: Is decreasing of radiosondes stations in 35-65N due to reducing number of radiosonde launches in Europe given the budget constrain?**

We really don´t know. The related trends at other latitudes, after 1990, seem more significant on a global perspective. A detailed analysis by country would probably give more information. However, this is out of the scope of the current analysis.

**Further author comment:**

The geographical coverage of radiosonde stations is now discussed in the related Sect. 3.1
A new figure was introduced (Fig. 3 in the revised manuscript), while Fig. 4 was redesigned.

Summary of figure changes:
Fig. 3 ← NEW  (page 46 of this document)
Fig. 4 ← former Fig. 3
Fig. 5 ← former Fig. 4, but redesigned according to the major changes in Sect. 2.3.1 and Sect. 3.1
Fig. 6 ← former Fig. 5
Fig. 7 ← former Fig. 9
Fig. 8 ← former Fig. 6
Fig. 9 ← former Fig. 7
Fig. 10 ← former Fig. 8
Fig. 11 ← former Fig. 10

[revised manuscript text omitted]

---

## Editor Decision (ED1)

Please check carefully all the Durre citations. You have Durre 2016 which describes IGRA 2 as a NOAA internal document, Durre et al. 2016 which references the actual IGRA 2 dataset, and Durre et al. 2018 which provides a published description of data coverage improvements in IGRA 2. I appreciate that at least one of these references appeared during evaluation of this manuscript, but you (and only you) can help future readers by ensuring correct references in appropriate locations. I see the qualifications on page 9 but I suspect that Durre et al. 2018 might replace Durre 2016 in many places?

Please pay close attention to tense. Technically, all references to processes and features of IGRA should occur in past tense, while all references to this work should occur in present tense. E.g. this statement from page 9 (and dozens others like it) should change: "is assured in IGRA since its first version" should instead become 'was assured in IGRA since its first version'. Otherwise, readers get confused about what happened before and what you have added here.

Page 4 line 25 "should address that issue too." I think you mean that rigorous assessment of completeness of radiosonde humidity data should include vertical and temporal coverage as well as geographic coverage? Perhaps you should write 'should address these issues simultaneously'?

Page 6 line 9: reader first encounters RAOB term. Define it as an acronym or as a coding term here? Not done in this manuscript until page 8 line 15.

Page 6 line 18: "That was no always been so" Need correction to proper English here.

Page 6 lines 28,29: I think you mean that, as opposed to expense and challenge of chilled-mirror hygrometers, weather services instead need to rely on lower-cost lighter instrumentation packages for their regular radiosonde operations? Perhaps some small changes in wording here?

Page 6 line 30: "since the time when balloon sondes were abandoned by national weather services". Very confusing, please revise. I think you mean the change from ground-tracked balloons to radio-tracked balloons, with associated changes in sensors? "electric hygrometers" not quite the correct word, I think you mean electronic sensors?

Page 7 line 3: were, not where

Page 7 line 7: "measurements in that region" What? I think you mean measurements at those altitudes and temperatures?

Page 7 line 13: "RH varied in the range 10–20 %; the lowest temperature " Make this two separate sentences: … RH varied in the range 10–20 %. The lowest temperature …

Page 7 line 15: 'new' instead of "newly".

Page 7 line 16: a Wang et al. paper that you do not cite, published jointly with Vaisala, showed that most contamination came from outgassing of radiosonde packaging materials. Protective cap solved that problem, would not have solved a 'rain' problem. Two sensors alternately heated, used initially more often in dropsondes than in up-sondes, did address the rain / cloud saturation problem. If you need to recount the contamination work, you should do it accurately. (I found the DOI for a paper in JTech, listed below.)

Page 7 line 21: again, if you intend to discuss humidity measurements in the low temperature conditions over Antartica, the French-US Concordiasi data set seems quite relevant. Notably, it includes dropsondes, upsondes and GPS occultation intercomparisons. Another Wang et al. paper, in GRL I think (I found the citation, included below). Again, if you choose to go into this detail, you should at least have the details correct! (Note: I do not have the Dirksen paper, which may include some of these references.)

Page 8 line 6: "on the observing practices intricated with sensor limitations" use a word other than 'intricated'? I think you mean 'combined with' or 'complicated by'?

Page 8 line 10: Leslie Hartten and her team did a very nice job of outlining present-day daunting logistic, communication and scientific challenges of radiosonde operations in remote environments, e.g. Earth Syst. Sci. Data, 10, 1165-1183, https://doi.org/10.5194/essd-10-1165-2018. One expects their data to appear in IGRA v3?

Page 8 line 21: confusing section. I think you miss a ')' after the Durre 2016 reference? The reference to WMO stations also introduces confusion because all or most of the 2700 IGRA stations already mentioned carry WMO station ID numbers. What distinction do you draw between "derived data for a selection of WMO stations" and standard IGRA station data?

Page 8 line 23: again, confusing. "and the former version". 'Former version' refers to IGRA? If so, state it clearly.

Page 8 line 25: 'prior' not "priory".

Page 9 line 2: 'also' not "too".

Page 9 lines 12, 13 "In wind data from pilot-balloons, the vertical coordinate is geopotential height (presumably adjusted from geometrical height measurements and the gravitational field)." Because this discussion refers to wind data, why do we care? Delete this sentence?

Page 10 line 11: "virtually" Not sure what you mean here, but perhaps you do not need this word?

Page 10 line 20: "the second world war" usually capitalized as 'second World War' or more often designated as World War II.

Page 10 line 20: 'tripled' not "triplicated".

Page 11 lines 1,2: I think you intend that readers should compare total numbers here? E.g. for 800 to 900 stations at each twice per day one would expect 1600 to 1800 soundings. Instead one observes consistently fewer than 1500 soundings per day, perhaps more like 1200 soundings. If so, be explicit. Do not assume readers will imagine your intentions. Tell us what you want us to see.

Page 11 line 5: but you have just told readers that IGRA avoids soundings without valid temperature data. So, now, by RAOB, you mean valid temperature, RH, pressure, etc? But technically a RAOB might include temperature but not RH or vice versa? So in fact you mean 'valid' or 'complete' RAOB messages, instead of generic RAOB? If I get confused, readers will get more confused.

Page 11 line 7: "relatively few RAOB data". Here, clearly, you define RAOB as consisting of full valid all-parameter data (T, RH, pressure, etc.). But earlier you have defined RAOB as a generic

radiosonde observation, quality unknown. To understand your selections and corrections, you need to adhere to, and give the reader, a clearer definition of what you mean by RAOB. Any sounding data, or only a valid full-parameter sounding? Which definition you intend makes a very large difference. Making the definition clearer will greatly improve the text of this manuscript.

Page 11 line 12 "the IGRA-RS subset retains practically all the RAOB soundings" Because of confusing definitions, how could any IGRA-RS subset not automatically consist entirely of valid RAOB data? I think the point you mean to make here is that taking only the radiosonde fraction of IGRA, e.g. IGRA-RS, still retains most of the original IGRA RAOB data. If so, then your earlier definition of a RAOB as a radiosonde observation (page 8 line 15) seems again confusing. Some RAOBs are not valid RAOBs? Or, some RAOBs are not valid radiosonde data, even though you define a RAOB as a radiosonde observation? I understand confusing often inconsistent meteorological terms, but here you have amplified the confusion? Readers need your best guidance but do not get it.

Page 11 line 16: "missing years are considered". I think you mean 'included' or 'included and identified'. 'Considered' does not tell us how your selection process treated missing years.

Page 11 line 17: here a reader learns that 1300 of 1700 stations (75%) carry WMO ID numbers. This does not explain nor accord with the statements on page 8 (noted above) about a "selection of WMO stations".

Page 11 line 21: "integrating" I think you mean 'integrated into' or 'coordinated through'?

Page 11 line 23: "together with the surface stations of the GCOS Surface Network" Why do we need this? Is this somehow relevant to the upper air humidity data? If not, omit?

P12 section 2.3: a very good description of the core motivation of this work! These questions should move to the top, even to the abstract. Readers should not need to wait until this point to understand the motivation!

Page 14 line 25: here a reader again finds reference to 'RAOB' reports when in fact the discussion pertains to IGRA-RS? More confusion?

Page 14 line 29: here the authors include moving stations but earlier, under global coverage, they only included fixed IGRA stations. Readers need better information about which subsets used when? If moving stations only a small fraction of total IGRA stations, why include them in this analysis? What value, if any, do they add?

Page 16 line 2: 'shown' rather than "show".

Page 16 line 7: increases in a step-wise manner

Page 16 line 20: "readiness"? I do not understand this word in this context. Change, please.

Page 16 line 23: now we have "RS" stations. So, IGRA, IGRA RS, RAOB, RS. Do the authors follow a deliberate plan to confuse readers?

Page 16 line 29: "repeated, by restricting" Remove this comma.

Page 17 lines 3 to 5: finally, here, a reader learns about fixed versus mobile and which analysis used which subset. We should have had this information much earlier, at the start of section 2 or even as part of the introduction?

Page 17, line 11: "IGRA-RS excludes the IGRA stations without any RAOB at all" this statement is **NOT** consistent with earlier use of or definitions of terms RAOB and IGRA-RS. I suspect the authors know what they intend, but they have only confused their readers.

Page 17 line 11: "comparison between Fig. 3 and Fig. 1a" Following (correctly, I hope) the authors' intent, extrapolating from Fig 1a, in Fig 3 I should see, between 1955 and 1975, a 10-fold increase in non-humidity (e.g. PIBAL) stations - which my eyes do NOT see - and, between 1975 and 2015, a steady number of humidity stations accompanied by a decreased number of PIBAL stations. Why do I not see the 1955 to 1975 differences? Bad eyes? Bad figure? Can I actually confirm the drop in number of PIBAL stations in 2015 relative to 1975? Authors should provide guidance to readers about what the authors wish readers to see, and ensure that Figures support that evidence? Not clear in this instance?

Page 17 section 3.1 At the low given resolution of Figure 3 (it does zoom in nicely on my screen), the reader doubts whether we can confirm the temporal changes in geographic patterns described by the authors. A reader almost certainly has zero ability to detect "four fixed weather-ships". We either need descriptions better scaled to the maps or better maps.

Page 17 line 29: observations reported by Driemel actually included a large fraction gathered during transit, e.g. north and south along the Atlantic oceans, with perhaps the largest fraction between 60N and 60S? Polar yes, and very valuable, but not exclusively polar.

Page 18 line 6: use arctic or Arctic, but at least use it consistently. Copernicus style sheet suggests 'Arctic'.

Page 18 and Figure 4: Whatever the authors may have intended here, they have largely failed. Figure 4 remains almost impossible to understand, readers need to spend way too much time trying to understand it. What the authors' claim as climate zones actually represent latitudinal bands instead, and not evenly distributed in any case. Properly speaking, climate zones include elevations, distance from coastlines, location with respect to monsoonal circulations, etc. We get (combined) 46 degrees of equatorial, 23 degrees of sub-tropical, 60 degrees of temperate and 46 degrees of polar (using my own guesses at names for the zones). Figure 4 exacerbates this confusion, with quantities and lines in no particular order or calibration. How do we compare a northern sub-tropical range of 12 (23 to 35) degrees with an equatorial region of 46 degrees or a south polar region of 23 degrees? I have no doubt the authors understand the data to the resolution of "two ships reporting radiosonde observations in waters around the Arctic Circle" (page 18 line 9) but readers will not find anything like that detail in these figures. Going back to questions on page 12 (and notice there that the authors used the word "latitudes" rather than climate zones) do we really need any of this detailed location by location discussion? Instead, authors could help readers by defining latitude zones appropriately (30, 60, 90, etc.) or at least of equal latitudinal extent and then draw our attention to temporal patterns within those zones. Describe the data sufficiently so that subsequent users can explore specific latitudinal or zonal features based on their own criteria?

Page 18 lines 11,12: "relative weight has been decreasing" "weight'? I think you mean number or frequency?

Page 18 throughout: "Tropics", "extratropics", "climate zone", "latitude band" - terminology and punctuation very inconsistent throughout this section.  Needs careful attention and

correction to achieve consistency as well as accuracy. Really too many to note them all, needs thorough scrubbing and appropriate revisions. Authors make appropriate notice of land to ocean differences by hemisphere but then compare Arctic (ocean) with Antarctic (land) without any such qualifications.

Page 18 line 30: "While the same is impracticable in many other parts of the world and over the oceans, distances up to two or three times larger than ideal are accepted, in view of the relatively mild climatic conditions on oceans and the fulfillment from surface and satellite observations." I believe the authors intend this as a description based on practical realities but many researchers would not agree that we should find the situation acceptable? We certainly need sea surface temperatures, surface roughness, cloudiness and rainfall, and interior ocean temperatures (e.g. by Argo) at much higher temporal and spatial resolution. Last sentence in this paragraph (e.g. top of page 19) also contradicts this statement? Statement represents a lightning rod, authors might do themselves a favor to omit it?

Page 19 line 9: "Besides, the corresponding …" Delete the first word.

Page 19 line 14: "we only care with sub-year missing days" I think you mean 'we focus only on sub-year'?

Page 19 line 15: "Fig. 7 gives a glint of the typical continuity …". I think you mean 'Fig 7 offers a summary of the typical ….' Or 'offers an indication'.

Page 19 line 16: "between 1945 and 1960" In fact, Fig 7 shows that number of missing days dropped much faster than your phrases suggests, over not more than 5 years. The pattern looks like an initiation or spin-up problem, which you have hinted at elsewhere.

Page 20 line 12: 3/4 (also in line 13). Please use percentages as you do elsewhere, not fractions.

Page 20 line 13: "data reach 22 km". In the previous line you gave us pressure then altitude, e.g. 100 hPa roughly 10 km. Here you should do the same, e.g. something like 50 hPa roughly 22 km.

Page 20 line 14: "This difference " refers to differences in maximum height or to differences in height in temperature records versus humidity records? Need clarity here.

Page 20 line 23: "This last feature and is coincident " remove the word 'and'

Page 20 line 30 - use percentage not fraction.

Page 20 line 31, 32 "fairly recent measurements in the upper-troposphere, and certainly more above too, was considered inadequate for climate". I think you mean ' and certainly into the lower stratosphere'? Recent as used here means before the Durre 2005 reference, so not the most recent. No widespread globally-useful solution, certainly, but other people have worked on this problem?

Page 21 line 14: Most mobile soundings come from ships and for those the baseline elevation is always sea level plus/minus 10 m at most?

Page 22 lines 5,6: Confusing. I think you mean that, for a fairly high standard such as expecting 95% of stations to have valid lower-troposphere humidity data, the number of stations meeting that standard remains very low from start of the records in 1945 to as recently as 1990. You use P in percentage to indicate fraction of stations while many readers familiar with statistical

analysis will understand P as probability. You need to give explicit explanation of what you describe and how you measure it.

Page 22 line 19: "two noteworthy change points: a sudden increase around 1970 and 2000". Singular plural problem: two change points result in sudden increases around 1970 and 2000.

Page 23 line 25: "Evidently, Fig. 11 is only …" Remove the first word.

Page 23 line 25: "the question of since when we have enough" remove the word 'since'

Page 25 lines 10, 11: "selecting the stations with a minimal amount of radiosonde" I think you mean by selecting those stations with a sufficient amount? Minimal as you use it in this case technically indicates few or fewest, not what you intend. You mean 'exceeded minimal standards'?

Note - the first five sentences of section 6 make a very good abstract, much better and more concise than the one you have.

Again, arctic vs Arctic. Please check and correct!

Page 26 line 9 - use percentage not fraction

Page 26 line 22 "was not standard and it was rarely used" delete word 'it'

Page 26 line 23 "consecutive years of data until a given year" I think you mean number of consecutive years greater than some specified value? Later in that sentence, depends on the value, the time span and the completeness criteria.

Page 26 line 24 "E.g.: the station-based" Do not start a sentence with an abbreviation. Instead, write it out: For example.

Page 26 line 27 "Evidently, the equivalent time-series …" delete the first word.

Page 38 - why do we get Figure 2 in black and while while we get other figures in useful helpful colors?

Page 39 - Figure 3 still very hard to read at page resolution, but works okay with page zoom.

Page 40 - Figure 4, latitude bands not climate bands, latitude bands inconsistent in extent, figure hard to read and harder to understand. Change to standard latitude bands as used in Figures 6 and 7.

Page 41 - potentially useful information but limited by use of the variable latitude bands. Use standard latitude bands as in Figures 6 and 7?

Page 42, Figure 6 - at a std deviation of 20%, evidently no significant differences among day fraction by latitude. If not, combine the lines into a composite, both for the absolute value and for the standard deviations. E.g. no valid distinctions between these lines so why show them separately?

Page 42 Figure 7 - see comment above about data that I made in reference to Page 19, above; needs color.

Page 43 Figure 8a - no differences statistically or visible between T and RH mean and quartile values in this panel, nothing gained by showing them super-imposed? Combine them? Or delete this panel and focus instead on panel b?

Page 44, figures 9 and 10.  Why not color instead of grey-scale? Do we really need both 9 and 10, as they basically tell the same story? Panel a in Figure 10 could group to two lines, one before 1990 and one after? Comparison with Panel b would still hold?

Page 45, Figure 11, useful, but why not use the same color scheme both panels?

 Page 46, Table 1: relevant to RAOB, IGRA, IGRA RS confusion above, here RAOB IGRA and RAOB IGRA RS have almost exactly identical numbers of soundings with humidity. Differences in total soundings only roughly 300 out of nearly 30 million, e.g. roughly 0.001% difference? What exactly drives the distinction in terms?

Two additional references mentioned above, authors to include if considered useful and relevant:

Wang et al., https://doi.org/10.1175/1520-0426(2002)019<0981:COHMEF>2.0.CO;2

Wang, J., et al., Geophys. Res. Lett., 40, 1231–1236, doi:10.1002/grl.50246

---

## Author Response (AR2)

13 March 2019

Dear David Carlson,

I am pleased to submit a revised version of the article 'Completeness of radiosonde humidity observations based on the IGRA', following the most recent evaluation of our previous revision.

The reply to the reviewer's comments and suggestions (Reviewer # 1, 1 March 2019) is provided below (pp ii-viii), followed by the marked-up manuscript version. Please note that the present revision includes technical corrections besides the modifications detailed in the authors' response. Other minor but significant corrections are highlighted in yellow.

Sincerely,

António P. Ferreira

Authors' Response to Reviewer #1 Comments (1 March 2019) on

"Completeness of radiosonde humidity observations based on the IGRA"
by António P. Ferreira, Raquel Nieto, Luis Gimeno

*Referee comments are quoted point-by-point in blue italic before the corresponding authors' responses*
*Page& line numbers refer to the previous revised, clean version of the manuscript*

*After the first review stage, the manuscript "Completeness of radiosonde humidity observations based on the IGRA" by António P. Ferreira, Raquel Nieto, Luis Gimeno, has been significantly improved. The authors considered most of the reviewers' comments discussing the way they totally or partly accepted many of the received comments.*

*I think the manuscript is in a better shape now.*

*I provide the authors with a few minor revisions, which to my opinion must be still considered to complete the good work they did and to avoid confusion in the reader.*

*I appreciated the changes in the abstract to reduce the misperception that the scope of the paper is to demonstrate that the described the data may support or not climate studies. I think the work already done by the authors to smooth the text should be further refined providing at the top of the manuscript (the first time this is mentioned) what they mean for "completeness". The authors concretely describe in the text, in particular, in section 2.3.2, that data completeness indicates the "regularity and continuity" of the available radiosounding: I think that a definition of the concept of completeness should be provided at top of the manuscript to further clarify the objectives of the paper.*

*Abstract L16: GCOES Upper-Air Network (GUAN), acronym must be corrected in "GCOS".*

We would like to thank Reviewer # 1 for helping us to further improve the manuscript as far as possible.

The concept of completeness of radiosonde humidity observations is described extensively in the Abstract, as follows:

«Completeness of humidity observations (either relative humidity or dewpoint-depression) for a radiosonde station and a full year is defined by: the number of humidity soundings; the fraction of days having humidity data; the mean vertical resolution of humidity data; the mean atmospheric pressure and altitude at the highest measuring level; and the maximum number of consecutive days without humidity data. The completeness of the observations eligible for calculating precipitable water vapor – i.e., having adequate vertical sampling between the surface and 500 hPa – is particularly studied. Individual soundings are described by the (vertically averaged) vertical resolution and the pressure level and altitude of the top of humidity measurements.»

For clarity and emphasis, we decided to move the phrase "(either relative humidity or dewpoint-depression)" and indicate the number of completeness parameters listed after the colon.

In the main text, the concept is defined right in the introductory part of Sect. 1 [on P3], in relation with the purpose of the paper:

«In view of the huge amount of data collected in IGRA (which is also a combination of radiosonde and pilot-balloon observations) and the differences in the observing period, temporal regularity, continuity, vertical resolution and vertical extension of humidity data among different stations, finding the most suitable humidity-reporting stations (or humidity soundings from different stations) for a specific purpose can be difficult to put into practice.»

«The purpose of this paper is to study the completeness of humidity observations collected in IGRA according to various needs – number and latitudinal distribution of observing stations, fraction of observing days in a year, resolution and range of vertical levels, length and continuity of the time-series, minimal sampling between the surface and the 500-hPa level – aiming to facilitate the use of radiosonde humidity data by atmospheric and environmental scientists. The task is two-fold: first, to elucidate the completeness of the humidity observations from IGRA for each year in global terms, including the latitudinal coverage of stations and the length of regular time-series; second, to provide metadata describing the completeness of humidity observations from each station.»

Sect. 1.1 [PP 4-5] further clarifies the concept, providing examples from literature and identifying the factors that contribute to differences in the completeness of humidity observations among radiosonde stations and individual soundings.

NB: Section 2.3.2 focusing on temporal completeness within each year is only a part of Sect. 2.3. Subsections 2.3.1 through 2.3.5 provide different measures of completeness of observations and the corresponding methods of statistical analysis used later in Sect. 3.

By mistake, in the title of Sect. 2.3.2 we wrote "regularity" instead of "temporal continuity".

**Changes in manuscript:**

**P1, L 16 – Typo corrected:** "GCOES" → "GCOS"

**P1, L 12 & L 17 – Text is parenthesis moved, rephrasing:**

"of humidity observations – i.e., of simultaneous observations of temperature and humidity – from radiosondes in different times and locations"
→ "of radiosonde humidity observations – i.e., of simultaneous measurement of pressure, temperature and either relative humidity or dewpoint depression – in different times and locations"

"Completeness of humidity observations (either relative humidity or dewpoint-depression) for a radiosonde station and a full year is defined by: the (…)"
→ "Completeness of humidity observations for a radiosonde station and a full year is defined by five basic parameters: (…)"

**P 13, title of Sect. 2.3.2 – Replacement:** "regularity" → "temporal continuity"

**P27 – Revised form of the acknowledgment section:**

"We thank Imke Durre for clarifying some points about the Integrated Global Radiosonde Archive Version 2 during the preparation of this work, David Carlson for editorial advice on the first draft of the paper, and two anonymous reviewers for helpful comments. We also thank Orlando García for IT support at EPhysLab (Environmental Physics Laboratory at the Faculty of Science of the University of Vigo, Ourense Campus). This work was funded by the Spanish government within the EVOCAR project (CGL2015-65141-R), co-funded by the European Regional Development Fund, and partially supported by the Galician regional government through the program Consolidation and Structuring of Competitive Research Units – Competitive Reference Groups (ED431C 2017/64-GRC)."

*1. At lines P1 L23-25, the authors state "This overview indicates that the number of radiosonde stations having a record length potentially long enough (multi-decadal) for climate studies involving humidity-related quantities depends not only on the temporal range, but also on the continuity, regularity and vertical sampling of the humidity time-series." I think in this sentence they may provide the impression that the paper quantifies the number of radiosonde station providing a good "climate data record". If I am not wrong they do not quantify this number but they provide a dataset where this quantification is possible for each specific climate application. The could be better clarified in the manuscript.*

The quoted sentence from the Abstract is related to the results presented in Sect. 3.5, Fig. 11. By "potentially long enough" we mean that completeness of time series may not be enough to have a good climate record from a single station, since data accuracy and inhomogeneities due to instrument changes are also important. Maybe the sentence in question is not clear enough and needs a bit of rephrasing.

The next sentence in the Abstract:

«It is hoped that the derived metadata will help climate and environmental scientists to find the most appropriate radiosonde data for humidity studies by selecting upper-air stations, observing years or individual soundings according to various completeness criteria – even if differences in instrumentation and observing practices require extra attention.»

points to different applications, other than climate studies (the ones which require long-term records). This can be clarified by adding "More generally" at the beginning of the latter sentence.

In the main text, different applications are also suggested in the Introduction [P3]:

«The purpose of this paper is to study the completeness of humidity observations collected in IGRA according to various needs (…) aiming to facilitate the use of radiosonde humidity data by atmospheric and environmental scientists.»

Section 3.5 clarifies how the availability of long-term time series for climate purposes can be quantified from the provided dataset, namely [on P 23]:

«Evidently, Fig. 11 is only illustrative. There is no simple answer to the question of since when we have enough data to (in theory) perform climate studies from radiosonde humidity data: it depends on the strictness of completeness criteria. Note that the length of the time-series under user-specified conditions for any of the metadata parameters defined in the next section, either backward or forward in time, can be derived from the information contained in the same metadata set. In addition, the years with humidity observations can be found in Table S1 (as well as in a file accompanying the two main metadata sets): one or more periods per station, depending on whether there are gap years present in the station's full period of record.»

However, maybe the words "In theory" in the above paragraph should be slightly explained.

Recall that the concluding Sect. 6 [in the last bulleted paragraph on P 26] remarks:

«(…) In short, the amount of humidity data that are available to perform climate studies, discounting discontinuities in accuracy, depends strongly on the strictness of the completeness criteria.»

But accuracy itself is important, not only "discontinuities in accuracy", as is clarified now.

Section 6 [near its end; references are omitted below as (…)] remarks how our dataset can be helpful to select good climate records:

«It is widely known that the usefulness of historical radiosonde data depends crucially on metadata information about instrumentation and observing practices (…). However, sampling differences among stations associated with geographical coverage, observing periods and missing data, are no less important than differences in data precision and accuracy. Reporting practices related to limitations of humidity sensors affect particularly the humidity records (…). In this respect, the metadata presented in this paper, if used as a tool to find out the more complete humidity time series – relatively long, regular and continuous; vertically extensive and well resolved –, should be crossed with the coordinates of stations and the station history metadata available in IGRA. On the other hand, the present metadata might accelerate progress in the current research on the homogenization of radiosonde humidity data (…), by sampling stations with coincident observing periods and satisfying reasonable completeness criteria.»

**Changes in manuscript:**

**Abstract, P 1 L 25 – Rephrasing:**

"having a record length potentially long enough (multidecadal) for climate studies involving humidity-related quantities depends not only on the temporal range, but also on the (…)"

➔ "having a record length long enough to be potentially useful for climate studies involving humidity-related quantities depends not only on the temporal range (multidecadal), but also on the (…)"

**Abstract, P 1 L 26 – Insertion:** "More generally" (…)

**P 23, L 16 – Insertion:**

"(In practice, it depends also on the accuracy and homogeneity of measurements by different instruments.)"

**P 26, last bulleted paragraph – Rephrasing:**

"(…) the amount of humidity data that are available to perform climate studies, discounting discontinuities in accuracy (…)"

➔ "(…) the amount of humidity time-series that are *potentially* available for climate studies, i.e., discounting accuracy requirements and biases due to instrument changes (…)"

*2. I think a comparative study between IGRA v1 and IGRA v2 could be at least mentioned as an outlook study for future studies given that the discrepancy between the two data version has been noticed by several scientists.*

We have decided not to engage in a comparative study between IGRA v1 and IGRA v2 for two reasons. First, IGRA v2 represents a major update of IGRA v1, and the former version is no longer maintained. Second, a comparison between the two versions would complicate unnecessarily our study and the related dataset, which refers to the updated version of IGRA.

Moreover, we are not aware of discrepancies between the two versions – i.e. disparities of data values for the same station, date and variable. Surely, they are not reported in literature because IGRA 2 is relatively recent. In this instance, we prefer not to discuss the subject.

Therefore, in the manuscript we have already mentioned the main differences between the two versions, providing the most adequate reference – i.e, the paper on IGRA 2 by Durre et al. (2018) published shortly after the first draft of pour work was written. Namely:

- The enhanced data coverage in IGRA 2 is first mentioned in Sect. 1, P 3, where Durre et al. (2018) is firstly cited, and detailed a bit in Sect. 2.1, P 8.

- The extension of humidity data and improvements in quality assurance are described also in Sect. 2.1, P 9.

- For further details on the differences between IGRA 1 and IGRA 2, on P 9 the reader is referred to Durre et al. (2018).

Regarding the quality assurance procedures in IGRA, we now provide a reference to a previous document prepared by Imke Durre in 2016, which is available on IGRA's NOAA landing page and thoroughly describes the data screening and the improvement of IGRA 2 relative to IGRA 1, even if without graphical illustrations.

**Changes in manuscript:**

**P 9, L 11– Insertion:** "(Durre, 2016)"

**Reference List – Added reference:**

"Durre, I.: Integrated Global Radiosonde Archive V2, Dataset Description, Version 1.0, 15 pp, [Word document available at https://www1.ncdc.noaa.gov/pub/data/igra], 2016"

*3. Moreover, going to the description of IGRA data archive in the manuscript, though the NOAA's IGRA web page says, "over 2700 stations" are available, but if the authors read the page until the end you will notice the following sentence: "The period of record of The IGRA data varies from station to station and among variables. Approximately 1,000 of the over 2,700 IGRA stations are currently reporting data.". Discussion in many context and also in the IGRA v2 paper are in compliance with this sentence which gives the real picture of the IGRA database at present. I think you must add this to your description to also better introduce the data analysis shown in the manuscript (e.g. Figure 3 and related text).*

We believe that the following excerpts of the manuscript agree with the above quoted sentence and provide enough information:

[P 3]

«In view of the huge amount of data collected in IGRA (which is also a combination of radiosonde and pilot-balloon observations) and the differences in the observing period, temporal regularity, continuity, vertical resolution and vertical extension of humidity data among different stations, finding the most suitable humidity-reporting stations (or humidity soundings from different stations) for a specific purpose can be difficult to put into practice.»

[P 4]

«The geographical coverage of radiosonde stations evolved over time, and so, the period of usage varies among stations.»

[P 8]

«The IGRA 2 consists primarily of radiosonde and pilot-balloon soundings taken at over 2700 globally and temporally distributed stations, even though the coverage over oceans is limited to ships, buoys and remote islands. IGRA 2 has also derived data for a selection of WMO stations, but this paper concerns with the sounding data (Durre et al. 2016, comprising over 45 million soundings from 2761 (2662 fixed and 99 mobile) stations [based on data accessed in September 2017].»

[P 9]

«The examination of the IGRA reveals that 958 stations have wind-only observations in their full period of record, i.e., 34.7 % of the stations represented in the entire archive.»

[P 10]

«Constituting the bulk of the IGRA stations until the early 1940s, the PIBAL stations represent nowadays only 13 % of the total. (…) The number of RAOB stations increased rapidly from the mid-1940s, staying in the range 850–900 from around 1970 to virtually present (2016).
(…)
«After a major development until around 1970, the number of reporting radiosonde stations has been in the range 850–900.»

Incidentally, the last sentence is redundant and will be removed. Also, referring to Fig. 1, the number range indicated for the radiosonde stations after 1970 must be rectified to 800–900.

Regarding specifically to the number of IGRA stations that are currently reporting data:

Figure 1 in the manuscript shows that, by 2016, the number of IGRA stations reporting RAOB data was about 830, while the number of stations reporting wind-only data was about 140. In sum, there is a total of almost 1000 stations reporting any data at present. We think, however, that showing numbers for PIBAL and RAOB stations separately, as we did, is more informative.

Since the recent paper by Durre et al. (2018) (referred to as 'IGRA v2 paper' by the Reviewer) gives a first-hand account on the above issues, we see no need for adding further details.

We should keep in mind the focus of the present paper, which is to describe the completeness of radiosonde humidity observations collected in IGRA (simultaneous measurement of P, T, U). The selection of IGRA data is described on PP 12-13 – indicating the # of stations with a minimum of radiosonde data during any year in their period of record until the end of 2016 (1723), the # of WMO stations in that subset (1300) and the # of GUAN and GRUAN stations among the WMO stations (178 and 16, respectively). Tables 1, S1 and S2 provide information on the selected stations (IGRA-RS): # of stations, total # of PIBAL/RAOB/humidity observations; identification of stations,  periods of observation for any variable and for humidity, # of humidity observations; detailed statistics for GUAN stations. In addition, Fig. 3 shows the location on the globe of the IGRA-RS stations at significant years outlining the evolution over time of the global radiosonde network; note that the # of active/reporting humidity stations is reported in the panel titles.

**Changes in manuscript:**

**P 10, L11 – Correction:** "850–900" → "800–900"

**P 10, L 23 – Removed text:** "After a major development until around 1970, the number of reporting radiosonde stations has been in the range 850–900".

*4. at P11, L4, I0d change the text from:*
*GRUAN aims to serve as reference to other radiosonde networks, by providing long-term high-quality records of vertical profiles of selected essential climate variables, accompanied by traceable estimates of measurement uncertainties.*
*to*
*GRUAN aims to serve as reference networks, for climate applications, satellite validation and in support of other networks by providing long-term high-quality records of vertical profiles of selected essential climate variables, accompanied by traceable estimates of measurement uncertainties.*

Done, with trivial adjustments.

We take the chance to abridge the next sentence in the same paragraph and move the citations. This is in line with the former reviewer's comments, prior to the first revision of the manuscript.

**Changes in manuscript:**

**P12 – Text amendment:**

"The GRUAN aims to serve as reference to other radiosonde networks, by providing long-term high-quality records of vertical profiles of selected essential climate variables, accompanied by traceable estimates of measurement uncertainties. Despite the differences in data processing between GRUAN internal data and real-time data, the quality of routine observations carried out at GRUAN sites in recent years should be above average (WMO, 2011a; Dirksen et al., 2014)."

→

"GRUAN aims to serve as a reference network for climate applications, satellite validation and in support of other radiosonde networks, by providing long-term high-quality records of vertical profiles of selected essential climate variables, accompanied by traceable estimates of measurement uncertainties (WMO, 2011a; Dirksen et al., 2014). Naturally, real-time meteorological data transmitted from GRUAN sites to the GTS may differ from GRUAN internal data regarding raw data processing."

*5. Finally, I want to remark again that not al the figures are of enough quality. They should be improved and the authors declared to be glad to work on it. I hope in the next stage of the paper production this will be recommended to the authors. Moreover, I'd suggest a careful revision to remove a few typos*

Figures are now improved, allowing enlarging without loss of resolution. Files for production are in pdf format.

Typos were corrected (some resulted from the rearranging of Sections 2 and 3 in the previous major revision, including the title of Sect. 3.1 which was not correctly updated). **These and other technical corrections are indicated in the marked-up manuscript.**

[revised manuscript text omitted]

---

## Author Response (AR3)

06 April 2019

Dear David Carlson,

I and my co-authors would like to thank you for all your corrections and suggestions. We hope our revision is sufficient for the manuscript to be accepted for publication in ESSD. The authors´ response is provided below, followed by the marked-up manuscript.

Sincerely,
António P. Ferreira
* * *
**Authors response to Topical Editor report for the manuscript**
'Completeness of radiosonde humidity observations based on the IGRA'
by António P. Ferreira et al. (MS No.: essd-2018-95)

*Topical Editor's comments are highlighted in blue. Page and line numbers in the authors responses refer to the previously revised manuscript, unless stated otherwise*
* * *
Please check carefully all the Durre citations. You have Durre 2016 which describes IGRA 2 as a NOAA internal document, Durre et al. 2016 which references the actual IGRA 2 dataset, and Durre et al. 2018 which provides a published description of data coverage improvements in IGRA 2. I appreciate that at least one of these references appeared during evaluation of this manuscript, but you (and only you) can help future readers by ensuring correct references in appropriate locations. I see the qualifications on page 9 but I suspect that Durre et al. 2018 might replace Durre 2016 in many places?

Since the (meta)dataset and the analysis presented in our work derive from the current IGRA, it would be impossible to neglect the proper citations.

The first version of the manuscript included an obvious reference to the paper by Durre et al. (2006), which describes IGRA v1; another obvious reference to Durre et al. (2016) indicating the IGRA v2 dataset used by us; and two personal communication citations, since Imke Durre had clarified some questions on IGRA 2 during the preparation of the first draft of the manuscript and gently provided us a first-author technical description of IGRA 2 (the NOAA internal document). Two other references to Imke Durre and co-authors (2005, 2009) were also included.

During the revision process, we added two major references for common-sense reasons. 1) Durre et al. (2018), the recent paper describing IGRA 2; when preparing the discussing manuscript, we were unaware of the publication of that paper a few weeks before; that explains why the paper was only cited in our first revision (on page 3 in the Introduction on page 9 in the section about IGRA sounding data). 2) Durre (2016), referring the NOAA internal document, replaced a personal communication citation since we noticed that

that document became available on the NOAA website for IGRA practically at the same time of the communication in which Imke Durre gently provided it to us.

Concerning the present revision:

**->** Durre et al. (2016) is added to Durre (2018) on page 3, line 10, where IGRA is first mentioned;

**->** Durre et al. (2018) is added to Durre (2016) on page 9, line 10, since both works give the same information in different ways;

**->** Durre et al. (2016) and Durre et al. (2018) are both cited again in the opening lines of the concluding section 6.

Please pay close attention to tense. Technically, all references to processes and features of IGRA should occur in past tense, while all references to this work should occur in present tense. E.g. this statement from page 9 (and dozens others like it) should change: "is assured in IGRA since its first version" should instead become 'was assured in IGRA since its first version'. Otherwise, readers get confused about what happened before and what you have added here.

We made a number of significant corrections. Note - sometimes we keep past tense to refer what we did before performing calculations.

Page 4 line 25 "should address that issue too." I think you mean that rigorous assessment of completeness of radiosonde humidity data should include vertical and temporal coverage as well as geographic coverage? Perhaps you should write 'should address these issues simultaneously'?

Changed:

"issue of geographical coverage" → 'geographical coverage of stations'
"should address that issue too" → 'should address both issues simultaneously'

Page 6 line 9: reader first encounters RAOB term. Define it as an acronym or as a coding term here? Not done in this manuscript until page 8 line 15.

The mistake resulted from the addition of the related paragraph in the last revision. We prefer not to us the term RAOB here since it is only needed from section 2 to 4. The less technical parts of the manuscript (abstract, introduction and concluding section) avoid unnecessary use of acronyms. "RAOB reports" → 'radiosonde reports'

Page 6 line 18: "That was no always been so" Need correction to proper English here.

'That has not always been so'.

Page 6 lines 28,29: I think you mean that, as opposed to expense and challenge of chilledmirror hygrometers, weather services instead need to rely on lower-cost lighter

instrumentation packages for their regular radiosonde operations? Perhaps some small changes in wording here?

"meteorological radiosondes are" was replaced by 'whether services need to rely on meteorological radiosondes consisting of'.

Page 6 line 30: "since the time when balloon sondes were abandoned by national weather services". Very confusing, please revise. I think you mean the change from ground-tracked balloons to radio-tracked balloons, with associated changes in sensors? "electric hygrometers" not quite the correct word, I think you mean electronic sensors?

"Balloon sondes" is one of the names for registering balloons – very confusing indeed since it can also refer (uncommonly) to radiosondes. Replaced by 'registering balloons'.

The terminology "electric hygrometers" follows from DuBois (2002) but it can be found in other places, e.g., Meteorological Measurement Systems by Fred V. Brock and Scott J. Richardson (Oxford University Press). Although radiosonde measurements involve a lot of electronics besides radio transmission of data, the above terminology derives from the fact that most humidity sensors for remote applications are based on the change of an electrical parameter caused by moisture: capacitance, in polymer capacitive sensors; resistance, in the carbon hygristor; ionic conductivity, in electrolytic humidity sensors like the LiCl electrolyte sensor invented in 1937.

Page 7 line 3: were, not where

Typo corrected.

Page 7 line 7: "measurements in that region" What? I think you mean measurements at those altitudes and temperatures?

We mean low HR region in the domain 0-100%. Replaced by 'measurements in that low RH region', where "that" refers to the low values mentioned in the previous sentence (less than 15–20%).

Page 7 line 13: "RH varied in the range 10–20 %; the lowest temperature " Make this two separate sentences: … RH varied in the range 10–20 %. The lowest temperature …

The two sentences form part of a list with many items. This is now separated by subject.

P7, L12-15 – Rephrasing (changes are underlined):

'Note that changes in instrument and reporting practices in different countries took place at different times: the threshold value of RH varied in the range 10–20 %; the lowest temperature of –40°C for reporting humidity, and the shift to lower temperatures, was applied in different periods depending on country; humidity could be reported up to a specified pressure level. Moreover, mechanical sensors are not exclusive to pre-1940s radiosondes: hair hygrometers were only abandoned in the mid-1950s and rolled hair hygrometers were used in a few places until about 1980; the goldbeater's skin sensors introduced in the 1950s became particularly important in the Soviet Union. [For historical details on these changes, see Gaffen (1993).]'

Page 7 line 15: 'new' instead of "newly".

Corrected.

Page 7 line 16: a Wang et al. paper that you do not cite, published jointly with Vaisala, showed that most contamination came from outgassing of radiosonde packaging materials. Protective cap solved that problem, would not have solved a 'rain' problem. Two sensors alternately heated, used initially more often in dropsondes than in up-sondes, did address the rain / cloud saturation problem. If you need to recount the contamination work, you should do it accurately. (I found the DOI for a paper in JTech, listed below.)

Thank you for enlightening us. We were referring to the protective rain cap, not to the protective shield to avoid chemical contamination. Both were used in RS80 radiosondes, but absent in RS90 and RS92. Now we understand the outgassing issue affecting RS80 sondes until 2000. While for RS90 and RS92 the major problem was solar radiation.

The text was extended for the sake of accuracy. Telling the main features of humidity sensors in the main Vaisala radiosondes seems a solution. We have cited the suggested paper (Wang et al., 2002) and also a book chapter (Smit et al. , 2013). Please see the changes on p. 8 of the marked-up manuscript.

Page 7 line 21: again, if you intend to discuss humidity measurements in the low temperature conditions over Antartica, the French-US Concordiasi data set seems quite relevant. Notably, it includes dropsondes, upsondes and GPS occultation intercomparisons. Another Wang et al. paper, in GRL I think (I found the citation, included below). Again, if you choose to go into this detail, you should at least have the details correct! (Note: I do not have the Dirksen paper, which may include some of these references.)

We have mentioned the low temperatures over Antarctica by way of example and see no need for further discussing the subject. We appreciate the suggestion of Wang et al. (2013), which we found interesting; however, it only deals with temperature sensing.

Page 8 line 6: "on the observing practices intricated with sensor limitations" use a word other than 'intricated'? I think you mean 'combined with' or 'complicated by'?

Yes, we mean "complicated by' in the sense that it may be difficult to tell if missing humidity data in a sounding report results from limitations of a specific humidity sensor (nominal working range) or rather to reporting practices related to sensor limitations; 'combined with' is suitable.

Page 8 line 10: Leslie Hartten and her team did a very nice job of outlining present-day daunting logistic, communication and scientific challenges of radiosonde operations in remote environments, e.g. Earth Syst. Sci. Data, 10, 1165-1183, https://doi.org/ 10.5194 /essd-10-1165-2018. One expects their data to appear in IGRA v3?

That paper is quite illustrating. It is cited now, with few words in sense indicated by you.

It would be wonderful if IGRA could include observations from such field campaigns. But for now, "data sources containing either field campaign observations or fewer than two years of data were set aside because of both their limited contribution to the dataset as a whole and the often considerable effort required for reformatting any one data source" (Durre et al. 2018)

P8, L9 – Added text:

'Radiosonde operations in remote environments, particularly performed from ships, present their own challenges; Hartten et al. (2018) give a vivid illustration.'

Page 8 line 21: confusing section. I think you miss a ')' after the Durre 2016 reference? The reference to WMO stations also introduces confusion because all or most of the 2700 IGRA stations already mentioned carry WMO station ID numbers. What distinction do you draw between "derived data for a selection of WMO stations" and standard IGRA station data?

Apart from the fact that we have missed the parenthesis after the Durre el al. (2016) reference, the sentence is accurate. There are hundreds of stations in IGRA that do not have a WMO ID number. The distinction between sounding data (raob and pibal) and derived data for a selection of WMO stations (derived from raob at surface level and standard levels) is inherent to IGRA, with separate datasets. However, since the latter is irrelevant to the paper, we have deleted the first part of the sentence, which now reads:

'This paper concerns with sounding data (Durre et al. 2016), comprising over 45 million soundings from 2761 (2662 fixed and 99 mobile) stations [based on data accessed in September 2017].'

Page 8 line 23: again, confusing. "and the former version". 'Former version' refers to IGRA? If so, state it clearly.

Yes, it refers to IGRA 1, as stated now.

Page 8 line 25: 'prior' not "priory".

Corrected.

Page 9 line 2: 'also' not "too".

Replaced by 'mostly'.

Page 9 lines 12, 13 "In wind data from pilot-balloons, the vertical coordinate is geopotential height (presumably adjusted from geometrical height measurements and the gravitational field)." Because this discussion refers to wind data, why do we care? Delete this sentence?

We find this information useful to distinguish PIBAL from RAOB data in IGRA. In IGRA files (one file per station) the sounding data are arranged in the same way no matter if

they contain PIBAL or RAOB data. RAOB uses pressure as default vertical coordinate, although geopotential height may be given. Pressure is only present in 28 PIBAL stations (0.03% of all PIBAL stations), as noted on footnote 5. Note: The word "presumably" means that we cannot ascertain if geopotential height was always properly calculated; since nothing proves the contrary either, that word was removed.

Page 10 line 11: "virtually" Not sure what you mean here, but perhaps you do not need this word?

We mean almost or practically. Since we give the end year (2016), it can be deleted.

Page 10 line 20: "the second world war" usually capitalized as 'second World War' or more often designated as World War II.

Corrected to capitalized and shortest form.

Page 10 line 20: 'tripled' not "triplicated".

Corrected.

Page 11 lines 1,2: I think you intend that readers should compare total numbers here? E.g. for 800 to 900 stations at each twice per day one would expect 1600 to 1800 soundings. Instead one observes consistently fewer than 1500 soundings per day, perhaps more like 1200 soundings. If so, be explicit. Do not assume readers will imagine your intentions. Tell us what you want us to see.

Your numbers refer to the years after around 1970, when the numbers become stable. Numbers of observing stations and of raob per day for other years are quite different, while the conclusion is the same since around 1955.

In the table below, NO-T (NO-H) denotes missing obs. for temperature (humidity) in %

```
YEAR:  1945   1955   1965   1975   1985   1995   2005   2011
NO-T:  76.8   25.5   24.7   13.2   14.8   26.9   26.0   23.6
NO-H:  77.4   30.5   24.6   13.6   14.9   27.0   26.0   23.6
```

(Note: for 1950, NO-T and NO-H exceed 60%)

Averaging the above values for the years 1955 through 2015, we get NO-T = 18.4% and NO-H = 22.9% – i.e., roughly 20% for both parameters (1 in 5 soundings). Note that this estimate agrees with Figure 6 for the selected stations 'IGRA-RS'.

We prefer to present a crude estimate rather than overloading the reader with numbers, and suggest the following adjustment:

"roughly 1 in 5 days during the year, on average, as it can be concluded by comparing Fig. 1a (RAOB) with Fig. 1b (TEMP, HUM)."

→

'roughly 1 in 5 days during the year, on average for the years after the mid-1950s [as concluded by comparing the yearly number of observing stations (TEMP, HUM in Fig. 1a) with half the global number of daily observations (TEMP, HUM in Fig. 1b)].'

Page 11 line 5: but you have just told readers that IGRA avoids soundings without valid temperature data. So, now, by RAOB, you mean valid temperature, RH, pressure, etc? But technically a RAOB might include temperature but not RH or vice versa? So in fact you mean 'valid' or 'complete' RAOB messages, instead of generic RAOB? If I get confused, readers will get more confused.

The identification of RAOB reports in IGRA, as opposed to PILOT reports, is essential for selecting the IGRA stations of interest to study humidity data and to derive our dataset. Both acronyms are defined in the beginning of section 2, on PP 8–9.

Technically, all RAOB messages include pressure, temperature, humidity and wind at several pressure levels (wind measurements require an external tracking device; only the vertical position is derived from radiosonde data). But humidity may be absent, as explained in Sections 1.1 and 1.3. This happens more often in the highest sounding levels, as shown in Section 3.3 (Fig. 8b), but it can happen also in the lower and middle troposphere. In addition, most radiosonde measurements prior to 1945 did not included humidity. This is shown in Fig. 1. Apparently, some stations begun to measure humidity much later than the advent of radiosondes. For example, the Lindenberg station was pioneering in measuring air temperature at fixed pressure levels (using meteographs before at least 1930). But humidity data for this station begins in 1958 in IGRA; it is very unlikely that this is has to do with IGRA's quality assurance procedures.

RH and DPD cannot be obtained without temperature measurements, of course. IGRA takes this into account in its quality checks by removing humidity data which is not accompanied by valid temperature data.

In short, a RAOB message must have at least temperature data, while humidity and/or wind may be missing. So, temperature data, together with pressure, clearly identifies a RAOB. While many IGRA stations have only PIBAL data, many other stations have both PILOT and RAOB data during their period of record – with PILOT (i.e., non-RAOB) data denoting the earlier years in many cases.

On page 9 we explain how the 'IGRA-RS' stations are selected based on the percentage of RAOB out of the yearly soundings of any kind (RAOB + PIBAL).

However, we understand that the manuscript might be unclear in some parts and hope that the following changes bring clarity. (Please see also response to comment on page 17, line 11.)

P10, L4-6 – Rewording:

"Figure 1a shows the yearly number of the RAOB stations – i.e., the stations with radiosonde observations at any time of the year, at least of temperature, regardless of humidity/wind – and of the PIBAL stations – i.e., the stations measuring only wind throughout the year (assuming data are not lost)."

[Figure]

'Figure 1a shows the yearly number of stations reporting RAOB any time of the year – meaning they have at least observations of temperature regardless of the simultaneous humidity/wind observations –, and of the stations reporting PIBAL observations alone – i.e., reporting only wind throughout the year.'

P9, L10-13 – Rephrasing, additions and shift to a separate paragraph:

"In radiosonde data, consistence between pressure and geopotential height (whenever the latter is reported in source data) is assured in IGRA since its first version. In wind data from pilot-balloons, the vertical coordinate is geopotential height (presumably adjusted from geometrical height measurements and the gravitational field)."

[Figure]

'Note that a RAOB message must have at least temperature data at several pressure levels, while humidity or wind data may be missing, and geopotential height is not always given. The recording of pressure levels, and consistence between pressure and geopotential height whenever the latter is reported in source data, has been assured in IGRA since its first version. IGRA uses a consistent data format, irrespective of the provenience of the data (PIBAL or RAOB). Therefore, RAOBs in IGRA can be simply identified by the presence of temperature data. Wind observations from pilot-balloons (PIBAL) have only wind data at several geopotential heights (adjusted from geometrical height measurements and the gravitational field).'

P10, L4 – Word addition:

"because temperature is required to measure RH."

[Figure]

'because temperature is required to measure RH or DPD and so all humidity data in IGRA are accompanied by temperature data.'

Page 11 line 7: "relatively few RAOB data". Here, clearly, you define RAOB as consisting of full valid all-parameter data (T, RH, pressure, etc.). But earlier you have defined RAOB as a generic radiosonde observation, quality unknown. To understand your selections and corrections, you need to adhere to, and give the reader, a clearer definition of what you mean by RAOB. Any sounding data, or only a valid full-parameter sounding? Which definition you intend makes a very large difference. Making the definition clearer will greatly improve the text of this manuscript.

The phrase "some of them contribute with relatively few RAOB data" means that some IGRA-RS stations have an amount of RAOBs below average, because of the relatively high percentage of PIBAL observations in their period of record. The term RAOB is always used as an abbreviation for radiosonde observation.

Page 11 line 12 "the IGRA-RS subset retains practically all the RAOB soundings" Because of confusing definitions, how could any IGRA-RS subset not automatically consist

entirely of valid RAOB data? I think the point you mean to make here is that taking only the radiosonde fraction of IGRA, e.g. IGRA-RS, still retains most of the original IGRA RAOB data. If so, then your earlier definition of a RAOB as a radiosonde observation (page 8 line 15) seems again confusing. Some RAOBs are not valid RAOBs? Or, some RAOBs are not valid radiosonde data, even though you define a RAOB as a radiosonde observation? I understand confusing often inconsistent meteorological terms, but here you have amplified the confusion? Readers need your best guidance but do not get it.

By "IGRA-RS subset" we mean the subset of IGRA stations which we refer to as "IGRA-RS" and is previously defined in Section 2.2 (as well as in the Abstract with less detail). However, despite discharging pilot-balloon stations (without any RAOBS), IGRA-RS does not automatically consist entirely of RAOB data. Some IGRA stations have performed a certain number of PIBAL observations during its period of activity, as explained in the beginning of Section 2.2. IGRA-RS includes those stations because the primary goal of our work is to derive a dataset based on data content for each station (not to extract humidity data from IGRA, which is up to users based on our metadata). Nevertheless, as shown in Table 1, "the IGRA-RS subset retains practically all the RAOB soundings" of IGRA.

Page 11 line 16: "missing years are considered". I think you mean 'included' or 'included and identified'. 'Considered' does not tell us how your selection process treated missing years.

"Since missing years are considered" was replaced by "Since the humidity time series may be interrupted for long periods of time". Missing years are included in the statistics of record in Table 1, as explained in the remainder of the sentence: "the full period of record of one station may be segmented into two or more periods for humidity". We added '(both are rounded to years)' at the end of the sentence.

Page 11 line 17: here a reader learns that 1300 of 1700 stations (75%) carry WMO ID numbers. This does not explain nor accord with the statements on page 8 (noted above) about a selection of WMO stations".

The number 1300 clearly refers to WMO stations among the IGRA-RS stations listed in Table 1. To avoid confusion, the clause about "a selection WMO stations" was deleted on page 8 (although referring explicitly to the derived data set belonging to IGRA 2, such data are not used in our study).

Page 11 line 21: "integrating" I think you mean 'integrated into' or 'coordinated through'?

Changed to 'integrated into'.

Page 11 line 23: "together with the surface stations of the GCOS Surface Network" Why do we need this? Is this somehow relevant to the upper air humidity data? If not, omit?

It is only complementary, so it can be omitted.

P12 section 2.3: a very good description of the core motivation of this work! These questions should move to the top, even to the abstract. Readers should not need to wait until this point to understand the motivation!

Actually, the abstract includes all the five points, although in a much formal manner. More importantly, our work has two purposes: 1) to elucidate the completeness of the radiosonde humidity observations in global terms, using IGRA since this a wide-ranging archive; 2) to provide metadata describing the completeness of humidity observations for each station selected from IGRA. This two-fold purpose is stated in the abstract and clarified in the Introduction (P3, L13-28). Lastly, we prefer to have these five questions here. They are intended to motivate the reader to go through Section 2.3 [before looking the results of Section 3] which is central to the paper and is structured in five related sub-sections.

Page 14 line 25: here a reader again finds reference to 'RAOB' reports when in fact the discussion pertains to IGRA-RS? More confusion?

Not at all. The sentence refers to finding the vertical extent of humidity soundings from RAOB reports in general. IGRA retains source data if they pass certain quality checks. So, the discussion applies equally to the RAOB data found in IGRA – particularly in IGRA-RS, since this is the subset *of IGRA stations* that contains practically all RAOB data.

Page 14 line 29: here the authors include moving stations but earlier, under global coverage, they only included fixed IGRA stations. Readers need better information about which subsets used when? If moving stations only a small fraction of total IGRA stations, why include them in this analysis? What value, if any, do they add?

Moving stations (about one hundred) form part of IGRA and so they are included in the dataset introduced in our work. Although individual time series are relatively short-lived in many cases, they have made an important contribution to upper-air observations.

The analysis provided by paper serves the purpose of illustrating the completeness of radiosonde humidity of observations on average terms. The dataset contains detailed information for each station.

Among the several aspects regarding completeness of observations, global coverage and temporal completeness are the ones which are expected to depend more on latitude. Including moving stations in a statistical analysis by latitude bands is impracticable. That is why Sections 3.1 and 3.2 excludes moving stations.

Concerning methodology, the use of only fixed station or all stations in the analysis of sounding data was stated in each of the five subsections of section 2.3 [see P12, L28; P14, L9; P14, L29; P16, L13; P16, L26]. However, for clarity we do now the following word additions:

P13, L27: "Eq. (1) was applied to the IGRA-RS stations" → 'We have applied Eq. (1) to the IGRA-RS fixed stations'

P15, L4: "humidity soundings" → 'humidity soundings from all IGRA-RS stations (including mobile)'

P16, L13: "percentage of stations" → 'percentage of stations (fixed and mobile)'

Page 16 line 2: 'shown' rather than "show".

Corrected.

Page 16 line 7: increases in a step-wise manner

We take the chance to be accurate by rephrasing:

'is almost step-wised, with a discontinuity around 1992" → 'increased almost as a step-function around 1992'

Page 16 line 20: "readiness"? I do not understand this word in this context. Change, please.

Corrected as 'To simplify'

Page 16 line 23: now we have "RS" stations. So, IGRA, IGRA RS, RAOB, RS. Do the authors follow a deliberate plan to confuse readers?

We can only agree that the acronym "RS" is superfluous and was used in an ambiguous way. It is now replaced by one of two different words (either 'radiosonde' or 'IGRA-RS') throughout the manuscript, depending on context:

P16, L23: "RS" → 'radiosonde'

P16, L24: "RS" → 'IGRA-RS'

P22, L5:  "RS" → 'radiosonde'

P22, L10: "RS" → 'IGRA-RS'

P22,L24:  "RS" → 'IGRA-RS'

Page 16 line 29: "repeated, by restricting" Remove this comma.

Removed.

Page 17 lines 3 to 5: finally, here, a reader learns about fixed versus mobile and which analysis used which subset. We should have had this information much earlier, at the start of section 2 or even as part of the introduction?

The lines in question specify the number of mobile and fixed stations involved in the results shown in Section 3. Section 2.3 (describing the method of analysis of IGRA data) provides this information beforehand. We hope this is now clearer in our revision: please see the changes in that section, detailed in our response to the comment on P14, L29. But we agree this may not be enough.

The same information is now made clear in the summary of contents given at the end of the introductory part of the Introduction. We take the chance to outline the selection of IGRA stations in that same part, introducing the term IGRA-RS (after the Abstract) even though this is formally defined in Sect. 2.2.

P3. L33 – P4, L4:

"Section 2 indicates the IGRA data set used in the study and explains the data analysis. Section 3 presents a global picture of the completeness of humidity observations over the years, as derived from the IGRA stations reporting a minimum of radiosonde data, i.e., discharging stations with practically wind-only data in their period of record. Section 4 provides the definition of the metadata parameters describing the completeness of humidity observations from each relevant IGRA station – both in terms of annual statistics and in individual soundings – and the format description of the corresponding data sets. (…)"

→

'Section 2 indicates the IGRA data set used in the study; selects the IGRA stations reporting a minimum of radiosonde data (coined as 'IGRA-RS') by discharging stations with practically wind-only data in their period of record; and explains the data analysis. Section 3 presents a global picture of the completeness of humidity observations over the years, as derived from the IGRA-RS stations. The geographical coverage and temporal completeness of annual observations are detailed by latitude bands, thus restricting to fixed stations, while other aspects on the global scale include both fixed and mobile stations indistinctively. Section 4 provides the definition of metadata parameters describing the completeness of humidity observations from each IGRA-RS station – either as annual statistics or for individual soundings – and the format description of the corresponding data sets. (…)'

Page 17, line 11: "IGRA-RS excludes the IGRA stations without any RAOB at all" this statement is NOT consistent with earlier use of or definitions of terms RAOB and IGRA-RS. I suspect the authors know what they intend, but they have only confused their readers.

RAOB stands for radiosonde observation(s), as defined on P8. IGRA-RS refer to the IGRA *stations* selected as described in Section 2.2. By excluding the IGRA with less than 5% RAOBs out of the annual soundings in every year [equivalent to retaining the stations with at least 5% RAOBs in any year], PIBAL *stations* are immediately removed (meaning the IGRA stations with only pilot-balloon observations during their period of record, i.e., without ant RAOB).

Please note that a single station can perform PIBAL observations during part of its period of activity and RAOB during other part. Thus, some IGRA-RS stations do have a significant fraction of PIBAL observations besides RAOB. Our analysis is based on all data from IGRA-RS stations. It is impossible in many cases to classify a station as a RAOB or PIBAL station – except for the sites which performed only PIBAL launches during their existence. Of course, some stations have performed consistently radiosonde launchings since they were open. Usually the term "radiosonde station" refers to sites that have taken RAOBs, even if not always (particularly at stations with a long history). The term "RAOB station" was used by us exceptionally in relation to Figure 1a, to denote a station reporting any RAOB during a specific year. Conversely, the stations not reporting RAOB during a given

year, but only PIBAL observations, were called "PIBAL stations". This terminology was convenient to study the yearly number of stations reporting either RAOB or PIBAL data. Now we understand that this terminology is quite confusing, which demands minor but significant changes in Figure 1 – by detailing the figure caption, removing unnecessary information and correcting the figure legend in panel-a – and in the related text on P8.

P8, L3-4

"RAOB stations" → 'stations reporting RAOB'

"PIBAL stations" → 'stations reporting PIBAL observations alone'

Figure legend of Figure 1a:

"RAOB" → 'TEMP'

"Higrom." → 'HUM'

"Higrom. > 95%' → 'HUM > 95%'

"PIBAL" → 'WIND-only'

For the related changes in figure caption, see p. 40 in the marked-up manuscript.

Page 17 line 11: "comparison between Fig. 3 and Fig. 1a" Following (correctly, I hope) the authors' intent, extrapolating from Fig 1a, in Fig 3 I should see, between 1955 and 1975, a 10-fold increase in non-humidity (e.g. PIBAL) stations - which my eyes do NOT see - and, between 1975 and 2015, a steady number of humidity stations accompanied by a decreased number of PIBAL stations. Why do I not see the 1955 to 1975 differences? Bad eyes? Bad figure? Can I actually confirm the drop in number of PIBAL stations in 2015 relative to 1975? Authors should provide guidance to readers about what the authors wish readers to see, and ensure that Figures support that evidence? Not clear in this instance?

As stated both in the text and figure captions, Figure 1 refers to IGRA whereas Figure 3 refers to IGRA-RS. IGRA has almost 1000 PIBAL stations (no RAOB in period of record), while IGRA-RS excludes them all. So, to answer to the objection, the evolution of PIBAL stations is not expected to be seen in Figure 3.

We wish readers can understand the difference between the total number of IGRA-RS stations (red crosses + blue dots) and the number of IGRA-RS stations reporting humidity (blue dotes), as both numbers are given above each map of Figure 3. But we admit that drawing a conclusion by comparing Figures 1 and 3 is not straightforward. The argument we had in mind is the following: Figure 1 shows that the relative number of IGRA stations with temperature data any time of the year but without humidity data at all (relative difference between the black and solid blue lines in Fig. 1a) is very small. Thus, most of the stations not reporting humidity in a given year must report PIBAL data rather than temperature data.

This is now clarified on P17, L11:

"Although IGRA-RS excludes the IGRA stations without any RAOB at all, a comparison between Fig. 3 and Fig. 1a indicates that most of the IGRA-RS station-years without humidity data correspond to periods of wind-only observations."

[Figure]
 →

'The IGRA-RS retains practically all RAOB data of IGRA; however, some stations have years with only PIBAL observations. Since almost all of the IGRA stations measuring temperature in a given year do also measure humidity at least part of the time (as seen by comparing black and solid blue lines in Fig. 1a), it is clear that most of the IGRA-RS station-years without humidity data (red crosses in Fig. 3) correspond to years of PIBAL observations alone (no RAOB).'

Page 17 section 3.1 At the low given resolution of Figure 3 (it does zoom in nicely on my screen), the reader doubts whether we can confirm the temporal changes in geographic patterns described by the authors. A reader almost certainly has zero ability to detect "four fixed weather-ships". We either need descriptions better scaled to the maps or better maps.

The size of dots and crosses in Figure 3 was optimized to resolve the location of stations, without merging the ones very close to each other. Since ocean weather-ships are not marked on the maps (being very few in the years represented), we think it is better to remove details about their varying number when describing specifically each map. Also, the lines about the importance of ships of any kind is now in a separate paragraph and was slightly revised. Please see changes on pp. 18-19 of the marked-up manuscript.

Page 17 line 29: observations reported by Driemel actually included a large fraction gathered during transit, e.g. north and south along the Atlantic oceans, with perhaps the largest fraction between 60N and 60S? Polar yes, and very valuable, but not exclusively polar.

We agree that Atlantic Ocean regions are worth mentioning. Driemel et al. (2016) also reported a large proportion of radiosonde launches at latitudes above the polar circles. Note: 29 Dec 1982 was reported as the date of the first radiosonde launching. However, data in IGRA begin in 1985, with an interruption of 6 years between 1994 and 1999 without any data. Maybe missing years have insufficient radiosonde data to be in IGRA.

P17, L28:

"The polar missions between 1985 and 2014 of the ice-breaker and research vessel Polarstern (Driemel et. al, 2017) (…)"

[Figure]
 →

'The missions to the Arctic and Antarctica performed by the ice-breaker and research vessel *Polarstern* (Driemel et al., 2016), covering also Atlantic Ocean regions during transit, provided substantial radiosonde humidity data in the periods 1985–1993 and 2000–2014 (…)'

Page 18 line 6: use arctic or Arctic, but at least use it consistently. Copernicus style sheet suggests 'Arctic'.

The adjective aṛctiҫ was replaced by the noun Arctic as part of the changes related to changes in Figure 4 (see next response)

Page 18 and Figure 4: Whatever the authors may have intended here, they have largely failed. Figure 4 remains almost impossible to understand, readers need to spend way too much time trying to understand it. What the authors' claim as climate zones actually represent latitudinal bands instead, and not evenly distributed in any case. Properly speaking, climate zones include elevations, distance from coastlines, location with respect to monsoonal circulations, etc. We get (combined) 46 degrees of equatorial, 23 degrees of sub-tropical, 60 degrees of temperate and 46 degrees of polar (using my own guesses at names for the zones). Figure 4 exacerbates this confusion, with quantities and lines in no particular order or calibration. How do we compare a northern sub-tropical range of 12 (23 to 35) degrees with an equatorial region of 46 degrees or a south polar region of 23 degrees? I have no doubt the authors understand the data to the resolution of "two ships reporting radiosonde observations in waters around the Arctic Circle" (page 18 line 9) but readers will not find anything like that detail in these figures. Going back to questions on page 12 (and notice there that the authors used the word "latitudes" rather than climate zones) do we really need any of this detailed location by location discussion? Instead, authors could help readers by defining latitude zones appropriately (30, 60, 90, etc.) or at least of equal latitudinal extent and then draw our attention to temporal patterns within those zones. Describe the data sufficiently so that subsequent users can explore specific latitudinal or zonal features based on their own criteria?

Concerning Figure 4

Figure 4 was intended to show how the number of humidity-reporting stations evolved with time in different latitudes, and to highlight hemispheric differences. We agree that is was not easy to read because of the large number of latitude bands plotted; and that the curves could not be compared between each other, except for homologous curves in opposite hemispheres.

So, Fig. 4 was modified by restricting the plot to four latitude bands of equal area, two per hemisphere: 0–30° and 30°–90°, representing tropical and extratropical latitudes on each hemisphere. In addition, the vertical scale changed from linear to logarithmic to better detail the rapid grow in the early years.

Concerning terminology

By "climate zones" we mean the Earth´s major climatic zones, which can be schematized as follows: tropical zone → region between the tropic circles (0–23.5°N/S); subtropics → 23.5°– 35°; temperate zone → 23.5°–66.5°; polar zone → latitudes above the polar circle (66.5°–90.5°). We know that it's possible to refine the definition of these major zones based on climatic parameters and surface features, so that they are not exactly

belt-shaped. Although the average boundary (mean latitude) between each other is arguable, the values given above are often used, among others certainly. The subdivision 30, 60, 90 is a modern simplification that rounds latitudes and neglects the subtropics; that is precisely why we used it later in Section 3.2, Figures 6-7 (further combining hemispheres since hemispheric differences are irrelevant in that context).

However, to avoid possible confusion with climate zones of specific areas of the globe (closely related to climate types), in the revised manuscript we replace the ambiguous term "climate zone" by 'latitude band'.

Page 18 throughout: "Tropics", "extratropics", "climate zone", "latitude band" – terminology and punctuation very inconsistent throughout this section. Needs careful attention and correction to achieve consistency as well as accuracy. Really too many to note them all, needs thorough scrubbing and appropriate revisions. Authors make appropriate notice of land to ocean differences by hemisphere but then compare Arctic (ocean) with Antarctic (land) without any such qualifications.

Maybe the reason why this page of Section 3.1 seems inconsistent in punctuation is that the paragraph break appearing in line 10 should be at the end of line 15. However, we must recognize that page 18 has deficiencies regarding clearness.

The comparison between the Arctic and Antarctica refer to Figure 4, appearing before the discussion of Figure 5 which is when land to ocean differences are considered.

We keep the detailed latitude bands only in Figure 5 for the sake of detailing the stations density by latitude. But we do not use the term "climate zone" anymore.

Apart from correcting the paragraph break, the changes in the manuscript consists in adapting the discussion of Figure 4 (modified as explained in the previous response), defining more precisely terminology, and caring about clarity and accuracy.

We also adjust the first bullet paragraph of Section 6, due to corrections to Section 3.1.

Please see the changes on pp. 18-20 and p. 28 of the marked-up manuscript.

Page 18 line 30: "While the same is impracticable in many other parts of the world and over the oceans, distances up to two or three times larger than ideal are accepted, in view of the relatively mild climatic conditions on oceans and the fulfillment from surface and satellite observations." I believe the authors intend this as a description based on practical realities but many researchers would not agree that we should find the situation acceptable? We certainly need sea surface temperatures, surface roughness, cloudiness and rainfall, and interior ocean temperatures (e.g. by Argo) at much higher temporal and spatial resolution. Last sentence in this paragraph (e.g. top of page 19) also contradicts this statement? Statement represents a lightning rod, authors might do themselves a favor to omit it?

We understand the objection, although we have written "accepted", not "acceptable". Yes, the description intends to represent reality. To circumvent misinterpretation, we rephrased the last part of the quoted sentence:

P18, L30: "are accepted, in view of the relatively mild climatic conditions on oceans and the fulfilment from surface and satellite observations." → 'need to be filled by satellite-based data and supplemented by surface observations, which are generally much denser than radiosonde stations.'

The last sentences in the same paragraph (end of page 18, top of page 19) are intended to quantify and stress the poor coverage over oceans even for climatic purposes. This is in line with the concern with spatial resolution. To highlight this point, we made slight but significant changes:

P18, L32: "On a scale suitable for climate monitoring, the WMO recommends" → 'Nevertheless, on a scale suitable for climate monitoring, the WMO recommends'

P19, L2: "indicates larger distances in most oceans" → 'indicates that fixed stations are too far apart in most oceans'

Page 19 line 9: "Besides, the corresponding …" Delete the first word.

Deleted.

Page 19 line 14: "we only care with sub-year missing days" I think you mean 'we focus only on sub-year'?

Absolutely. Corrected.

Page 19 line 15: "Fig. 7 gives a glint of the typical continuity …". I think you mean 'Fig 7 offers a summary of the typical ….' Or 'offers an indication'.

Changed to 'summarizes'

Page 19 line 16: "between 1945 and 1960" In fact, Fig 7 shows that number of missing days dropped much faster than your phrases suggests, over not more than 5 years. The pattern looks like an initiation or spin-up problem, which you have hinted at elsewhere.

Indeed, the fastest decrease seems to be in period 1945–50, although the curves of Fig. 7 are very noisy in subsequent years until 1960.

P19, L16 - Clause changed as follows:

"the average size of missing days decreased from 4–6 months to about 1 month between 1945 and 1960"

→ 'the average size of missing days dropped from 4–6 months to about 1 month between 1945 and 1960; much of this change occurred before 1950, indicating that radiosonde measurements became rapidly regular in the early years'

Page 20 line 12: 3/4 (also in line 13). Please use percentages as you do elsewhere, not fractions.

Converted to percentages. See next response for other changes in the same paragraph.

Page 20 line 13: "data reach 22 km". In the previous line you gave us pressure then altitude, e.g. 100 hPa roughly 10 km. Here you should do the same, e.g. something like 50 hPa roughly 22 km.

IGRA et al. (2006) used observed pressure, while our results (P20) uses computed height. (Incidentally, we have wrongly given the approx. height of 250 hPa instead of 100 hPa.) So that readers can clearly understand the terms being compared, we have rephrased and shortened lines 10-15 as follows:

'According to Fig. 8b, by 2003, 75% of the temperature soundings with surface data reached an altitude of 22 km, i.e. ~ 50 hPa. Note that Durre et al. (2006) reported that, by the same year, 74% of all IGRA 1 soundings reached at least the 100-hPa level, i.e. ~ 16 km; this lower height is due to the inclusion of PIBAL data in their analysis of IGRA 1, whereas our analysis is restricted to RAOB in IGRA 2.'

Page 20 line 14: "This difference " refers to differences in maximum height or to differences in height in temperature records versus humidity records? Need clarity here.

The text was clarified as explained in the previous response.

Page 20 line 23: "This last feature and is coincident " remove the word 'and'

Typo corrected.

Page 20 line 30 - use percentage not fraction.

In this context, we think that "¾" (opposed to "half" in the same sentence) is a nice way to mean about 75% without needing to give an exact percentage. To be consistent, we prefer 'three quarters'

Page 20 line 31, 32 "fairly recent measurements in the upper-troposphere, and certainly more above too, was considered inadequate for climate". I think you mean ' and certainly into the lower stratosphere'? Recent as used here means before the Durre 2005 reference, so not the most recent. No widespread globally-useful solution, certainly, but other people have worked on this problem?

The words "fairly recent" refer to 2003, when the international Workshop to Improve the Usefulness of Operational Radiosonde Data (involving 30 data users and providers) took place, with conclusions reported in Durre et al. (2005). To our knowledge, things have not changed significantly in the meantime, nor the problem has been specifically addressed in a manner as to clarify the real value of humidity measurements close and above the tropopause – despite more evidences from observational studies using parallel soundings with standard sensors (like the frost point hygrometer) as reference.

The words "and certainly more above" were added by us (as an inference) and can be deleted. To be a bit more accurate in the citation, we have rephrased the sentence:

"; however, the accuracy of fairly recent measurements in the upper-troposphere, and certainly more above too, was considered inadequate for climate studies and a challenge for future operational radiosondes (Durre et al., 2005)."

→ '. However, only 15 years ago, international experts pointed out that the accuracy of current operational measurements of humidity in the upper-troposphere was inadequate for addressing climate variability and change (despite the usefulness of some sensors) and a challenge for future operational radiosondes (Durre et al., 2005)'

Page 21 line 14: Most mobile soundings come from ships and for those the baseline elevation is always sea level plus/minus 10 m at most?

A very good suggestion to improve the dataset in the next updated version, by replacing missing values for TOPZ (max. height of hum. measuring level, rounded to decametres) where appropriate. This will only affect a tiny percentage of the metadata for individual observations, not the metadata by year which uses pressure instead of height. However, assuming that baseline for our calculations require caution when a ship-based vertical sounding has no other upper-air humidity except at the level of the balloon release or very close it.

We made the following corrections (please see different contexts):

P14, L29:

"For moving stations, the elevation of station is taken equal to the geopotential height at the surface level, if given (otherwise, the vertical extent was not calculated)"

→

'For mobile stations (ships and buoys), the elevation of the stations can be approximated to zero, unless the vertical extent of the sounding is too small, requiring data for the balloon release height.'

P20, L9:

"Also, the soundings from mobile stations with missing geopotential height at the surface level had to be excluded."

→

'In addition, the soundings from mobile stations with missing geopotential height data at the surface level are excluded, for consistency with the hypsometric calculations used in the dataset presented in this paper.'

P21, L14:

"The same applies to soundings from mobile stations, since the station's elevation is variable, and the geopotential height of the surface level is missing in 30 % of the corresponding RAOB"

→

'The same applies to soundings from mobile stations if geopotential height is not given at the surface level (it is missing in 30 % of the corresponding RAOB). Although the local sea level is normally within ± 10 m from mean sea level, taking zero as the baseline height can lead to large relative errors if humidity is only measured very close to the radiosonde station elevation (balloon release height)'

Page 22 lines 5,6: Confusing. I think you mean that, for a fairly high standard such as expecting 95% of stations to have valid lower-troposphere humidity data, the number of stations meeting that standard remains very low from start of the records in 1945 to as recently as 1990. You use P in percentage to indicate fraction of stations while many readers familiar with statistical analysis will understand P as probability. You need to give explicit explanation of what you describe and how you measure it.

P represents the percentage of Sfc-to-500hPa humidity soundings, out of all soundings with humidity in each year from an arbitrary station (P is defined on the top labels of both Figure 9 and Figure 10). Note that Sfc-to-500hPa humidity soundings are defined in two alternate ways just a few lines above (see end of page 21). The lines in question refer to the strict definition 'HUM-A'.

What the text means is that for a very high standard such as expecting to have more than 95% HUM-A soundings, the number of stations meeting that standard (shown in Fig. 9a) remains much smaller than the total number of radiosonde stations performing observations (shown in Fig. 1a) from 1945 to as recently as 1990.

To improve clarity, we made the following amendment:

P22, L3-6

"Figure 9a shows the evolution of the number of stations with the percentage of Hum-A soundings exceeding given values, out of all soundings with humidity in each year – since nearly the time radiosonde humidity data at the surface level are first available. Comparing Fig. 1a with Fig. 9a, we can see that between 1945 and 1991 only a small fraction of the RS stations carried out Hum-A observations in most of the soundings, say, within a percentage range P > 95 %."

[Figure]

'Let P be the percentage of Sfc-to-500hPa soundings (HUM-A or HUM-B, at our choice), out of all humidity soundings from an arbitrary station in a given year. Figure 9a shows the evolution of the number of stations with P for Hum-A soundings exceeding given values, since nearly the time radiosonde humidity data at the surface level are first available. Comparing Fig. 9a with Fig. 1a, we can see that between 1945 and 1991 only a small fraction of the radiosonde stations carried out Hum-A observations in most of the soundings, say, with P within the range P > 95 %.'

Page 22 line 19: "two noteworthy change points: a sudden increase around 1970 and 2000". Singular plural problem: two change points result in sudden increases around 1970 and 2000.

Changed to 'two noteworthy change points: a sudden increase around 1970 and another one around 2000'

Page 23 line 25: "Evidently, Fig. 11 is only …" Remove the first word.

Removed.

Page 23 line 25: "the question of since when we have enough" remove the word 'since'

Correct, curves in Fig. 11 are not all monotonic. Word removed.

Page 25 lines 10, 11: "selecting the stations with a minimal amount of radiosonde" I think you mean by selecting those stations with a sufficient amount? Minimal as you use it in this case technically indicates few or fewest, not what you intend. You mean 'exceeded minimal standards'?

Given the changes related to the next comment, the word is no longer needed.

Note - the first five sentences of section 6 make a very good abstract, much better and more concise than the one you have.

We agree that the style of the first lines of the concluding section make a good abstract. The Abstract cannot be abridged too much since it includes other important elements: what do we mean by completeness of humidity observations; what is the content of the dataset presented by the paper; and how it may be useful to the scientific community.

Upon careful consideration, we hope to have reached this time a compromise between concision and comprehensiveness. The abstract is much shorter now.

Please see the changes to the Abstract in the marked-up manuscript, as well as the related changes in section 6 (appearing there on p. 28).

Again, arctic vs Arctic. Please check and correct!

Referring to page 26, we now have only 'Arctic'. This first bullet paragraph was revised to fit the revision of section 3.1

Page 26 line 9 - use percentage not fraction

Maybe 'three quarters' is better, since we have "at present" while 13 km refers to 2016.

Page 26 line 22 "was not standard and it was rarely used" delete word 'it'

Deleted.

Page 26 line 23 "consecutive years of data until a given year" I think you mean number of consecutive years greater than some specified value? Later in that sentence, depends on the value, the time span and the completeness criteria.

We mean what we wrote, perhaps not well written. In other words, and giving more emphasis to the number of years of past data:

P26,L23 – Changed to:

'The amount of humidity time-series with a given number of consecutive years of data until a given year depends not only on the year and the length of record, but also on the completeness criteria.'

Page 26 line 24 "E.g.: the station-based" Do not start a sentence with an abbreviation. Instead, write it out: For example.

Corrected.

Page 26 line 27 "Evidently, the equivalent time-series …" delete the first word.

Deleted.

Page 38 - why do we get Figure 2 in black and while while we get other figures in useful helpful colors?

Changed to colour-blind friendly solid lines.

Page 39 - Figure 3 still very hard to read at page resolution, but works okay with page zoom.

See previous response to comment on page 17, section 3.1.

Page 40 - Figure 4, latitude bands not climate bands, latitude bands inconsistent in extent, figure hard to read and harder to understand. Change to standard latitude bands as used in Figures 6 and 7.

The figure was reshaped two show number of stations is equal-area latitude bands in each hemisphere. "climate bands" changed to 'latitude bands'. Note that the more usual latitude bands used in Figures 6 and 7 do not show hemispheric differences since they are irrelevant for the case. In Figure 4, hemispheric differences are very important.

Page 41 - potentially useful information but limited by use of the variable latitude bands. Use standard latitude bands as in Figures 6 and 7?

For a figure representing the station density in different latitudes, we think that dividing the globe in major latitudes with climatic significance – chiefly defined by the Tropic and Polar Circles – will be more informative to many readers. The interval $23\text{-}5^{o}\text{–}35^{o}$ is a very common approximation for the subtropical regions.

Page 42, Figure 6 - at a std deviation of 20%, evidently no significant differences among day fraction by latitude. If not, combine the lines into a composite, both for the absolute value and for the standard deviations. E.g. no valid distinctions between these lines so why show them separately?

Considering the average values in Figure 6, we think that the differences between the tropics and extratropics are large enough to be highlighted. Making a 'composite' of the AVG and STD curves (AVG ± STD) would make the graph confusing; besides, AVG + STD sometimes exceeds a little 100%. This is not unusual in a descriptive statistic. In the case, a radiosonde station cannot have data in more than 100% of the days of the year; but the STD represents both the dispersion of the values above and below the mean.

Page 42 Figure 7 - see comment above about data that I made in reference to Page 19, above; needs color.

Color added. See text correction in our response above.

 - no differences statistically or visible between T and RH mean and quartile values in this panel, nothing gained by showing them super-imposed? Combine them? Or delete this panel and focus instead on panel b?

Panel a is important to show how vertical resolution of RAOB has evolved over time. The almost coincidence between the curves for T and RH demonstrates that temperature measurements alone, with missing humidity at any levels up to the maximum height of humidity measurements, are exceptional. We prefer showing it rather than just stating it.

 Why not color instead of grey-scale? Do we really need both 9 and 10, as they basically tell the same story? Panel a in Figure 10 could group to two lines, one before 1990 and one after? Comparison with Panel b would still hold?

Figure 9 uses absolute number of stations and gives a detailed temporal description. Figure 10 uses percentages of stations and a course temporal description. Although we give more emphasis to Figure 9, readers familiar with (inverse) cumulative distribution functions will probably prefer to see Figure 10. The evolution with time is an important feature also in Figure 10. Grouping lines as suggest would be less informative, and, yes, panels a and b will still be different.

Figures 9 and 10 changed to color.

 useful, but why not use the same color scheme both panels?

Changed as suggested.

 relevant to RAOB, IGRA, IGRA RS confusion above, here RAOB IGRA and RAOB IGRA RS have almost exactly identical numbers of soundings with humidity. Differences in total soundings only roughly 300 out of nearly 30 million, e.g. roughly 0.001% difference? What exactly drives the distinction in terms?

Please see above responses to comments referring to p. 11, line 12 and p. 17, line 11.

The negligible difference noted in the last comment shows that our selection of IGRA stations (i.e., IGRA-RS) retains practically all humidity data; in other words, the IGRA stations left out are irrelevant to our study. This point is elucidated in Sect. 2.2, namely on P11, L3-13. The percentage (99.999%) of RAOB present in IGRA-RS is now reported.

Two additional references mentioned above, authors to include if considered useful and relevant:

Wang et al., https://doi.org/10.1175/1520-0426(2002)019<0981:COHMEF>2.0.CO;2
Wang, J., et al., Geophys. Res. Lett., 40, 1231–1236, doi:10.1002/grl.50246

[revised manuscript text omitted]